# Satellite glial cells promote regenerative growth in sensory neurons

Oshri Avraham [1], Pan-Yue Deng[2], Sara Jones[1], Rejji Kuruvilla [3], Clay F. Semenkovich [2,4], Vitaly A. Klyachko[2] & Valeria Cavalli[1,5,6 ✉]

Peripheral sensory neurons regenerate their axon after nerve injury to enable functional recovery. Intrinsic mechanisms operating in sensory neurons are known to regulate nerve repair, but whether satellite glial cells (SGC), which completely envelop the neuronal soma, contribute to nerve regeneration remains unexplored. Using a single cell RNAseq approach, we reveal that SGC are distinct from Schwann cells and share similarities with astrocytes. Nerve injury elicits changes in the expression of genes related to fatty acid synthesis and peroxisome proliferator-activated receptor (PPARα) signaling. Conditional deletion of fatty acid synthase (*Fasn*) in SGC impairs axon regeneration. The PPARα agonist fenofibrate rescues the impaired axon regeneration in mice lacking *Fasn* in SGC. These results indicate that PPARα activity downstream of FASN in SGC contributes to promote axon regeneration in adult peripheral nerves and highlight that the sensory neuron and its surrounding glial coat form a functional unit that orchestrates nerve repair.

[1] Department of Neuroscience, Washington University School of Medicine, St Louis, MO 63110, USA. [2] Department of Cell Biology and Physiology, Washington University School of Medicine, St Louis, MO 63110, USA. [3] Department of Biology, Johns Hopkins University, Baltimore, MD 21218, USA. [4] Division of Endocrinology, Metabolism & Lipid Research, Washington University School of Medicine, St Louis, MO 63110, USA. [5] Center of Regenerative Medicine, Washington University School of Medicine, St. Louis, MO 63110, USA. [6] Hope Center for Neurological Disorders, Washington University School of Medicine, St. Louis, MO 63110, USA. ✉email: cavalli@wustl.edu

Unlike neurons in the central nervous system (CNS), peripheral sensory neurons with cell soma in dorsal root ganglia (DRG) switch to a regenerative state after nerve injury to enable axon regeneration and functional recovery. Decades of research have focused on the signaling pathways elicited by injury in sensory neurons[1,2] and in Schwann cells that insulate axons[3,4] as central mechanisms regulating nerve repair. However, virtually nothing is known about the contribution of the glial cells that envelop the neuronal soma, known as satellite glial cells (SGC), to the nerve repair process.

In adult animals, multiple SGC form an envelope that completely enwraps each sensory neuron soma[5,6]. The number of SGC surrounding sensory neurons soma increases with increasing soma size in mammals[7]. Each sensory neuron soma and its surrounding SGC are separated from adjacent neurons by connective tissue[5,6]. The neuron and its surrounding SGC thus form a distinct morphological unit[5,6]. SGC have been identified mostly based on their morphology and location. Several SGC markers have been characterized, including the inwardly rectifying potassium channel (Kir4.1)[8], cadherin 19[9], the calcium activated potassium channel (SK3)[10,11], and glutamine synthetase (GS)[5,12]. SGC also share several properties with astrocytes, including expression of glial fibrillary acidic protein (GFAP)[5,13] and functional coupling by gap junctions[5,14].

SGC have been studied in the context of pain responses and are known to modulate pain thresholds[14–17]. It is known that SGC are altered structurally and functionally under pathological conditions, such as inflammation, chemotherapy-induced neuropathic pain, and nerve injuries[5,18,19]. Communication between neuron soma and SGC via glutamatergic transmission could impact neuronal excitability and thus nociceptive threshold after injury[20]. Nerve lesions induce an increase in GFAP[13,18,21], p75, and pERK expression[22,23] in SGC. An increase in SGC ongoing cell division following nerve injury has also been suggested[13]. These studies suggest that SGC can sense and respond to a distant nerve injury and actively participate in the processing of sensory signals[24]. Whether SGC play a role in regenerative responses has not yet been established. This is largely due to the lack of molecular and genetic tools to study SGC. In this study, using single cell RNA-seq (scRNA-seq), we reveal that nerve injury alters the gene expression profile in SGC, which is mostly related to lipid metabolism, including fatty acid synthesis and PPARα signaling.

PPARs are ligand-activated nuclear receptors with the unique ability to bind lipid signaling molecules and transduce the appropriate signals derived from the metabolic environment to control gene expression[25]. After binding the lipid ligand, PPARs form a heterodimer with the nuclear receptor RXR, followed by binding to specific DNA-response elements in target genes (PPAREs). Three different PPAR subtypes are known; PPARα, PPARβ/δ, and PPARγ[26]. The committed enzyme in de novo fatty acid synthesis FASN[27] generates endogenous phospholipid ligands for PPARα[28] and PPARγ[29]. PPARγ activity has been associated with neuroprotection in different neurological disorders[30]. In rat sensory axons, PPARγ contribute to the proregenerative response after nerve injury[31]. In the CNS, astrocytes produce lipids far more efficiently than neurons and FASN was found in astrocytes but not in neurons[32]. Lipids secreted in ApoE-containing lipoproteins by glial cells appear to support growth in cultured hippocampal neurons and regulate expression or proregenerative genes such as Gap43[33]. In the peripheral nervous system (PNS), synthesis of phospholipids and cholesterol in sensory neurons is required for axonal growth[34]. Lipids can also be exogenously supplied by lipoproteins secreted from glial cells to stimulate neurite growth[33,35], possibly via ApoE, whose expression is increased in glial cells after nerve injury[36,37].

Whether SGC express PPAR or FASN and contribute to support sensory axon growth after nerve injury has not been determined.

Here, we demonstrate that conditional deletion of Fasn specifically in SGC impairs axon regeneration in peripheral nerves. Treatment with fenofibrate, an FDA-approved PPARα agonist, rescues the impaired axon regeneration observed in mice lacking Fasn in SGC. These results indicate that PPARα activity downstream of FASN in SGC represents an important mechanism mediating axon regeneration in adult peripheral nerves. These results also highlight that the neuron and its surrounding glial coat form a functional unit that orchestrates nerve repair.

## Results

**Profiling naïve and injured DRG at the single cell level**. To define the biology of SGC and to understand the role of SGC in nerve injury responses, we performed scRNA-seq of mouse L4,L5 DRG in naïve and injured conditions (3 days post sciatic nerve crush injury), using the Chromium Single Cell Gene Expression Solution (10x Genomics) (Fig. 1a). An unbiased (Graph-based) clustering, using Partek flow analysis package, identified 13 distinct cell clusters in the control and injury samples (Fig. 1b). The number of sequenced cells was 6541 from 2 biological replicates, with an average of 45,000 reads per cell, 1500 genes per cell and a total of 17,879 genes detected (see filtering criteria in the methods). To identify cluster-specific genes, we calculated the expression difference of each gene between that cluster and the average in the rest of the clusters (ANOVA fold change threshold >1.5), illustrated by a heatmap of the top 5 differentially expressed genes per cluster relative to all other clusters (Fig. 1c and Supplementary Data 1). Examination of the cluster-specific marker genes revealed major cellular subtypes including neurons (Isl1), SGC (Kcnj10/Kir4), Schwann cells (Periaxin, Mpz), endothelial cells (Pecam1/CD31), macrophages (Alf1/Iba-1, CD68), mesenchymal (CD34) connective tissue (Col1a1), T-cells (CD3G) and vasculature associated smooth muscle cells (Des) (Fig. 1b and Supplementary Fig. 1a). We then compared the cell clustering between naïve and injury conditions. Unique cell clusters were not altered by nerve injury (Fig. 1d). We found that the number of SGC and macrophages were increased after injury by 8% and 7%, respectively (Fig. 1e). Although prior studies suggested that SGC proliferate after injury[5,11,13], a recent study demonstrated that cell proliferation occurred in macrophages but not in SGC seven days after injury[38]. We thus examined expression of the cell cycle markers Mki67 and Cdk1 in injury conditions and found that these cell cycle markers were mainly expressed in macrophages and blood cells/monocytes but not in SGC (Supplementary Fig. 1b). Our results thus suggest that 3 days post injury, there is little SGC proliferation and the increase in SGC cell number we observed could be a result of tissue dissociation. We obtained a similar number of neurons in both uninjured and injured conditions, which represented about 1% of high quality sequenced cells (Fig. 1e). The recovered neurons included nociceptors (TrkA), mechanosensors (TrkB), and proprioceptors (TrkC) (Supplementary Fig. 1c). We anticipated obtaining a larger representation of neurons in our dataset. The dissociation procedure, which requires multiple steps of enzymatic and mechanical disruption to separate the SGC from neurons might affect neuronal survival. Our protocol is thus achieving recovery of SGC, but might not be suitable for the recovery of neurons from DRG.

**SGC are molecularly distinct from Schwann cells in DRG**. The scRNAseq results indicate that SGC represent the largest glial subtype in the DRG (Fig. 1e). Overlaid cells in t-SNE plots with marker genes for SGC and Schwann cells show the relative levels

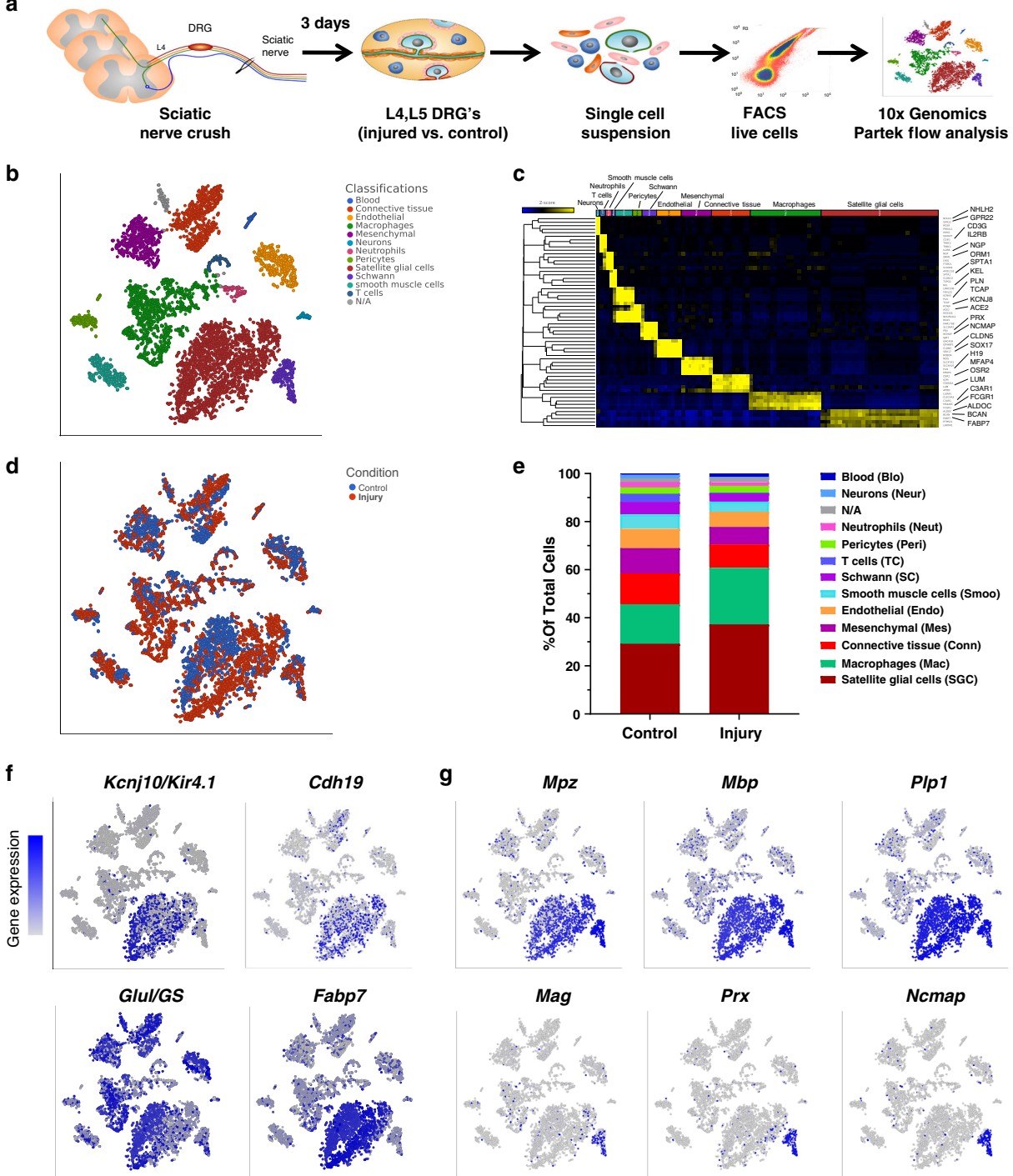

**Fig. 1 Characterization of cell populations in the DRG. a** Schematic of the experimental design for scRNAseq. **b** t-SNE plot of 6,541 cells from L4, L5 mouse dissociated DRG, with 13 distinct cell clusters (Unbiased, Graph based clustering). Classifications were assigned based on known marker genes. **c** Heatmap of the top five differentially expressed genes per cluster relative to all other clusters. **d** t-SNE plot of cells from naïve (blue) and injury (red) conditions. **e** Fraction of each cell type within control (2915 cells) and injury (3626 cells) conditions. *n* = 2 biologically independent experiments. Source data are provided as a Source Data file. **f** t-SNE overlaid for expression of marker genes for SGC. **g** t-SNE overlaid for expression of marker genes shared between Schwann cells and SGC (top panel) and marker genes for Schwann cell only (bottom panel).

of expression as a color gradient (Figs. 1f, g). The SGC cluster identity was confirmed by the expression of known SGC markers such as Kir4.1[8] and *Cdh19*[9] (Fig. 1f and Supplementary Fig. 1d). Whereas glutamine synthetase (Glul/GS) has been used as a SGC specific marker at the protein level[38,39], our analysis showed a nonspecific expression in almost all cells in the DRG (Fig. 1f and Supplementary Fig. 1d). Another marker used for SGC in rat is

SK3[39], but it was not detected inmouse SGC nor any other cells in the DRG. The most highly expressed gene in SGC, Fatty Acid Binding Protein 7 (*Fabp7*, also known as BLBP and BFABP) was enriched in SGC (Fig. 1f, Supplementary Fig. 1d and Supplementary Data 1). Whereas SGC and Schwann cells share the expression of known Schwann cell markers, such as *Mpz*, *Mbp*, and *Plp1* (Fig. 1g and Supplementary Fig. 1e), the Schwann cell

marker genes Myelin Associated Glycoprotein (*Mag*), Periaxin (*Prx*), and Non-Compact Myelin Associated Protein (*Ncmap*) genes were not expressed in SGC (Fig. 1f and Supplementary Fig. 1e). We next compared the top differentially expressed genes in SGC (605 genes) and Schwann cells (572 genes) (fold-change >4, significant differences across groups by ANOVA, and $p < 0.05$ compared to average gene expression in all other populations in the DRG), which revealed that SGC share only 2% of those gene transcripts with Schwann cells in the DRG (Supplementary Fig. 1f and Supplementary Data 2). A recent study highlighted the molecular differences between myelinating and nonmyelinating Schwann cells in the sciatic nerve[40]. Comparison of the SGC molecular profiles revealed higher similarity between SGC and myelinating Schwann cells over nonmyelinating Schwann cells (Supplementary Data 2). SGC also share several properties with astrocytes, including expression of Kir4.1 and glial fibrillary acidic protein (*Gfap*)[5,8,41]. We thus examined the transcriptional similarity between astrocytes and SGC by comparing the top differentially expressed genes in each cell type, using our scRNAseq data for SGC (605 genes, >4 fold change *p* value < 0.05 compared to all other populations in the DRG) and a previously published transcriptional analysis of astrocytes (500 genes, >6 fold change compared to other populations in the cerebral cortex)[42]. We found that SGC share about 10% of those gene transcripts with astrocytes, among them *Fabp7*, *Gfap*, *Pparα*, and *Aldoc* (Supplementary Fig. 1g, Supplementary Data 2). Our single cell RNAseq analysis using freshly dissociated tissue thus unravels the unique transcriptional profile of SGC in DRG and reveals that Schwann cells and SGC are transcriptionally distinct in the DRG.

***Fabp7* is a specific marker for adult mouse SGC.** The scRNAseq data revealed that one of the top differentially expressed genes in the SGC cluster is *Fabp7* (Fig. 1f and Supplementary Data 1). FABP7 is a nervous system specific member of the hydrophobic ligand binding protein family involved in uptake and transport of fatty acid[43]. FABP7 is involved in signal transduction and gene transcription in the developing mammalian CNS[44], but its precise function remains quite elusive. In the PNS, *Fabp7* is expressed during embryonic development and distinguishes glia from neural crest cells and neurons during the early stages of development[45,46]. These observations prompted us to further test if *Fabp7* can be used as a marker of SGC in adult mouse DRG. Consistent with our single cell data, we found that FABP7 labels SGC surrounding sensory neuron soma (Fig. 2a). The specificity of the FABP7 antibody was verified using DRG and sciatic nerve from the *Fabp7KO* mouse[47] (Supplementary Fig. 2a–e), in which FABP7 signal surrounding neurons is lost, but SGC are present and stained with GS[5,12]. Importantly, FABP7 does not label Schwann cells surrounding axons in the DRG (Fig. 2a, asterisks), or in the sciatic nerve (Supplementary Fig. 2d, e) consistent with previous reports[48,49] and our single cell analysis (Fig. 1f). These results suggest that FABP7 protein expression is highly enriched in SGC in the DRG.

We next tested if the *BLBPcre-ER* mouse line[50] can be used to label and manipulate SGC specifically. We crossed *BLBPCre-ER* to the *Rosa26-fs-TRAP* and observed that following a 10 days tamoxifen treatment, *BLBPcre-ER* mice drove expression of the GFP reporter in SGC (Fig. 2b, upper panel). To further ensure that the *BLBPcre-ER* is specifically expressed in SGC in the DRG and does not drive expression in Schwann cells in the nerve, we crossed *BLBPcre-ER* to the *Sun1-sfGFP-myc* (INTACT mice: *R26-CAG-LSL-Sun1-sfGFP-myc*)[51], which allows GFP expression in the nuclear membrane. We found GFP positive nuclei around the neurons in DRG sections (Fig. 2b, lower panel) and no nuclear GFP expression in naïve sciatic nerves (Fig. 2c, d). Since a role for

FABP7 in regulating Schwann cell-axon interactions has been proposed[48], with FABP7 beeing expressed 2 to 3 weeks after nerve injury when Schwann cell process formation is exuberant, we also examined nuclear GFP expression in injured nerves. We found that 3 days post injury, 4 and 7% of the nuclei expressed the GFP reporter ~0.5 mm proximal and distal to the injury site, respectively (Fig. 2c, d). This is consistent with a prior report showing that BLBP is not present 4–7 days post injury, but is expressed in some cells starting 14 days post injury[48]. Since typically 50% of nuclei express c-JUN, a marker for Schwann cells response to injury, two days after injury in the nerve[52], the GFP positive nuclei we observed may represent a very small subset of Schwann cells responding to injury. To further confirm the specificity of the *BLBPCre-ER* reporter following nerve injury, we re-analyzed a single cell data set from injured nerve (9 days post injury)[53]. We found that less than 5% of cells in the Schwann cell cluster expressed *Fabp7* (Supplementary Fig. 2f). A comparative analysis of the transcriptome of injured nerve (3 days post injury) also demonstrated very low counts of *Fabp7* in proximal and distal nerve segments[54]. Because *Fabp7* is also expressed at a low level in a subset of other cell types including macrophages, endothelial cells, and mesenchymal cells (Fig. 1f and Supplementary Fig. 1e), we cannot rule out the possibility that *BLBPCre-ER* may also label a subset of other cells in the DRG. Together, these experiments reveal that FABP7 represents a marker of SGC and that the *BLBPcre-ER* mouse line can be used to label and manipulate SGC, with only minimal impact on Schwann cells at early time points after nerve injury.

**SGC upregulate lipid metabolism in response to nerve injury.** To define the transcriptional response of SGC to nerve injury, cells in the SGC cluster were pooled, control, and injury conditions were compared to identify differentially expressed genes. 1255 genes were differentially upregulated in SGC after injury (FDR > 0.05, Log2Fold change >2) (Supplementary Data 3) and analyzed for enriched biological pathways using KEGG 2016 (Kyoto Encyclopedia of Genes and Genomes). This analysis revealed enrichment in lipid metabolic pathways, including fatty acid biosynthesis (Fig. 3a). Our findings also confirmed the upregulation of previously reported injury induced genes in SGC such as *Connexin43/Gja1*[14] and *Gfap*[13] (Supplementary Fig. 3g, Supplementary Data 3). 300 genes were differentially downregulated in SGC, with enrichment for cell cycle and p53 signaling pathways (Supplementary Data 3 and 4). Pathway analysis of upregulated genes in other major cell types in the DRG confirmed fatty acid metabolism as a unique pathway enriched in SGC in response to nerve injury (Supplementary Fig. 3a–f). This analysis also indicated that the cell cycle term was enriched in macrophages, and downregulated in SGC (Supplementary Fig. 3a, Supplementary Data 4), supporting the recent finding that macrophages but not SGC undergo cell cycle after nerve injury[38].

One of the genes enriched in SGC is fatty acid synthase (*Fasn*), which controls the committed step in endogenous fatty acid synthesis[27] (Fig. 3b and Supplementary Data 2). Although *Fasn* levels were not differentially regulated after injury in SGC, we decided to explore the role of *Fasn* further, because *Fasn* has the potential to impact major signaling and cellular pathways[55]. The FASN product palmitic acid is converted to other complex lipids that are critical for membrane structure, protein modification, and localization[55] and are also utilized for phospholipid synthesis. Immunostaining of DRG sections confirmed FASN expression in SGC surrounding sensory neuron soma (Fig. 3c, upper panel). To investigate the role of *Fasn* in SGC, we generated an SGC specific *Fasn KO* mouse (*FasncKO*) by crossing *BLBPcre-ER*[50] mice to mice carrying floxed *Fasn* alleles[56]. Eight week old *BLBPcre-ER*;

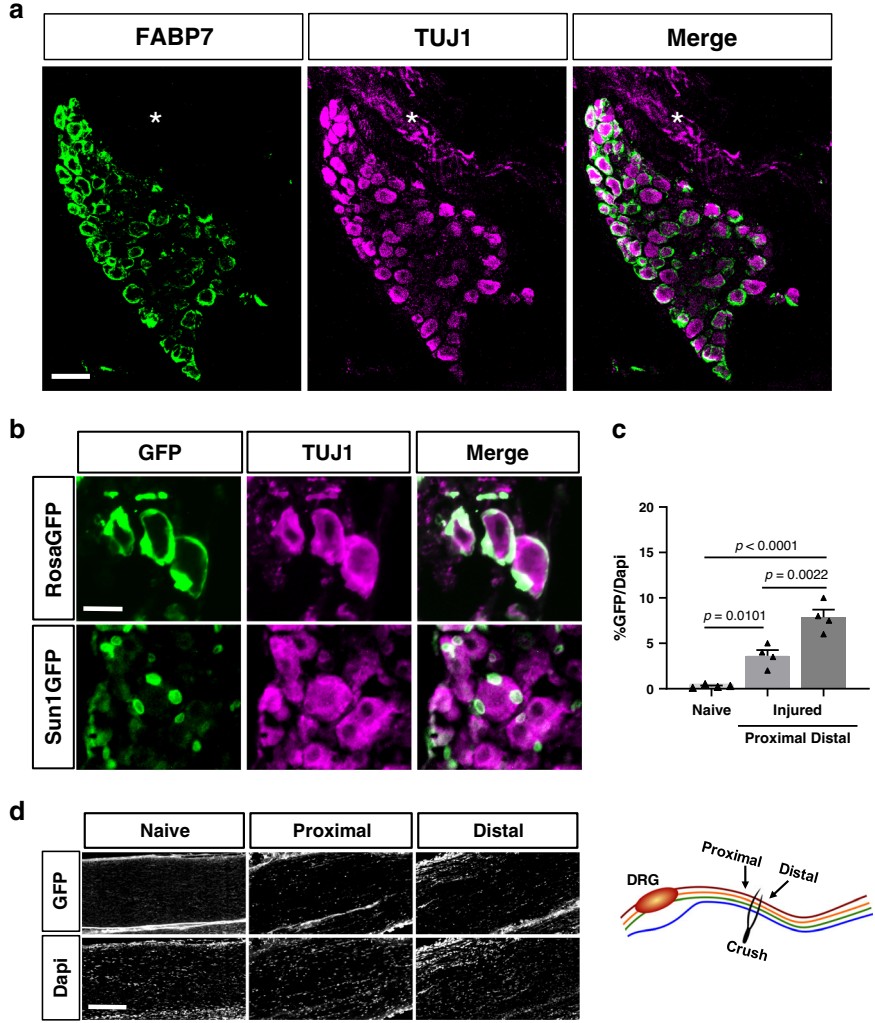

**Fig. 2 Fabp7 is a specific marker for SGC. a** Representative images of immunofluorescence staining of mouse DRG sections with FABP7 (green) which labels SGC surrounding neurons marked with TUJ1 (magenta). No FABP7 expression in the axon rich area (asterisks) is observed. $n = 3$ biologically independent animals, Scale bar: 100 μm. **b** *BLBPCre-ER* mice crossed with *RosaGFP* or *Sun1GFP* show expression of GFP in the SGC surrounding the neurons marked with TUJ1 (magenta). *RosaGFP* $n = 3$ and *Sun1GFP* $n = 4$ biologically independent animals, Scale bar: 50 μm. **c** Quantification of sciatic nerves from *BLBPCre-ER* mice crossed with *Sun1GFP*. GFP positive nuclei were normalized to the total number of DAPI positive nuclei. One-way analysis of variance (ANOVA) followed by Sidak's multiple comparisons test. Data are presented as mean values ± SEM. Source data are provided as a Source Data file. **d** Representative images from naive and injured nerves, ~0.5 mm proximal and distal to the injury site as shown in the scheme. $n = 4$ biologically independent animals. Scale bar: 50 μm.

*Fasn*^f/f (*FasncKO*) and *BLBPcre-ER*⁻;*Fasn*^f/+ mice (control) were fed with tamoxifen for 10 days to conditionally delete *Fasn* in SGC. This design allows us to ensure that both control and *FasncKO* groups are treated with tamoxifen and thus differences between groups do not result from the tamoxifen treatment itself. Expression of FASN in SGC visualized by immunofluorescence was significantly reduced in the *FasncKO* DRG compared to control littermates (Fig. 3c, bottom panel). Western blot analysis of FASN in DRG also confirmed a significant reduction of FASN expression in the *FasncKO* compared to control littermates (Fig. 3e). FASN has been shown to be expressed in Schwann cells[57], and immunostaining for FASN in the sciatic nerve support these results (Fig. 3d). However, FASN expression was not affected in the nerve of *FasncKO* mice (Fig. 3d, f) supporting the specificity of the *BLBPCre-ER* line for SGC. We also examined FASN protein expression in DRG and sciatic nerve by western blot in naive and injured conditions. Nerve injury increased the protein level of FASN in the DRG of *FasncKO* but not control mice (Fig. 3e). Nerve injury also increased FASN levels in the

nerve of both control and *FasncKO* (Fig. 3f) which could reflect an increase in FASN levels in Schwann cells after nerve injury[54,57]. Together, these results indicate that *Fasn* can be efficiently deleted from SGC in the DRG, with minimal impact in Schwann cells.

**Deletion of *Fasn* in SGC does not alter neuronal properties.** To determine the impact of *Fasn* deletion on SGC and neuron morphology, we performed transmission electron microscopy (TEM). SGC surrounding the neuron soma were pseudo colored (Fig. 4a), We observed an increase in the SGC nuclear area and a more circular, less elliptic nuclear morphology in the *FasncKO* animals compared to controls (Fig. 4a–c), whereas there was no change in the neuronal nucleus circularity (Fig. 4d). It has been reported that under certain conditions, such as alterations in lipid composition, the overall nuclear structure can be modified[58]. A study in yeast demonstrated that deletion of certain genes affecting lipid biosynthesis leads to nuclear expansion[58], suggesting that *Fasn* deletion in SGC may affect nuclear morphology.

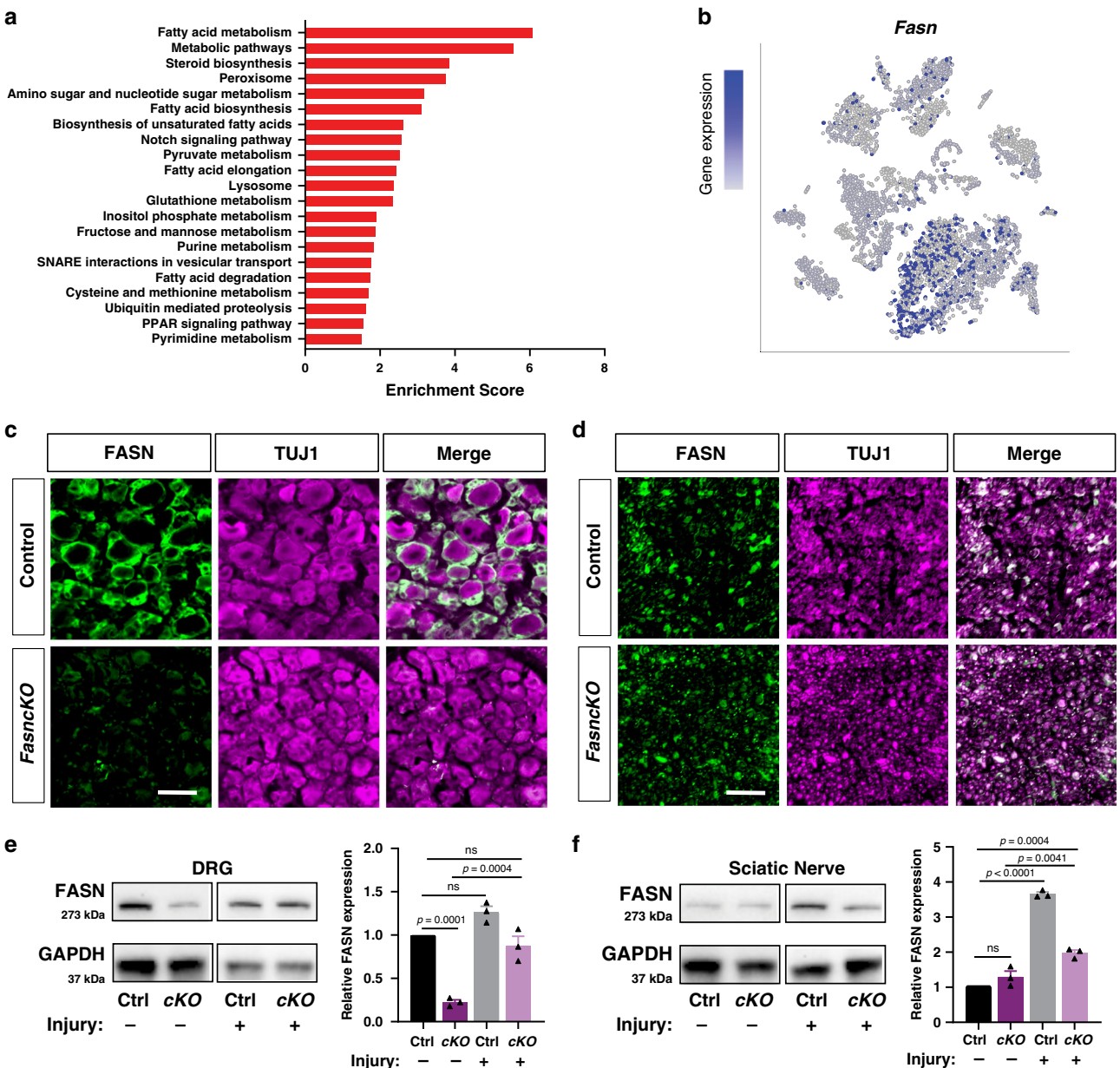

**Fig. 3 SGC upregulate genes involved in lipid metabolism in response to nerve injury. a** Pathway analysis (KEGG 2016) of differentially upregulated genes in the SGC cluster. *n* = 2 biologically independent experiments. (FDR < 0.05, Log2Fold change >2). **b** t-SNE overlay for expression of *Fasn* gene. **c** DRG sections from control and *FasncKO* mice, immunostained for FASN (green) and the neuronal marker TUJ1 (magenta). Scale bar: 50 μm. **d** Nerve sections from control and *FasncKO* mice, immunostained for FASN (green) and the neuronal marker TUJ1 (magenta). Scale bar: 50 μm. **e** Western blot analysis and quantification of FASN protein expression in DRG from control and *FasncKO* mice with and without injury. Quantification of FASN expression normalized to GAPDH expression. ns-non significant. Source data are provided as a Source Data file. **f** Western blot analysis and quantification of FASN protein expression in Sciatic nerve from control and *FasncKO* mice with and without injury. Quantification of FASN expression normalized to GAPDH expression. ns-non significant. Source data are provided as a Source Data file. *n* = 3 biologically independent animals in (**e**, **f**). Data are presented as mean values ± SEM in (**e**, **f**). One way ANOVA followed by Dunnett's multiple comparisons test in (**e**, **f**).

However, in both control and *FasncKO*, the SGC sheath is smooth and separated from the sheaths enclosing the adjacent cell bodies by a connective tissue space (Fig. 4a), as described extensively previously[6,59]. The SGC coat is also in direct contact with the neuronal membrane (Fig. 4a), as previously described[6,59]. To ensure that *Fasn* deletion in SGC does not impact nerve morphology, we next evaluated both Remak bundle structure and myelin sheath thickness in the nerve of the *FasncKO* mice compared to control animals (Fig. 4e–i). Neither the number of axons per Remak bundle nor the axon diameter and myelin thickness, measured as the ratio between the inner

and outer diameter of the myelin sheath (g-ratio), was altered in the *FasncKO*. The overall nerve structure examined by TUJ1 staining was also not altered in the *FasncKO* nerves compared to controls in naïve conditions or following injury (Supplementary Fig. 4a).

To ensure that the reduction of *Fasn* in SGC does not cause neuronal cell death, we immunostained sections of DRG from *FasncKO* and control mice for the apoptotic marker cleaved caspase 3 before and after injury (Supplementary Fig. 4b). Quantification of Caspase 3 intensity revealed no change in apoptotic cell death in naïve conditions and three days post injury

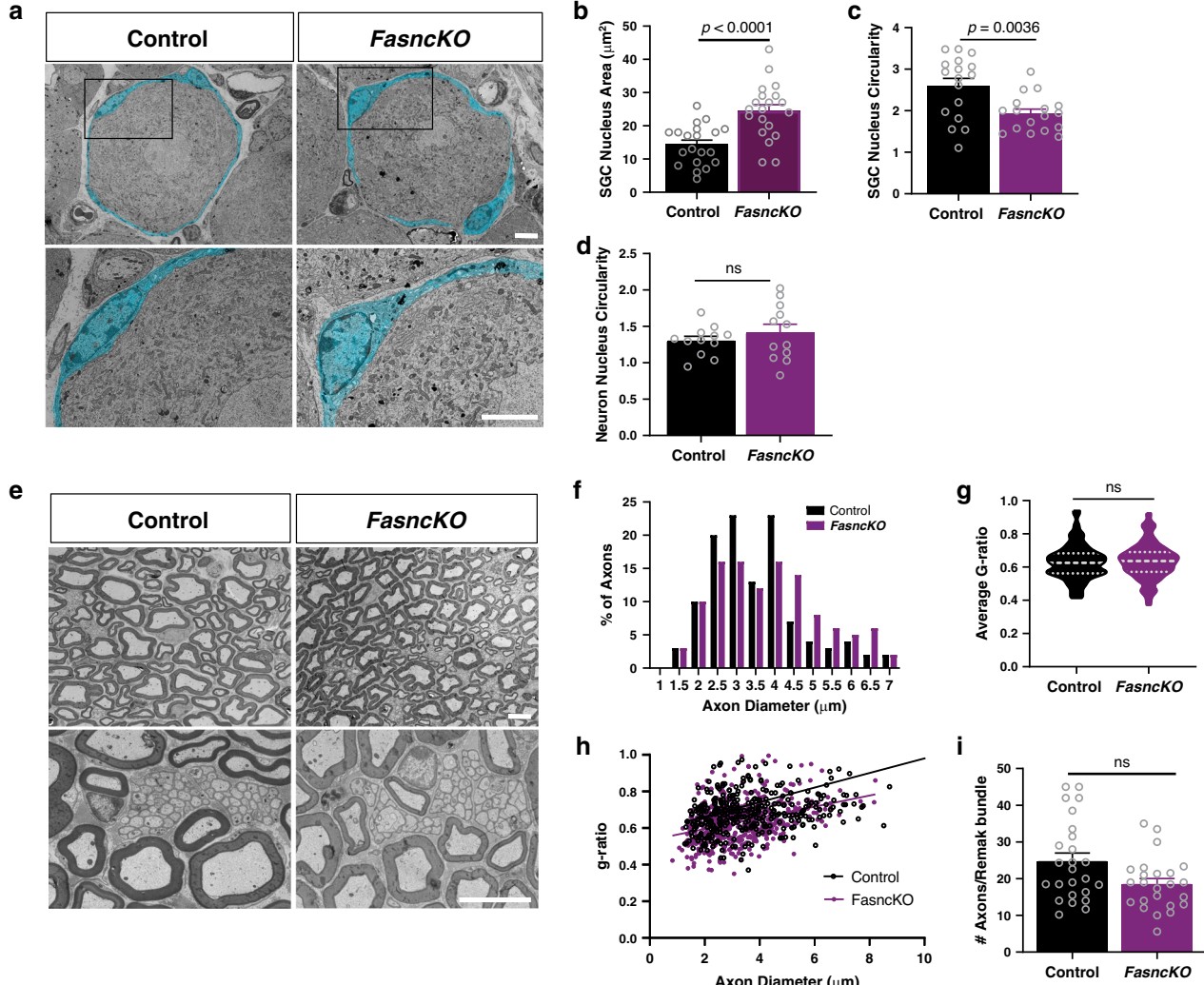

**Fig. 4 _Fasn_ deletion in SGC does not alter neuronal morphology. a** Representative TEM images of a DRG cell body and its enveloping SGC sheath, pseudo-colored in turquoise, from control and _FasncKO_ mice. High magnification of boxed area from top panels are showed in bottom panels. Scale bar: 5 μm. **b** Average SGC nuclear area. $n = 20$ cells examined over 3 independent experiments. **c** SGC nuclear circularity (ratio between major axis(X) and minor axis(Y). 1 = circular, <1=elliptic). $n = 17$ cells examined over 3 independent experiments. **d** Neuron nuclear circularity (ratio between major axis(X) and minor axis(Y). 1 = circular, 1<elliptic). $n = 12$ cells examined over 3 independent experiments. ns-non significant. **e** Representative TEM images of sciatic nerve cross sections from control and _FasncKO_ mice. Scale bar: 5 μm. **f** Quantification of axon diameter distribution in sciatic nerves of _FasncKO_ and control mice (average axon diameter Control = 3.574 μm _FasncKO_ = 3.353 μm $n = 130$ axons). **g** Quantification of g-ratio in sciatic nerves of _FasncKO_ and control mice. $n = 110$ cells in control and $n = 116$ cells in _FasncKO_ examined over 3 independent experiments. Mean control = 0.6286, mean _FasncKO_ = 0.6359. ns-non significant. Plots lines at the median (dashed) and quartiles (dotted). **h** Linear correlation of g-ratio versus axon diameter in sciatic nerves of _FasncKO_ mice compared with controls. non-significant. **i** Quantification of the number of axons per Remak bundle in _FasncKO_ and control nerves. $n = 3$ biologically independent animals from 2 independent experiments in **a**–**i**. Data are presented as mean values ± SEM in (**b**, **c**, **d**, **I**). _P_ values determined by two-tailed _t_ test in (**b**, **c**, **d**, **g**, **i**). Source data are provided as a Source Data file for (**b**, **c**, **d**, **f**, **g**, **h**, **i**).

between _FasncKO_ and control (Fig. 5a and Supplementary Fig. 4b). These experiments do not exclude the possibility that other forms of cell death such as necrosis may occur.

We next examined the expression of _Atf3_, an established intrinsic neuronal injury marker[1,60]. _Atf3_ levels in control and _FasncKO_ DRG were very low in the absence of injury (Fig. 5b), indicating that in naïve condition, _Fasn_ deletion in SGC did not cause a stress response in DRG neurons. Nerve injury increased the mRNA levels of _Atf3_ in control and _FasncKO_ DRG, but this response was less robust in the _FasncKO_ (Fig. 5b). Together, these results indicate that loss of _Fasn_ in SGC does not lead to neuronal morphological deficits in the DRG and does not elicit a stress response in neurons.

The bidirectional communication between neurons and SGC participates in neuronal excitability and the processing of pain

signals[24]. The excitability of sensory neurons is controlled in part by the surrounding SGC[17,61,62]. To examine whether _Fasn_ deletion in SGC affected functional properties of DRG neurons, we compared intrinsic excitability and firing properties of DRG neurons from _FasncKO_ and control mice. Whole-cell recordings were performed in short-term cultures of DRG neurons and glia. Medium diameter neurons that were associated with at least one glial cell were targeted for recordings (Supplementary Fig. 4c). A subset of recorded cells was filled with biocytin via the patch pipette for _post hoc_ verification of neuronal identity. The vast majority of filled cells were identified as IB4-positive nociceptors (Supplementary Fig. 4d). We observed that all major features of intrinsic neuronal excitability were unaffected in DRG neurons from _FasncKO_ mice compared to controls, including the resting membrane potential (Fig. 5c), membrane capacitance (Fig. 5d),

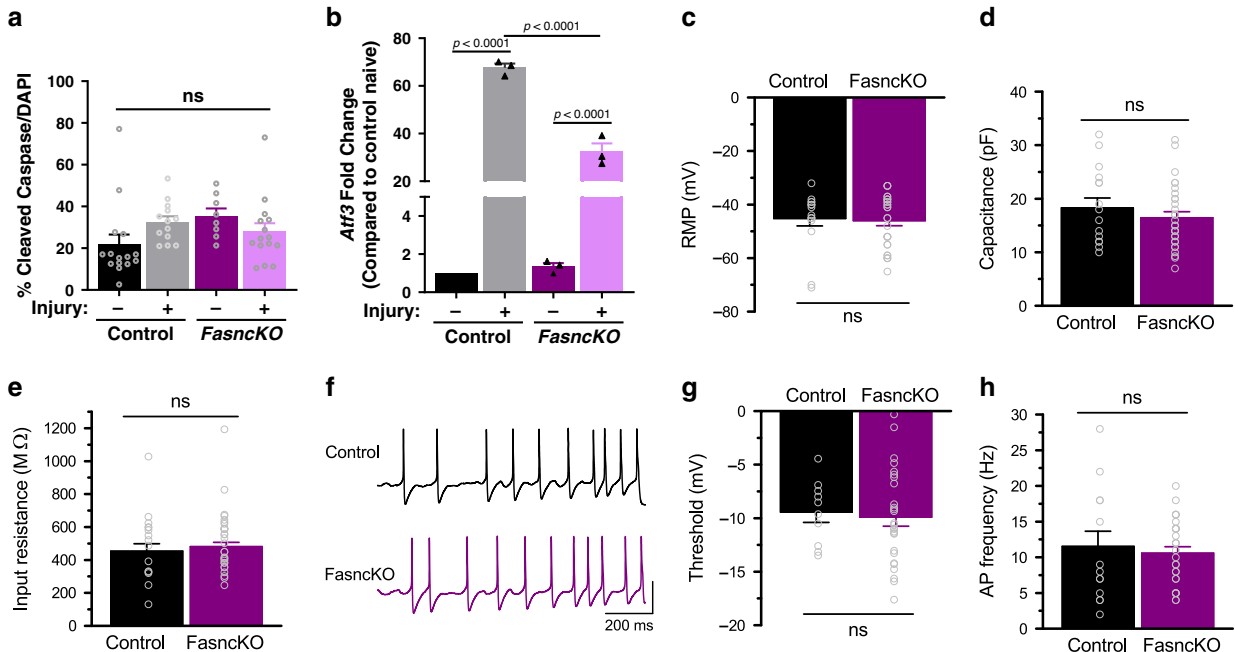

**Fig. 5 *Fasn* deletion in SGC does not alter neuronal functional properties. a** Quantification of cleaved caspase 3 in DRG sections from control and *FasncKO* mice in naïve and 3 days after sciatic nerve injury. The ratio of cleaved caspase 3 positive vs. DAPI nuclei was measured. $n = 4$ biologically independent animals. ns-non significant. **b** qPCR analysis of *Atf3* expression in DRG from control and *FasncKO* mice in naïve and 3 days after sciatic nerve injury. $n = 3$ biologically independent animals. **c** Whole-cell recordings were performed from cultured DRG neurons associated with at least one glial cell, from *FasncKO* or control mice. Resting membrane potential (RMP) in control $-45.3 \pm 2.7$ mV, $n = 16$ cells from 8 independent experiments; in *FasncKO* $-46.1 \pm 1.8$ mV, $n = 30$; $p = 0.79$. **d** Membrane capacitance in control $18.3 \pm 1.8$ pF, $n = 16$ cells from 8 independent experiments; in *FasncKO* $16.5 \pm 1.1$ pF, $n = 30$ cells from 8 independent experiments; $p = 0.36$. **e** Input resistance in control $456 \pm 43$ MΩ, $n = 16$ cells from 8 independent experiments; in *FasncKO* $482 \pm 24$ MΩ, $n = 30$ cells from 8 independent experiments; $p = 0.56$. **f** Spiking properties of DRG neurons from controls and *FasncKO*. **g** Action Potential threshold in control $-9.46 \pm 0.94$ mV, $n = 13$ cells from 8 independent experiments; in *FasncKO* $-9.91 \pm 0.84$ mV, $n = 28$ cells from 8 independent experiments; $p = 0.77$ ns- non-significant. **h** Neuronal firing rate in control $11.57 \pm 2.09$ Hz, $n = 14$ cells from 8 independent experiments; in *FasncKO* $10.62 \pm 0.86$ Hz, $n = 29$ cells from 8 independent experiments; $p = 0.62$. ns- non significant. *P* values determined by One way ANOVA followed by Sidak's multiple comparisons test in (**a**, **b**) and two-tailed *t* test in (**c**, **d**, **e**, **g**, **h**). Data are presented as mean values ± SEM. Source data are provided as a Source Data file for (**a–e**, **g**, **h**).

and input resistance (Fig. 5e). We further examined the spiking properties of DRG neurons and found no detectable changes in action potential (AP) threshold (Fig. 5g), which represents the principle determinant of neuronal excitability, or the neuronal firing frequency (Fig. 5f, h) in *FasncKO* mice compared to controls. Therefore, *Fasn* deletion in SGC does not affect functional properties of DRG neurons in the naïve, dissociated conditions.

**Deletion of *Fasn* in SGC impairs axon regeneration.** We observed that *Atf3* expression after injury was reduced in *FasncKO* compared to control (Fig. 5b), suggesting that absence of *Fasn* in SGC may impact the neuronal response to injury. To test the consequence of *Fasn* deletion in SGC on axon regeneration, we used our established in vivo and ex-vivo regeneration assays[63–65]. Two weeks after tamoxifen treatment was completed, we performed a sciatic nerve crush injury in *FasncKO* and control mice and measured the extent of axon regeneration past the injury site 3 days later by labeling nerve sections with SCG10, a marker for regenerating axons[66]. The crush site was determined according to highest SCG10 intensity along the nerve. We used two measurements to quantify axon regeneration. First, we measured the length of the 10 longest axons, which reflect the extent of axon elongation, regardless of the number of axon that regenerate. Second, we measured a regeneration index by normalizing the average SCG10 intensity at distances away from the crush site to the SCG10 intensity at the crush site. This measure takes into account both the length and the number of

regenerating axons past the crush site. Loss of *Fasn* in SGC impaired axon regeneration, demonstrated by reduced axonal length and lower regeneration capacity (Fig. 6a–c). We also tested if loss of *Fasn* has an effect on the conditioning injury paradigm, in which a prior nerve injury increases the growth capacity of neurons[67]. Isolated DRG neurons from naïve and injured *FasncKO* and control mice were cultured for 20 h (Fig. 6d). In naïve animals, typically only few neurons extend short neurites, whereas a prior injury leads to more neurons initiating neurite growth and longer neurites (Fig. 6d–f). In *FasncKO* mice, naïve neurons presented similar neurite length to control naïve animals with a similar number of neurons initiating neurite growth. In contrast, a prior injury in *FasncKO* mice only partially conditioned the neurons for growth. Neurons in injured *FasncKO* displayed reduced neurite length compared to injured controls, but a similar number of neurons initiating neurite growth (Fig. 6d–f). These results indicate that *Fasn* expression in SGC contributes to the conditioning effect and the elongating phase of axon growth.

**Activation of PPARα in SGC contributes to axon regeneration.** To understand the mechanism by which *Fasn* in SGC regulates axon regeneration, we examined the role of PPARs, which represent a unique set of lipid regulated transcription factors[25]. FASN synthesizes palmitic acid, which is the substrate for the synthesis of more complex lipid species[27], including phospholipids. Importantly, it is not palmitic acid per se that is required for PPAR activation, but phospholipids[28,29]. Our scRNAseq analysis revealed that *Pparα*, but not *Pparγ*, was enriched in the SGC

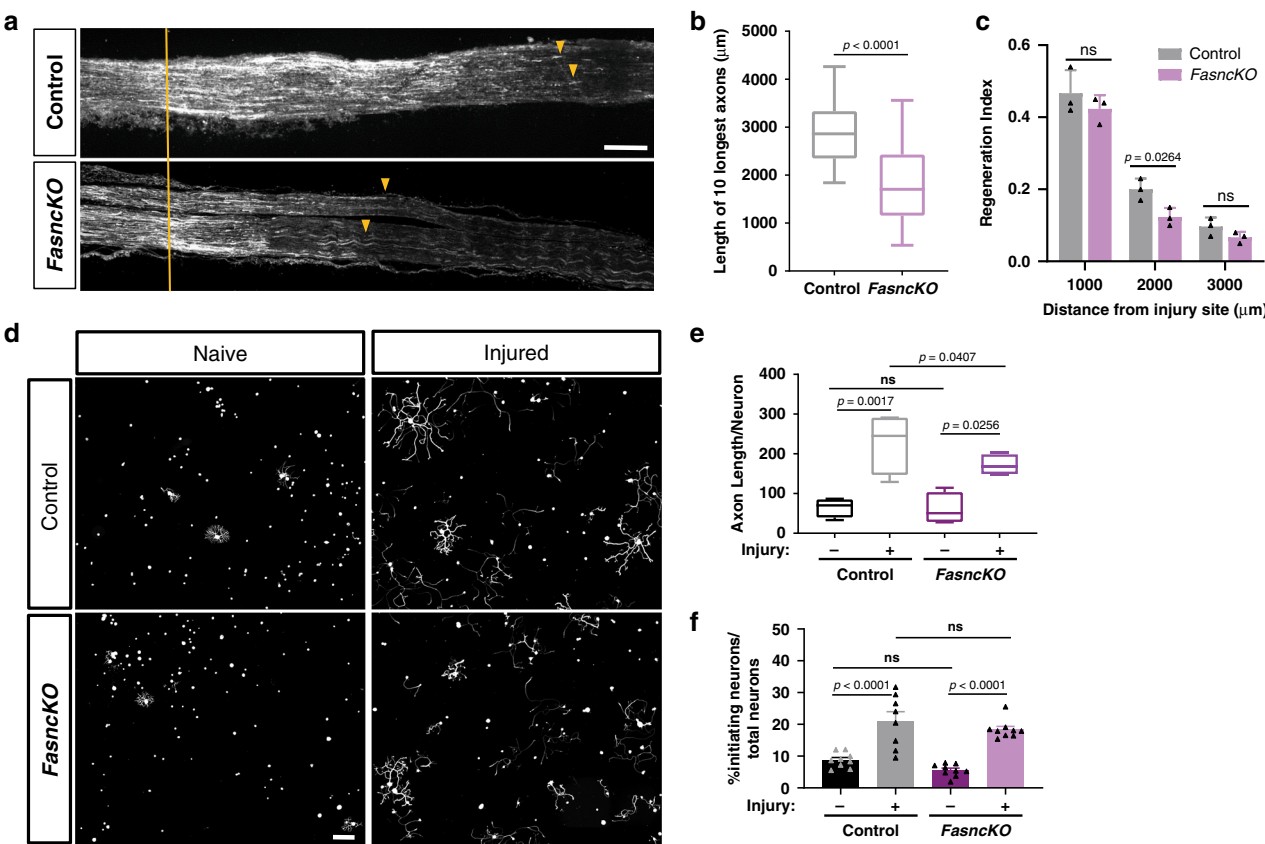

**Fig. 6 *Fasn* deletion in SGC impairs axon regeneration. a** Representative longitudinal sections of sciatic nerve from control and *FasncKO* mice 3 days after sciatic nerve injury, stained for SCG10. Orange lines indicate the crush site, identified as the maximal SCG10 intensity. Arrowheads indicate longest regenerating axons. Scale bar: 500 μm. **b** Length of the longest 10 axons was measured in 10 sections for each nerve. **c** Regeneration index was measured as SGC10 intensity normalized to the crush site. **d** Representative images of dissociated DRG neurons from control and *FasncKO*, cultured for 20 h, from naïve and injured (3 days post conditioning sciatic nerve injury) and stained with the neuronal marker TUJ1. Scale bar: 200 μm. **e** The length of axons per neuron was measured. Automated neurite tracing analysis using Nikon Elements of about 500 neurons per replicate. **f** Percentage of initiating neurons normalized to the total number of neurons was measured in 8 independent animals, average of 500 neurons per replicate. Automated neurite tracing analysis using Nikon Elements. One way ANOVA. Sidak's multiple comparisons test ns-non-significant. $n = 8$ biologically independent animals examined over 2 independent experiments for (**a–f**). *P* values determined by two tailed *t* test presented in (**b**) and One way ANOVA followed by Sidak's multiple comparisons test in (**c**, **e** and **f**). Data are presented as mean values ± SEM. The boxplots in **b** and **e** extends from the 25th to 75th percentiles and the whiskers go down to the smallest value and up to the largest where the box middle line is the median. Source data are provided as a Source Data file for (**b**, **c**, **e**, **f**).

cluster, along with known PPARα target genes *Ppargc1α*, *Fads2* and *Pex11a*[68] (Fig. 7a). These PPARα target genes were also upregulated after injury in the SGC cluster (Fig. 7b and Supplementary Data 3, 4). PPARα regulates the expression of genes involved in lipid and carbohydrate metabolism[25]. This is consistent with our GO analysis of upregulated genes after injury that indicated an enrichment of the PPAR signaling pathway and lipid metabolism in SGC after injury (Fig. 3a and Supplementary Data 4). We further confirmed the enrichment of PPARα target genes in SGC vs. neurons using an RNAseq data set from purified nociceptors in naïve condition and 3 days post injury that we previously generated[69] (Fig. 7b). Immunostaining of DRG sections in naïve and injured conditions revealed that PPARα expression is enriched in SGC, but not in neurons (Fig. 7c and Supplementary Fig. 6a). Furthermore, a recent study that examined the transcriptional profile of neurons at single cell resolution in naïve and injured conditions revealed that *Pparα* and PPARα target genes are not expressed in neurons[70]. These observations indicate that PPARα signaling is enriched in SGC and suggest that fatty acid synthesis in SGC regulates PPARα-dependent transcription.

Whereas biologic PPARα agonists consist of a broad spectrum of ligands[68], synthetic PPARα agonists include clofibrate,

fenofibrate, bezafibrate, gemfibrozil, Wy14643, and GW7647. To test the effect of these PPARα agonists on axon regeneration, we modified our spot culture assay, in which embryonic DRG are dissociated and cultured in a spot, allowing axons to extend radially from the spot[63,65]. This assay recapitulates for the most part what can be observed in vivo in the nerve[63–65], and is thus suitable to test compounds affecting axon regeneration. By not including the mitotic inhibitor 5-deoxyfluoruridine (FDU) to eliminate dividing cells, we observed SGC, labeled with FABP7 in the cell soma area, where they surround neurons by DIV7 (Fig. 7d, Supplementary Fig. 5a, b) but not in the axon region (Fig. 7d, Supplementary Fig. 5a, b). The PPARα agonists clofibrate, fenofibrate, Wy14643 and GW7647 were added to the media at 2 concentrations (10 μM and 100 nM) at DIV6. Axons were cut using a microtome blade at DIV7 and allowed to regenerate for 24 h, fixed and stained for SCG10 to visualize axon regeneration[66]. Regenerative length was measured from the visible blade mark to the end of the regenerating axon tips (Fig. 7e, f). Whereas Wy14643 and GW7647 had no effect on axonal regeneration compared to DMSO, clofibrate (100 nM) and fenofibrate (100 nM, 10 μM) increased axon growth after axotomy (Fig. 7f). However, clofibrate at a higher concentration (10 μM) caused cell death and axonal degeneration. To ensure

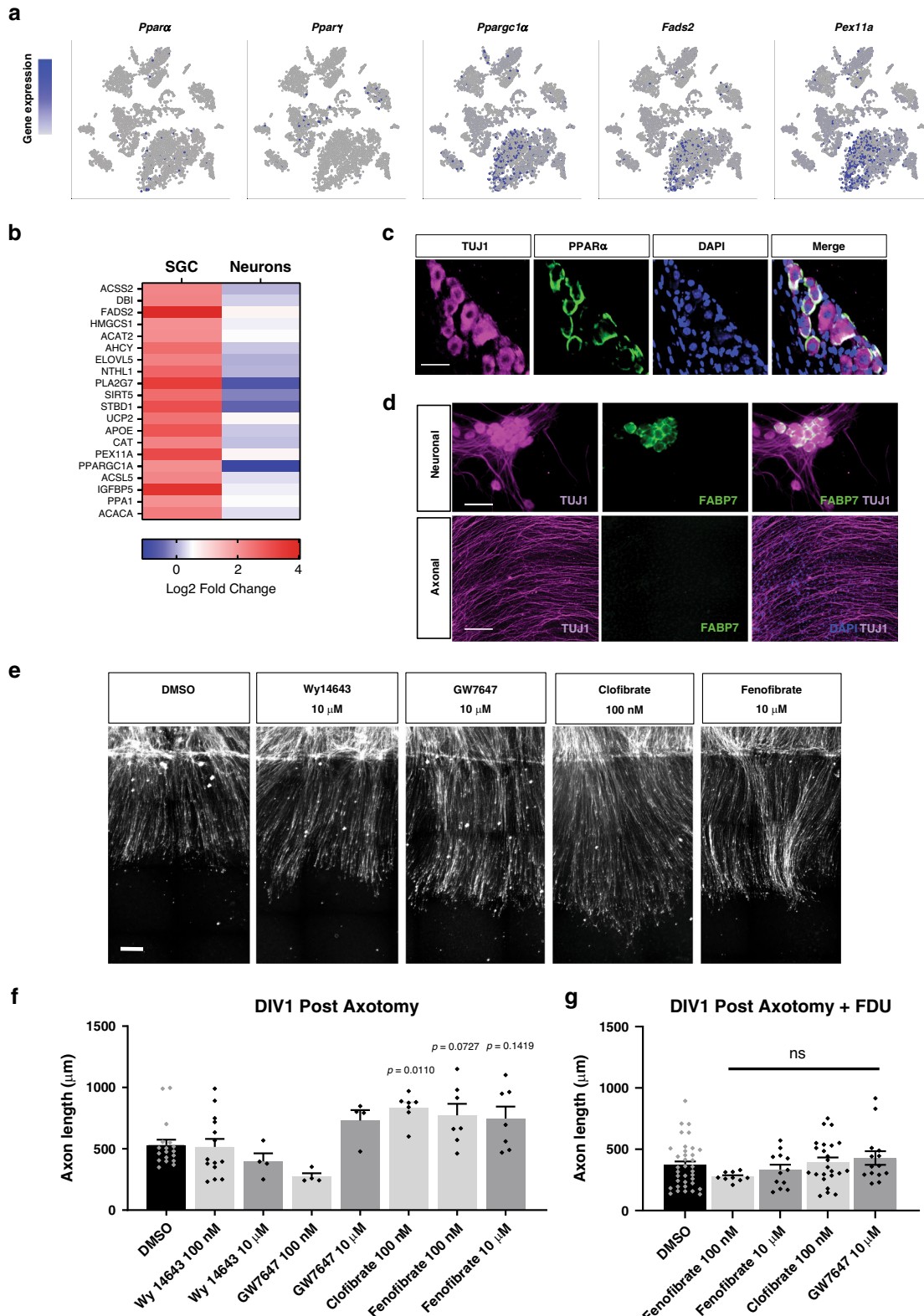

that the increased regeneration effect was mediated by SGC and was not due to PPARα activation in neurons, we tested fenofibrate, clofibrate, and GW7647 in spot cultures treated with FDU, in which no FABP7 positive cells was detected at DIV7 (Supplementary Fig. 5b). In these conditions, we observed no increase in axon regeneration (Fig. 7g and Supplementary Fig. 5c). These results indicate that activation of PPARα signaling in SGC promote axon growth, although we cannot fully exclude that

other cell types present in the spot culture contribute to this effect.

**Fenofibrate treatment rescues the absence of *Fasn* in SGC**. In the absence of *Fasn* in SGC, PPARα may lack its endogenous agonist and PPARα signaling following nerve injury in SGC may be compromised. To test this hypothesis, we determined if fenofibrate can rescue the axon regeneration defects that we

**Fig. 7 Activation of PPARα in SGC promotes axon regeneration in neuron-SGC cocultures. a** t-SNE overlay for expression of the genes *Pparα*, *Pparγ*, and *Pparα* target genes *Ppargc1a*, *Fads2*, and *Pex11a*. **b** Heat map of PPARα target genes expression after nerve injury in SGC and neurons. **c** Representative images of DRG sections immunostained for PPARα (green) and TUJ1 (magenta). *n* = 3 biologically independent animals. Scale bar: 100 μm. **d** Neuronal and axonal area in embryonic DRG spot co-cultures at DIV7, immunostained for FABP7 (green) and TUJ1 (magenta). *n* = 3 biologically independent experiments. Scale Bar: 100 μm (neuronal) and 50 μm (axonal). **e** Embryonic DRG spot co-cultures axotomized at DIV7 after a 24 h pre-treatment with the indicated PPARα agonists at the indicated concentration. Cultures were fixed after 24 h and stained with SCG10. Scale Bar: 50 μm. **f** Distance of regenerating axons was measured from the injury site from images shown in (**e**). **g** Embryonic DRG spot culture was supplement with FDU (5-deoxyfluoruridine) to eliminate dividing cells in the culture. Axotomy was performed at DIV7 after a 24 h pre-treatment with the indicated PPARα agonists. Distance of regenerating axons was measured from injury site after 24 h. ns-non significant. *n* = 4 biologically independent animals examined over 2 independent experiments for (**f** and **g**). *P* values determined by One way ANOVA followed by Dunnett's multiple comparisons test are presented in (**f** and **g**). Data are presented as mean values ± SEM. Source data are provided as a Source Data file for (**f** and **g**).

observed in the *FasncKO* mice. We chose fenofibrate for these in vivo experiments because it is FDA-approved to treat dyslipidemia[71], is a specific PPARα agonist with minimal activity toward PPARγ and can be delivered easily in the diet[72]. Control and *FasncKO* mice were fed with chow supplemented with fenofibrate, as described[72] for 2 weeks prior to nerve injury. *Pparα* as well as PPARα target genes were upregulated in the DRG of fenofibrate fed mice compared to mice fed with a normal diet (Fig. 8a). Immunostaining of DRG sections showed that fenofibrate treatment did not lead to PPARα expression in neurons (Supplementary Fig. 6a), indicating that the upregulation of PPARα target genes observed in whole DRG is occurring in SGC and not in neurons. The reduced *Atf3* expression in the DRG after nerve injury in the *FasncKO* compared to control was rescued in the *FasncKO* mice fed with fenofibrate (Fig. 8b). Since ATF3 upregulation in response to injury occurs mainly in neurons (Supplementary Fig. 6b), this result suggest that fenofibrate can compensate for the lack of *Fasn* in SGC and can rescue the neuronal response to injury. Another proregenerative neuronal gene, *Gap43*, showed a similar trend as *Atf3*. *Gap43* reduced expression in response to injury in the *FasncKO* was rescued with the fenofibrate diet (Supplementary Fig. 6c). In the *FasncKO*, we also noted an increase in *Gap43* in naïve conditions with the fenofibrate diet compared to normal diet (Supplementary Fig. 6c). This is consistent with previous reports that lipoproteins secreted from glial cells upregulate mRNA expression of *Gap43* in hippocampal neurons[33]. In contrast, *Jun*, which is also considered to be a proregenerative gene, was not affected in the *FasncKO* or by fenofibrate treatment (Supplementary Fig. 6d). Because we assessed *Atf3*, *Gap43*, and *Jun* expression in whole DRG, we cannot exclude the possibility that other cell types regulate the expression of these genes in response to injury and fenofibrate treatment. We found that the impaired axon regeneration in *FasncKO* mice was rescued by the fenofibrate diet (Fig. 8c–e). The fenofibrate diet also rescued the impaired axon growth in the conditioning paradigm (Fig. 8f, g), with no change in the percent of neurons initiating growth (Fig. 8h). The regeneration of peripheral sensory neurons after injury is a relatively efficient process and we did not observe further improvement in both in vivo and in vitro experiment in control animals treated with fenofibrate (Fig. 8c–h). These results suggest that by activating PPARα, fenofibrate rescues the absence of *Fasn* in SGC. We cannot exclude the possibility that fenofibrate may also operate via other mechanisms in other cell types. However, these results are consistent with the notion that in response to axon injury, FASN expression in SGC activates PPARα signaling to promote axon regeneration in adult peripheral nerves.

## Discussion

A role for SGC in nerve regeneration has not been considered and the biology of SGC is very poorly characterized under normal or pathological conditions. Our unbiased single cell approach characterized the molecular profile of SGC and demonstrated that SGC play a previously unrecognized role in peripheral nerve regeneration, in part via injury induced activation of PPARα signaling. Our results highlights that the neuron and its surrounding glial coat form a functional unit that orchestrates nerve repair.

SGC have been identified mostly based on their morphology and location. Several SGC markers have been characterized, but no useful markers that can be used to purify SGC and understand their biology at the molecular level have been identified so far. Although a previous study reported that SGC are very similar to Schwann cells[9], this conclusion was drawn from the analysis of cultured and passaged SGC and Schwann cells. Other SGC isolation methods rely on immunoreactivity to SK3[39] or on multiple sucrose gradients and cell plating[9]. One recent paper used immunisolation based on GS expression in SGC to examine their transicptonal profile and response to injury[38]. Whereas at the immunostaining level, GS appear specifc to SGC, it does not appear to label all SGC[38]. At the gene expression level, our data reveal that GS is expressed by many cell types in the DRG. Therefore immunoisolation based on GS expression might represent a subset of SGC and might not entirely be speciifc to SGC[38]. Our single cell approach is unbiased to any predetermined markers and provides the advantage of minimal time between dissociation and analysis to capture more accurately the transcription status of the cells. Using this method, we demonstrate that SGC represent a unique cell population in the DRG, which share some molecular markers with Schwann cells as well as astrocytes.

We also characterized *Fabp7* as a marker for adult SGC and a mouse line, *BLBPcre-ER*[50], that can be used to manipulate SGC specifically in the DRG. Because *Fabp7* is expressed in radial glial cells as well as neuronal progenitors and is critical for neurogenesis in the CNS[44,73] this mouse line needs to be used in an inducible manner in adult mice to avoid targeting other cells during development. It is also important to note that *Fabp7* is expressed in astrocytes, where it is important for dendritic growth and neuronal synapse formation[74] as well as for astrocyte proliferation during reactive gliosis[75], and thus the *BLBPcre-ER* line will also target astrocytes in the CNS. SGC are believed to share common features with astrocytes, such as expression of Kir4.1 and functional coupling by gap junctions[5,11,13], with SGC surrounding the same neuron connected by gap junctions. The *BLBPcre-ER* mouse line thus represents a useful tool to study SGC in the PNS.

Injury related changes in SGC have been studied in pathological conditions such as inflammation, chemotherapy-induced neuropathic pain, and nerve injuries[5,18,19,76]. Nerve injury was shown to increases SGC communication via gap junctions[76], leading to increased neuronal excitability[19]. Nerve lesions also induce an increase in GFAP expression[13,18,21]. In our single cell data, we also found increased expression of connexin43/*Gja1* and *Gfap* after nerve injury. Whereas prior studies suggested

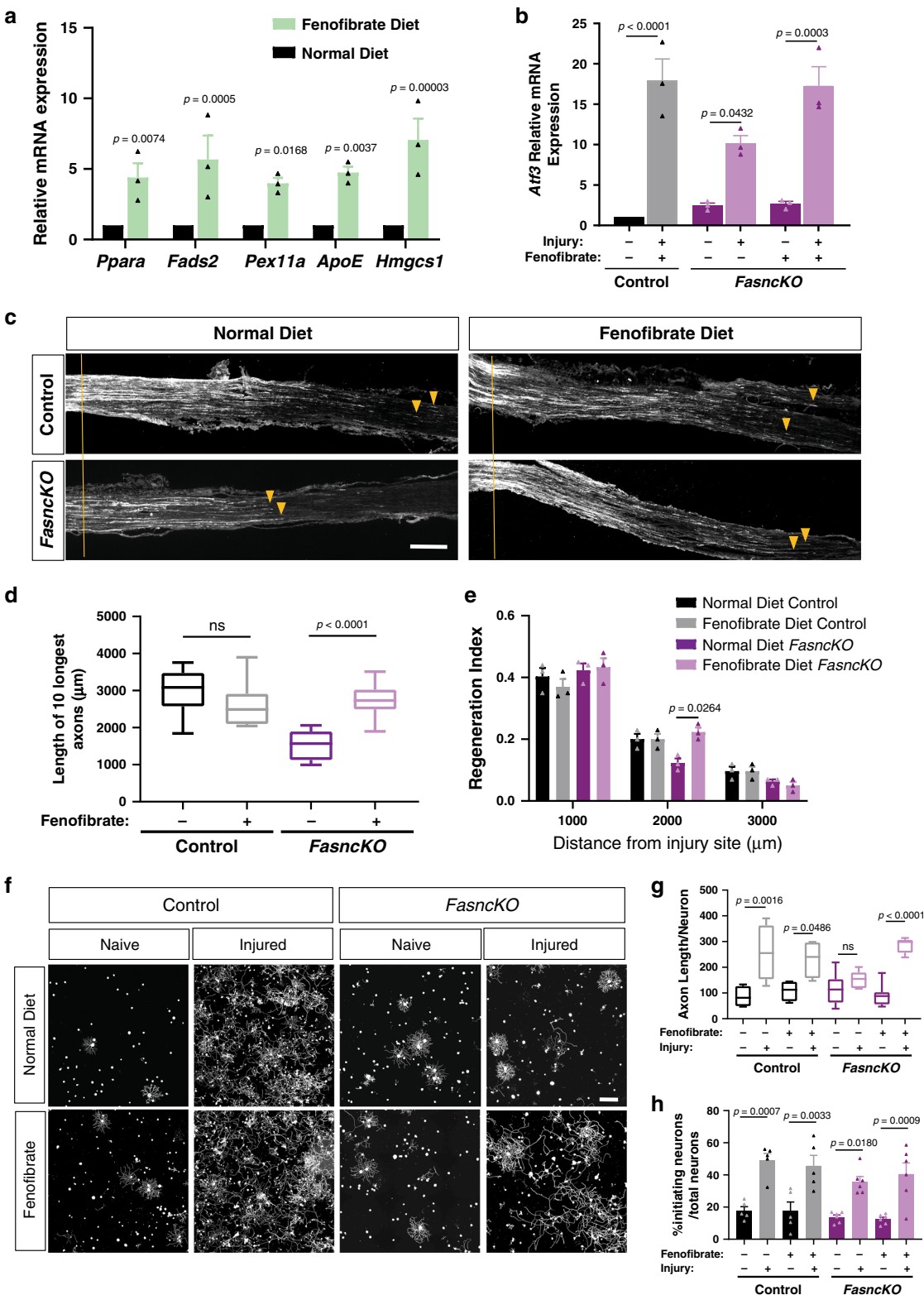

SGC proliferation after injury[5,11,13], a recent study demonstrated that cell proliferation occurred in macrophages but not in SGC after nerve injury[38]. Our results are in agreement with this latter study, as we did observe enrichment for cell cycle markers in macrophages, and these cell cycle markers were downregulated in SGC 3 days after nerve injury. Prior studies relied largely on morphological location of cell cycle markers relative to neurons[11,13,15], and it is thus possible that

macrophages infiltrating between neuron and SGC[77] were mis-identified as proliferating SGC.

Our analysis revealed that SGC activate PPARα signaling to promote axon regeneration in adult peripheral nerves. PPARs are ligand-activated nuclear receptors with the unique ability to bind lipid signaling molecules and transduce the appropriate signals derived from the metabolic environment to control gene expression[25]. Accordingly, PPARs are key regulators of lipid and

**Fig. 8 Activation of PPARα in SGC promotes axon regeneration. a** qPCR analysis of PPARα target genes expression in DRG from mice fed with a normal diet vs. fenofibrate diet. **b** qPCR analysis of *Atf3* expression in DRG from control and *FasncKO* mice in naïve and 3 days after sciatic nerve injury with and without fenofibrate. **c** Representative longitudinal sections of sciatic nerve from *FasncKO* mice fed with a normal diet or fenofibrate diet, stained for SCG10. Orange lines indicate the crush site, arrow heads indicate longest axons Scale Bar: 500 μm. **d** Length of the longest 10 axons was measured in 10 sections for each nerve. ns-non significant. **e** Regeneration index was measured as SGC10 intensity normalized to the crush site. **f** Representative images of dissociated DRG neurons from control and *FasncKO* mice fed with a normal diet or fenofibrate diet, cultured for 20 h and stained with TUJ1. Scale bar: 200 μm. **g** Length of axons per neuron was measured with average of 500 neurons per replicate. ns-non significant. **h** Percentage of initiating neurons out of total number of neurons was measured. Average of 500 neurons per replicate. Automated neurite tracing analysis using Nikon Elements. $n = 3$ biologically independent animals in (**a** and **b**) and $n = 8$ in (**c–e**). $n = 8$ biologically independent animals examined over 2 independent experiments in (**f–h**). $P$ values determined by One way ANOVA followed by Sidak's multiple comparisons test for (**a, b, d, e, g, h**). The box plots in (**d** and **g**) extends from the 25th to 75th percentiles and the whiskers go down to the smallest value and up to the largest where the box middle line is the median. Data are presented as mean values $+/-$ SEM. Source data are provided as a Source Data file for (**a, b, d, e, g, h**).

carbohydrate metabolism. PPARs can sense components of different lipoproteins and regulate lipid homeostasis based on the need of a specific tissue[26]. Three different PPAR subtypes are known; PPARα, PPARβ/δ, and PPARγ[26]. PPARγ is important in neurons for axon regeneration[31]. However *Pparγ* is not expressed in SGC according to our scRNAseq data. Rather, our results indicate that *Pparα* is enriched in SGC and required for efficient nerve regeneration, at least at early time points after nerve injury. Whether SGC contribute to sustain regenerative growth until the target is reinnervated and whether different neuronal subtypes are differentially affected by PPARα signaling in SGC remains to be tested. Fenofibrate is a specific PPARα agonist, making it unlikely that the rescue effects we observed in *FasncKO* mice are due to activation of PPARγ in neurons. PPARα and PPARα target genes are also highly enriched in SGC, suggesting that fenofibrate acts on SGC rather than other resident cells in DRG to enhance axon regeneration. We also show that fenofibrate does not improve axon regeneration in neuronal cultures that do not contain SGC. However, we cannot exclude the possibility that fenofibrate may operate via other mechanisms in other cell types in vivo. Fenofibrate is used clinically to treat lipid disorders, but was unexpectedly shown in clinical trials to have neuroprotective effects in diabetic retinopathy[78,79] and in traumatic brain injury[80]. Our findings suggest that fenofibrate may be used to activate SGC and enhance axon regeneration in circumstances with poor sensory axon growth, such as after injury to dorsally projecting sensory axons in the spinal cord or peripheral nerve repair. Although fenofibrate treatment did not improve locomotor recovery following spinal contusion injury in mice[81], a slight trend for increased tissue sparing was observed. Whether fenofibrate can improve centrally projecting sensory axon growth will need to be rigorously tested.

Biologic PPARα agonists consist of saturated and unsaturated phospholipids, eicosanoids, and glucocorticoids[26]. FASN synthesizes palmitic acid, which is the substrate for the synthesis of more complex lipid species[27]. FASN is partially localized in peroxisomes, where ether lipids such as plasmalogens are synthesized[27]. In the liver, FASN is required for generating the endogenous ligand for PPARα[28], whereas in adipose tissue, FASN is required for generating endogenous ligands for PPARγ[29]. Importantly, it is not palmitic acid per se that is required for PPAR activation but phospholipids. Phospholipid synthesis through the Kennedy pathway preferentially requires endogenous palmitate synthesis by FASN rather than circulating palmitate from the diet[28,29]. Our findings thus suggest that in SGC, FASN generates ligands PPARα following nerve injury and that FASN dependent PPARα activation contributes to promote axon regeneration. Future experiments to conditionally delete Pparα from SGC may further strengthen the case that FASN dependent PPARα activation in SGC after injury represents an important part of the injury response to stimulate axon repair.

How do lipid metabolism and PPARα signaling in SGC contribute to stimulate axon regeneration? Although addressing this question will require further investigation, our data indicate that *Fasn* expression in SGC participates in the regulation of a subset of regeneration associated genes after injury, including *Atf3* and *Gap43*. This is consistent with previous reports that ApoE-containing lipoproteins secreted from glial cells lead to the upregulation of *Gap43* mRNA in hippocampal neurons[33,82]. Together, these findings suggest that SGC contribute to regulate gene expression in neurons. It is also plausible that SGC impact neurons through lipid transfer. Indeed, lipids can be exogenously supplied by lipoproteins secreted from glial cells to stimulate neurite growth. Several studies have suggested a role for *ApoE*, a PPARα target gene, in lipid delivery for growth and regeneration of axons after nerve injury[34,82,83] and ApoE expression is increased in glial cells after nerve injury[36,37]. In agreement with these previous reports, our data indicate that *ApoE* mRNA levels are elevated in SGC after nerve injury and also following fenofibrate treatment. A recent study described downregulation of a few genes related to cholesterol metabolism after injury in SGC, but whether and how this contributes to axon repair has not been evaluated[38]. Cholesterol depletion was also shown to improve axon regeneration[84], which contrasts with earlier studies showing that cholesterol is required for axonal growth and can be synthesized in neurons or exogenously supplied from lipoproteins to axons of cultured neurons[34]. It will therefore be interesting to determine if lipid metabolism affecting cholesterol, fatty acids, and phospholipids in SGC regulates axon regeneration through paracrine effects on neurons.

Our study suggests that what has been defined as a neuronal intrinsic response to injury, namely upregulation of regeneration associated genes, such as *Atf3* and *Gap43* in neurons[1,2], may depend in part on signaling from SGC surrounding the neuronal soma. The neuron and its surrounding glial coat may thus form a functional unit that orchestrates nerve repair. Our single cell data set also highlights that other cell types, including endothelial cells, mesenchymal cells, and macrophages in the DRG respond to a distant nerve injury. Future studies are needed to elucidate the complex cellular cross talks operating in the DRG after nerve injury that contribute to nerve repair.

## Methods

**Animals and procedures**. All surgical procedures were completed as approved by Washington University in St. Louis School of Medicine Institutional Animal Care and Use Committee's regulations. During surgery, 8–12 week old C57Bl/6 mice of the indicate genotype were anesthetized using 2% inhaled isoflurane. For analgesia, 1 mg/kg buprenorphine SR-LAB (ZooPharm) was administered subcutaneously. For sciatic nerve injuries, the sciatic nerve was exposed at mid thigh level and crushed for 10 s using a #55 forceps coated with charcoal, which is used to label the injury site for analysis. The wound was closed using wound clips and both injured L4 and L5 DRG and sciatic nerve were dissected at the indicated time post-surgery. Contralateral nerve and DRG served as uninjured controls, when needed.

Tamoxifen (500 mg per kg diet, TD.130858) and Fenofibrate (0.2%, Sigma Cat# F6020) were administrated as chow pellets (Envigo Teklad).

**Mouse strains**. A 8–12 weeks old male and female mice were used for all experiments, except for scRNAseq experiment, where only C57Bl/6 females were used *Rosa26-ZsGreen* also known as Ai6(RCL-ZsGreen) was obtained from Jackson RRID: IMSR_JAX:007906; *Fabp7KO* mouse line was a generous gift from Dr. Owada[47]. The *Sun1GFP* mice (*Gt(ROSA)26Sortm5(CAG-Sun1/sfGFP)Nat*) is a generous gift from Dr. Harrison Gabel. Mice carrying floxed *Fasn* alleles were previously generated[56]. The *BLBPcre-ER* mouse line[50] was a generous gift from Dr. Toshihiko Hosoya.

**Single cell RNAseq**. L4 and L5 DRG's from mice 8–12 weeks old were collected into cold Hank's balanced salt solution (HBSS) with 5% Hepes, then transferred to warm Papain solution and incubated for 20 min in 37 °C. DRG's were washed in HBSS and incubated with Collagenase for 20 min in 37 °C. Ganglia were then mechanically dissociated to a single cell suspension by triturating in culture medium (Neurobasal medium), with Glutamax, PenStrep and B-27. Cells were washed in HBSS + Hepes +0.1%BSA solution, passed through a 70-micron cell strainer. Hoechst dye was added to distinguish live cells from debris and cells were FACS sorted using MoFlo HTS with Cyclone (Beckman Coulter, Indianapolis, IN). Sorted cells were washed in HBSS + Hepes+0.1%BSA solution and manually counted using hemocytometer. Solution was adjusted to a concentration of 500cell/ microliter and loaded on the 10X Chromium system. Single-cell RNA-Seq libraries were prepared using GemCode Single-Cell 3′ Gel Bead and Library Kit (10x Genomics), which recognizes a general gene but not a unique transcript or splicing variants, therefore we use the term 'gene expression' and not "transcript expression." A digital expression matrix was obtained using 10X's CellRanger pipeline (Build version 2.1.0) (Washington University Genome Technology Access Center). Quantification and statistical analysis were done with Partek Flow package (Build version 9.0.20.0417).

Filtering criteria: Low quality cells and potential doublets were filtered out from analysis using the following parameters; total reads per cell: 600–15000, expressed genes per cell: 500–4000, mitochondrial reads <10%. A noise reduction was applied to remove low expressing genes < = 1 count. Counts were normalized and presented in logarithmic scale in CPM (count per million) approach. An unbiased clustering (graph based clustering) was done and presented as t-SNE (t-distributed stochastic neighbor embedding) plot, using a dimensional reduction algorithm that shows groups of similar cells as clusters on a scatter plot. Differential gene expression analysis performed using an ANOVA model; a gene is considered differentially-expressed (DE) if it has an false discovery rate (FDR) step-up (p value adjusted). $p \leq 0.05$ and a Log2fold-change $\geq \pm 2$. The data was subsequently analyzed for enrichment of GO terms and the KEGG pathways using Partek flow pathway analysis. Partek was also used to generate figures for t-SNE and scatter plot representing gene expression.

**Tissue preparation and immunohistochemistry**. After isolation of either sciatic nerve or DRG, tissue was fixed using 4% paraformaldehyde for 1 h at room temperature. Tissue was then washed in PBS and cryoprotected using 30% sucrose solution at 4 C overnight. Next, the tissue was embedded in O.C.T., frozen, and mounted for cryosectioning. All frozen sections were cut to a width of 12 μm for subsequent staining. For immunostaining of DRG and nerve sections, slides were washed 3x in PBS and then blocked for in solution containing 10% goat serum in 0.2% Triton-PBS for 1 h. Next, sections were incubated overnight in blocking solution containing primary antibody. The next day, sections were washed 3x with PBS and then incubated in blocking solution containing a secondary antibody for 1 h at room temperature. Finally, sections were washed 3x with PBS and mounted using ProLong Gold antifade (Thermo Fisher Scientific). Images were acquired at 10x or 20x using a Nikon TE2000E inverted microscope and images were analyzed using Nikon Elements. Antibodies were as follow: SCG10/Stmn2 (1:1000; Novus catalog #NBP1-49461, RRID:AB_10011569), Tubb3/βIII tubulin antibody (BioLegend catalog #802001, RRID:AB_291637), Griffonia simplicifolia isolectin B4 (IB4) directly conjugated to Alexa Fluor 488 or Alexa Fluor 594 (Thermo Fisher Scientific catalog #I21411 and #I21413), Fabp7 (Thermo Fisher Scientific Cat #PA5-24949, RRID:AB_2542449), cleaved caspase 3 (CST Cat #9664, RRID: AB_2070042), Fasn (Abcam, Catalog #ab128870), Glutamine synthase (Abcam, Catalog #ab49873), PPARα (Thermo Fisher Scientific, Catalog #PA1-822A). Stained sections with only secondary antibody were used as controls.

**DRG cultures and regeneration assays**. For in vitro regeneration assay, DRG were isolated from time pregnant e13.5 CD-1 mice. DRG were kept in cold HBSS media until all DRGs were collected. After a short centrifugation, dissection media was aspirated and ganglia were digested in 0.05% Trypsin-EDTA for 25 min in 37 °C. Next, cells were pelleted by centrifuging for 2 min at 500 × g, the supernatant was aspirated, and Neurobasal was added. Cells were then triturated 25x and added to the growth medium containing Neurobasal media, B27 Plus, 1 ng/ml NGF, Glutamax, and Pen/Strep, with or without 5 μM 5-deoxyfluorouridine (FDU). Approximately 10,000 cells were added to each well in a 2.5 μl spot. Spotted cells were allowed to adhere for 10 min before the addition of the growth medium. Plates were pre-coated with 100 ug/ml poly-D-lysine. For regeneration assays,

PPARα agonists were added to the culture on DIV6. Fenofibrate (Cayman cat #49562-28-9), Clofibrate (Cayman cat #637-07-0), Wy14643 (Cayman cat #50892-23-4), GW7647 (Cayman cat #265129-71-3). Axons were then injured using an 8 mm microtome blade on DIV7 and fixed 24 h later. Cells were washed with PBS and stained for SCG10 as described above.

For adult DRG cultures, DRG were dissected from naïve and injured mice. Cells were prepared as described above for single cell protocol and cultured on 100ug/ml poly-D-lysine coated plates and fixed 20 h later. Cultures were then used for electrophysiological recording 24 h after plating or fixed and stained with the indicated antibody. Images were acquired at 10x using a Nikon TE2000 microscope and image analysis was completed using Nikon NIS-Elements (Version 4.60).

**Image analysis**. For sciatic nerve injury experiments, images were quantified in two ways. First, the injury site was defined as the area with maximal SCG10 intensity by drawing lines (16 × 32 pixels per line) perpendicular to the long nerve axis for every nerve section around the injury area (labeled by charcoal). The line with highest intensity was determined as the injury site. To measure the longest 10 axons, vertical lines were drawn away from the injury site till labeling SCG10 positive axon tips and the average length reported. To measure the regeneration index, SCG10 intensity was quantified at 1, 2, and 3 mm from the injury site using a line drawn perpendicular to the long nerve axis. The intensity was normalized to the injury site and the percent intensity was reported. This measure takes into account both the length and the number of regenerating axons. For both length and intensity quantifications, five sections per biological replicate were averaged.

For embryonic DRG experiments, regenerative length was measured from the visible blade mark to the end of the regenerating axons. Each technical replicate was measured 4–6 times and three technical replicates were measured per biological replicate.

To determine the cleaved caspase staining area, a binary was generated to fit the positive signal, and positive staining area was measured. That area was internally normalized to Dapi positive staining area.

**RNA isolation and quantitative PCR**. DRG and nerves were lysed and total RNA was extracted using Trizol reagent (Thermo Fisher, Cat #15596026). Next, RNA concentration was determined using a NanoDrop 2000 (Thermo Fisher Scientific). First strand synthesis was then performed using the High Capacity cDNA Reverse Transcription kit (Applied Biosystems). Quantitative PCR was performed using PowerUp SYBR Green master mix (Thermo Fisher, Cat #a25742) using 5 ng of cDNA per reaction. Plates were run on a QuantStudio 6 Flex and analyzed in Microsoft Excel. The average Ct value from three technical replicates was averaged normalized to the internal control Rpl13a. All primer sequences were obtained from PrimerBank[85] and product size validated using agarose gel electrophoresis.

Pparα (PrimerBank ID 31543500a1) Forward Primer AGAGCCCCATCTGTC CTCTC

Reverse Primer ACTGGTAGTCTGCAAAACCAAA

Pex11a (PrimerBank ID 6755034a1) Forward Primer GACGCCTTCATCCGA GTCG Reverse Primer CGGCCTCTTTGTCAGCTTTAGA.

Fads2 (PrimerBank ID 9790070c1) Forward Primer TCATCGGACACTATTC GGGAG Reverse Primer GGGCCAGCTCACCAATCAG.

Hmgcs1 (PrimerBank ID 31981842a1) Forward Primer AACTGGTGCAGAAA TCTCTAGC Reverse Primer GGTTGAATAGCTCAGAACTAGCC

ApoE (PrimerBank ID 6753102a1) Forward Primer CTGACAGGATGCCTAG CCG Reverse Primer CGCAGGTAATCCCAGAAGC

ATF3 (PrimerBank ID 31542154a1) Forward Primer GAGGATTTTGCTAACC TGACACC Reverse Primer TTGACGGTAACTGACTCCAGC

Gap43 (PrimerBank ID 6679935a1) Forward Primer TGGTGTCAAGCCGGA AGATAA Reverse Primer GCTGGTGCATCACCCTTCT

Jun (PrimerBank ID 6680512a1) Forward Primer TCACGACGACTCTTACG CAG Reverse Primer CCTTGAGACCCCGATAGGGA

Rpl13a (PrimerBank ID 334688867c2) Forward Primer AGCCTACCAGAAAG TTTGCTTAC Reverse Primer GCTTCTTCTTCCGATAGTGCATC

**Electrophoresis and western blot**. DRG and sciatic nerve samples were lysed in 5 × SDS loading buffer, heated at 95 °C for 5 min and were run in on 8–16% NuPAGE SDS-PAGE gradient gel (Life Technologies) in MOPS SDS Running Buffer (Life Technologies) and transferred to 0.45 μm nitrocellulose membranes (Amersham). Next, these membranes were blocked in blocking buffer 3% BSA in 0.1% Tween-20 at pH 7.6 for 1 h, and incubated at 4 °C overnight in blocking buffer with primary antibodies. Membranes were then washed in TBST 3 times, incubated with horseradish peroxidase-conjugated antirabbit or antimouse IgG (Invitrogen) for 1 h at room temperature, washed in TBST three times, and developed with SuperSignal West Dura Chemiluminescent Substrate (Thermo Scientific). The membranes were imaged using ChemiDoc System (Bio Rad) and analyzed with ImageLab software. Gapdh (Santa Cruz, catalog#sc25778) was used as loading control for quantification of the protein blots, Fasn intensity was normalized to the intensity of Gapdh.

**Whole-cell electrophysiology**. Whole-cell patch-clamp recordings in a current-clamp mode were performed using a Multiclamp 700B amplifier (Molecular

Devices) from short-term cultures (24 h after plating) of isolated DRG neurons and glia, visually identified with infrared video microscopy and differential interference contrast optics (Olympus BX51WI). Current-clamp recordings were made with pipette capacitance compensation and bridge-balance compensation. Recordings were conducted at near-physiological temperature (33–34 °C). The recording electrodes were filled with the following (in mM): 130 K-gluconate, 10 KCl, 0.1 EGTA, 2 MgCl$_2$, 2 ATPNa$_2$, 0.4 GTPNa, and 10 HEPES, pH 7.3. The extracellular solution contained (in mM): 145 NaCl, 3 KCl, 10 HEPES, 2.5 CaCl$_2$, 1.2 MgCl$_2$, and 7 glucose, pH 7.4 (saturated with 95% O$_2$ and 5% CO$_2$). For determination of AP threshold, APs were evoked by a ramp current injection (0.1 pA/ms)[86] with a hyperpolarizing onset to ensure maximal Na$^+$ channel availability before the first AP. The AP thresholds were determined only from the first APs of ramp-evoked AP trains. AP threshold (i.e., threshold voltage) was defined as the voltage at the voltage trace turning point, corresponding to the first peak of 3rd order derivative of AP trace[86]. Data were averaged over 5–8 trials for each cell. Resting membrane potential (RMP) was measured immediately after whole-cell formation. Cell capacitance was determined by the amplifier's auto whole-cell compensation function with slight manual adjustment to optimize the measurement if needed. Under current-clamp mode, a negative current (−50 pA for 500 ms) was injected every 5 s to assess the input resistance.

**TEM**. Mice were perfused with 2.5% glutaraldehyde and 4% paraformaldehyde in 0.1 M Cacodylate buffer, followed by post fix. A secondary fix was done with 1% osmium tetroxide. For Transmission electron microscopy (TEM), tissue was dehydrated with ethanol and embedded with spurr's resin. Thin sections (70 nm) were mounted on mesh grids and stained with 8% uranyl acetate followed by Sato's lead stain. Sections were imaged on a Jeol (JEM-1400) electron microscope and acquired with an AMT V601 digital camera. (Washington University Center for Cellular Imaging).

**Quantification and statistical analysis**. Quantifications were performed by a blinded experimenter to genotype and treatment. Fiji (ImageJ) analysis software was used to measure TEM nerve images. Nikon Elements analysis software was used to trace regenerating axon in nerve sections and in the embryonic DRG spot culture. An Automated analysis for axon tracing and neurons soma count was used for ex-vivo adult DRG culture experiments using "General Analysis" for skeletonize neurites with no background reduction. Parameters were set to Bright Spot Detection of circular area with typical soma diameter 50 μm, Cut Branches Recursively on Skeleton set to smaller then 15 px, Filter on Line Length was set to Min 60 μm to remove short filaments. (Nikon elements commercial software package, code is available upon request). Statistics was performed using GraphPad (Prism8) for either Student's t test or ANOVA analysis. Sidak or Dunnett tests were used as part of one- and two-way ANOVA for multiple comparison tests. Error bars indicate the standard error of the mean (SEM).

**Reporting summary**. Further information on research design is available in the Nature Research Reporting Summary linked to this article.

## Data availability
The raw Fastq files and the processed filtered count matrix for scRNA sequencing were deposited at the NCBI GEO database under the accession number GSE139103. Processed data are also available for visualization and download at https://mouse-drg-injury.cells.ucsc.edu/. Data analysis and processing was performed using commercial code from Partek Flow package at https://www.partek.com/partek-flow/. Dataset analyzed in Supplementary Fig. 2f obtained from NCBI GEO under the accession number GSE120678. Dataset analyzed in Fig. 7b obtained from NCBI GEO under the accession number GSE125685. Axon tracing and neurons soma count was performed using Nikon NIS-elements, which is a commercial software package for image analysis. The specific analysis code for digital reconstruction of axons is available upon request (requires NIS-Elements and General Analysis) https://www.microscope.healthcare.nikon.com/products/software/nis-elements. Source data are provided with this paper.

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

## Acknowledgements

We would like to thank members of the Cavalli lab for valuable discussions. We would like to thank the following investigators for their generous gifts of mouse lines: Dr. Toshihiko Hosoya for the BLBP-cre ER mice, Dr. Yuji Oawada for the Fabp7KO mice, and Dr. Harrison Gabel for the Sun1GFP mice. We gratefully acknowledge Greg Strout, Ross Kossina, and Dr. James Fitzpatrick from the Washington University Center for Cellular Imaging (WUCCI), which is supported in part by Washington University School of Medicine, The Children's Discovery Institute of Washington University, and St. Louis Children's Hospital (CDI-CORE-2015-505 and CDI-CORE-2019-813) and the Foundation for Barnes-Jewish Hospital (3770) for assistance in acquiring and interpreting Transmission Electron Microscopy (TEM) data. We also thank Anushree Seth and Madison Mack in association with InPrint for illustration in Fig. 1a. This work was funded in part by a post-doctoral fellowship from The McDonnell Center for Cellular and Molecular Neurobiology to O.A., by NIH grant NS111596 to V.A.K., and by The McDonnell Center for Cellular and Molecular Neurobiology and NIH grant NS111719 to V.C.

## Author contributions

O.A. and V.C. wrote the manuscript. O.A., P.D., R.K., C.F.S., V.A.K., and V.C. designed research as well as edited and approved the manuscript; O.A. performed sequencing,

bioinformatics, in vitro and in vivo experiments, analyzed the data and wrote the manuscript; P.D. performed electrophysiological recordings experiments; S.J. analyzed and quantified in vitro and in vivo regeneration assays and TEM images.

## Competing interests

The authors declare no competing interests.
