## [Peer Review File · Nature Communications]

Reviewers' Comments:

Reviewer #1:

Remarks to the Author:

In this manuscript titled "Fatty acid synthesis in satellite glial cell promotes regenerative growth in sensory neurons", Avraham and colleagues provide strong evidence that satellite glial cells (SGC) play a previously unknown role in sensory neuron regeneration after axonal injury. Using single-cell RNA sequencing of mouse DRG in naive and injured conditions, the authors demonstrate that SGCs are molecularly distinct from other glial cells in the DRG and they can be labeled with a specific marker called Fatty acid-binding protein 7 (Fabp7 or BLBP). They also discover an upregulation of lipid metabolism pathways in the SGCs of injured DRG compared to the naive. The authors then utilize transgenic mice expressing Cre recombinase under the BLBP promoter to manipulate lipid metabolism in SGCs by ablating fatty acid synthase (Fasn). This leads to a novel finding that the deletion of Fasn in SGCs impairs the axon regeneration of DRG neurons post-injury. And this impairment can be rescued by activating the downstream target of Fasn, peroxisome proliferator-activated receptors alpha (PPAR α). The investigators then use both in vitro and in vivo assays to screen synthetic fatty acid compounds and demonstrate that dietary supplement with two of these compounds, clofibrate, and fenofibrate, is enough to rescue proper axon regeneration by DRG sensory neurons. From these data, the authors propose a mechanism for the transcriptional response of SGCs after an injury that is mediated via peroxisome proliferator-activated receptors (PPARs), and support this claim by rescuing the impairments in axon regeneration in their Fasn cKOs via PPAR activation. The work presented by the authors, both in vivo and in vitro, strongly support their overall conclusions and reveal an important contribution of SGC fatty acid synthesis pathway in axon regeneration after peripheral nerve injury. The manuscript is well written and the authors should be commended that they addressed both the limitations of specific techniques but also highlighted the importance of present findings, which are likely to lead to future areas of study. As it stands, the work is largely ready for publication, however, there are a few minor concerns/points listed below, which through addressing would strengthen the work further and also improve the clarity of text and figures.

Figures:

1. Figure 2D is difficult to interpret because there is no clear marker of the injury site for the reader to use as a reference for the proximal and distal site of injury.
2. The legend for figure 3D does not include the size of the scale bar.
3. In figure 4A, the SGCs enveloping the DRG cell body in the Fasn cKO condition appears to be larger compared to the SGCs in the wildtype condition. Is this a common difference between conditions or just the choice of this image? If there is a size change, quantification of SGC size may be necessary to determine whether or not Fasn deletion alters SGC morphology.
 - a. The authors should highlight which part of the top images in figure 4A was enlarged for closer observation in the bottom images.
 - b. Is the magnification of both bottom images the same? The bottom right image appears to be of a higher magnification compared to the bottom left image.
4. Please provide sample sizes for figures 4C and 4D in the figure legends.
5. There was no clear explanation of what exactly g-ratio is measuring in figures 4D and 4E.
6. The representative images used for the FABP7 KO condition in Fig. S2A appears to be of a higher magnification compared to the control.
7. There appears to be more FABP7 positive fluorescence in the FABP7 KO condition compared to the Control condition in Fig. S2C.

Text:

1. Missing word: The SGC cluster identity ... SGC markers such (as) Kir4.1.
2. One interpretation of our findings is that in SGC ... promote axon regeneration. (This statement needs to be rewritten for better clarity).
3. Missing word: DRG sections from Fabp7KO and control mice were (immunostained) for and Tuji (red).

4. The figure S3 legend title should be bolded for consistency.

Out of the scope of the current study but should be discussed in the manuscript as possibilities or future directions:

1) In figure 6, the authors make use of an in vitro system to screen for synthetic fatty acid compounds that could rescue the regeneration impairment they have found in their FASN conditional knockout animals. However, the authors do not test the endogenous fatty acid, palmitic acid, in its ability to rescue this phenotype. One advantage of this approach the authors mention is the specificity synthetic compounds provide for a particular pathway in SGC versus neurons, however, it would be helpful to have an idea of the dose-response to the endogenous PPAR ligand for the interpretation of certain aspects of the data namely, that low doses of certain compounds seem to be worsening the phenotype (GW7647 100 nM) while at higher doses they appear to be rescuing this phenotype.

2) In figure 7, the investigators show that fenofibrate dietary supplement is sufficient to rescue the axon regeneration impairments they find in the FASN conditional knockout animals. While the authors point out that fenofibrate preferentially activates PPAR, which is enriched in SGC, they cannot rule out a potential role for PPAR signaling in other cell types during this process. The authors could perform the necessity experiment by conditionally knocking out PPAR in SGC and test whether this also leads to the same predicted impairment and test whether fenofibrate is still able to rescue this impairment. This would strengthen the case that this response is primarily mediated by SGC as opposed to other cell types.

Reviewer #2:

Remarks to the Author:

Avraham et al report on the requirement of fatty acid synthase in satellite glial cells to support the sensory axon regeneration. Through the single cell seq approach, they identified several genes enriched in SGC but not in Schwann cells. One of them, BLBP, was used to label SGC and BLBPCre-ER was used to manipulate SGCs. Further analysis showed that lipid metabolism genes were upregulated in SGC by sciatic nerve injury. In BLBPCre-ER;Fasn conditional KO mice, peripheral sensory axon regeneration was mildly suppressed after crush injury, especially for the fastest growing axons and at 2mm distal to the lesion. Consistently, in the isolated DRG culture, axon elongation induced by conditioning lesion was partially suppressed. Then, through a pharmacological approach in vitro and in vivo, they proposed that PPAR α may function downstream in the axon growth effect shown in Fasn KO mice.

It is always challenging to study cell-cell Interaction. The hypothesis is of considerable interest, to propose the interaction between SGC and DRG neurons through single cell analysis. However, there are some deficiencies that reduce enthusiasm.

Major points

1. In the single cell profiling, DRG neurons were not adequately represented. Because there are quite a few reports on the single neuron seq by 10X, it will be important to demonstrate that the isolation protocol worked properly. Also, Fabp7 was the only gene that was verified through another approach. How about other genes?

2. It is interesting to see that axon injury can regulate gene expression in SGC. How about other aspects, such as cell number? Do they proliferate or differentiate after injury? These are fundamental to understand the role of SGC in axon growth in response to injury.

3. Although gene expression data were shown, it is still not clear what happened in SGC after

injury. Which step of the fatty acid metabolism is regulated by injury, fatty acid synthesis, uptake or turnover? Why Fasn was chosen to study? It appears that Fasn is not upregulated after injury, which makes it unlikely to be the mediator of fatty acid metabolism change, if any. If fatty acid synthesis is the key to mediate the effect, fatty acid level should be directly measured in SGC, neurons or other cell types, and whole DRG in naive and injured conditions. Does Fasn KO change the fatty acid level in SGC? The same argument can be applied on FABP7, which could mediate fatty acid update.

4. All the in-vivo and in-vitro experiments cannot exclude the role of schwann cell in axon regeneration. Fig2A indicates that FABP7 have no staining in schwann cells in DRG, but in Fig2B some schwann cells and even some other cell types indeed have high expression of FABP7. It should be noticed that FABP7 is upregulated in schwann cells in nerve after injury. It is not clear how Tamoxifen was given. If tamoxifen diet was still given at the time of injury, it's still possible BLBPCreER may also have leakage expression in schwann cells after injury. Authors should also check FASN staining in the sciatic nerve in FASNcKO mice after injury.

5. It is very difficult to interpret the regeneration data shown in Fig 7B-D. What is the difference between the length of 10 longest axons (median at 3000um) vs. regeneration index at 3000um? The index difference was only seen at 2000um.

6. The interpretation of the Fenofibrate diet in vivo experiment could be many ways and it is premature to conclude that it is through SGC rather other other possibilities. A more specific regulation on SGC is required to draw a conclusion.

Minor points

Authors used Tuj1 staining in all the DRG sections. To validate the expression pattern of FABP7 or FASN in different scenarios, it would be better to use specific SGC and schwann cell markers for co-staining.

Before analyzing transcription change in SGC, it's important to quantify the number of SGC or other cell types in DRG before and after injury.

In Fig7E, to draw the conclusion "fenofibrate rescues the impaired axon regeneration in FasnKO mice", a complete group of WT naïve, WT injured, WT naive+fenofibrate, WT injured+fenofibrate should be included into the experiment.

Reviewer #3:

Remarks to the Author:

This manuscript from Cavalli and colleagues reports on work examining the biology of satellite glial cells of the dorsal root ganglion. The work is highly novel, reporting a number of new characteristics of the SGCs, including a transcriptional profile providing a set of SGC-specific genes, their intimate relationship with sensory neurons (possibly constituting a functional unit), responsiveness to nerve injury (which is a stimulus that is remote to them), and the potential of a regulatory role in the sensory neuron response to the injury, including at least some aspects of axonal regeneration. It is largely a very well-controlled and designed study with strong quality control on reagents and on genetic manipulations. The work provides specific mechanisms where only general suggestions were available previously, and promises to be welcomed by the field as a cornerstone for future reference. There are a few points that require attention to ensure the reality meets the potential. It is likely that no new data is required (though the authors may wish to add), but simply revision in the text to enhance clarity and credibility, particularly regarding axonal regeneration. There is sufficient over-interpretation of some data to warrant revision. The most significant single point appears to be the treatment of the data related to regeneration, and the

framing of the manuscript. Reading the Title, Summary, Abstract, and Introduction would lead one to believe that the manuscript is largely concerning the regulation of axonal regeneration by SGC-derived fatty acids. But this is not the actual major content of the manuscript, and the data supporting that message are not the strongest in the manuscript. The manuscript appears to base many the strongest claims on the effects on expression of ATF3, but the assessment of ATF3 expression is not sufficient to support those claims. It bases other strong claims on assessments of an unspecified very small group of axons/neurons and uses those observations to extend the claims to the entire population. The claims may yet be true, but they are not sufficiently-supported. The authors need to revise some of the claims and should consider re-framing the manuscript to center on the strongest and greatest-volume of the data – SGC-neuron structural relationship and SGC biology – OR they can enhance the data related to regeneration to bring those data to the point where they can adequately support the focus of the message on axonal regeneration.

Introduction

“In adult animals, multiple SGC form an envelope that completely enwraps each sensory neuron 5-7. The number of SGC surrounding sensory neurons increases with increasing soma size in mammals 8, 9. The neuron and its surrounding SGC thus form a distinct morphological and functional unit”

- The statement “The neuron and its surrounding SGC thus form a distinct morphological and functional unit” seems a bit premature in this place in the manuscript. The statements preceding this in the Introduction only suggest the possibility of a functional unit (correlative). Although the data generated and analyzed in this report somewhat support the claim of a functional unit, those data have not been presented yet in the manuscript. The claim at the end of the Introduction (“These results also highlight that the neuron and its surrounding glial coat form a functional unit that orchestrates nerve repair”) is much more appropriate as it follows from a summary of the data to be presented. (However, there remains a concern that those data may not support that conclusion, as outlined below)

“Structural neuron-glia units, similar to these, do not exist in the central nervous system”
This could be debated. The authors may wish to provide some more description to clarify their point and ensure credibility.

“These studies show that communication between neuron and SGC is critical for neuronal excitability and nociception...”

It is not clear that those cited studies indicated a relationship that is “critical”. The author’s own language elsewhere implies a less-direct relationship (“modulate” as opposed to “regulate” or “is required for”).

With the exception of a description of SGCs, the overall design of the Introduction narrative lacks a relationship to the major experimental manipulations and conclusions. The major GOF/LOF manipulations focus on SGC Fasn and PPAR-alpha and their role in modulating regenerative axon growth, but there is nothing in the introduction that speaks to anything that is known about these systems in axonal regeneration, even though there are relevant data. Describing those data will provide better context and does not impact the novelty or impact of the current work. These changes are only suggested.

RESULTS and METHODS

“We anticipated obtaining a larger representation of neurons in our dataset, but their large cell diameter (up to 50 micron for proprioceptors), is likely limiting them from passing through the microfluidic pipes in the Chromium 10X device.”

- It is not clear if the speculation is that all neurons are impeded through the microfluidics, or only the large diameter. There should be data available from the company or core facility regarding the tolerance of the microfluidics. The vast majority of neurons are small and medium, so the implication that the small proportion of neurons that are 40-50um might account for the unexpectedly low recovery of neurons is not entirely logical.
- It is also possible that the large cells died, as they are more susceptible to mechanical dissociation.
- Examination of genes specific for, or enriched in, different populations of sensory neuron types, could be useful to determine if the recovery/loss was selective. Ultimately, however, it is not clear why this speculation is included in the Results. It may be more suitable for the Discussion.

The manuscript must enhance its description of the bioinformatic and gene-expression analyses. The existing description is terse and not sufficient for reviewers to determine suitability or to provide utility for readers. For example, the description: "...using the Chromium Single Cell Gene Expression Solution (10x Genomics) (Figure 1A). Graph-based clustering, using Partek flow analysis package..." is not at all sufficient for assessment or reproducibility.

For further example: "We thus examined the transcriptional similarity between astrocytes and SGC by comparing the top expressed genes in each cell type, using our scRNAseq data for SGC and a previously published transcriptional analysis of astrocytes (500 genes). We found that SGC share about 10% of genes with astrocytes..."

- This description (not supplemented by anything useful in the Methods) does not describe how "top expressed genes" were determined, or what cut-offs were used to determine expression levels for comparing between cell types.

"We next compared the top expressed genes in SGC (605 genes) and Schwann cells (572 genes), which revealed that SGC share only 2% of genes with Schwann cells in the DRG"

- Correct to "...share only 2% of THOSE gene TRANSCRIPTS with..."
- Again, not clear what criteria establishes the "top expressed" genes.

The manuscript includes excellent data demonstrating that the Fabp7 antibody does not bind elsewhere – examined in KO mouse tissue – but it is not clear that there are data showing it does bind to Fabp7 (although it does bind to the expected cells). Clarification of this point would strengthen the manuscript.

"We observed no abnormalities in the morphology of the SGC surrounding the neuron somas in the FasnckO animals compared to controls..."

- Unless there was a quantitative assessment of some set of criteria, this should be corrected to read "We observed no QUALITATIVE abnormalities...". If a quantitative analysis was performed it should be described.

[The examination of ATF3 was a strength of the project – it could have revealed stress on neurons or other cells (SGC) related to the KO. Unfortunately, the strength was not fully-realized because only the mRNA from homogenized DRG was examined, so cell-type specificity is absent.]

Using cleaved-caspase as an indicator of cell death assumes only a single mechanism of death (and stereological counts were not performed, which could have revealed a difference in overall cell number, which could have indicated cell death via some other mechanism(s)). This is largely acceptable because it represents the mechanism known to exist in WT animals, but it should be acknowledged in the Results or Discussion that other forms of cell death exist and were not assessed.

The reduced ATF3 expression in the injury condition was assessed by qPCR and the interpretation of the data appear to assume that the DRG neurons are the only source of post-injury ATF3. Regardless, the difference could be due to decreased expression per cell or fewer cells expressing at the same level/cell as control. Also, homogenized DRG includes SGC, which might (do?) express

ATF3 after nerve injury. The reduced ATF3 mRNA could be related in some way to the loss of Fasn in SGC, as implied in the text. However, the text currently implies the reduced ATF3 is only in neurons and so the effect of Fasn-KO in SGC is somehow trans-cellular, but this is not necessarily true. This possibility can be dealt with in two ways – either the interpretation can be expanded to allow for these possibilities, or new data can be collected to assess ATF3 expression in terms of number and type of cell, as well as magnitude per cell (if necessary).

Average axon diameter (Figure 4C) may not be the best measure, because there is such a wide range of sizes with very different numbers giving rise to the different ends of the spectrum. A cumulative sum histogram (or similar) would be far more representative of reality.

“The robust bidirectional communication between neurons and SGC is critical for neuronal excitability and nociception and the excitability of sensory neurons is controlled in part by the surrounding SGC”

- As with the comment regarding a similar claim in the Introduction, please ensure that the terms “critical” and “controlled” are accurate representations of what the cited articles show.

“Whole-cell recordings were performed in acutely dissociated cocultures of DRG neurons and glia”

- Electrophysiological recordings starting 24h after plating does not qualify as recording from “acutely dissociated” DRG neurons. Acutely-dissociated neurons from naïve animals rarely fire multiple APs to the depolarizing stimulus applied, but neurons from cultured DRG neurons or from injured DRG neurons (acute-dissociated or cultured) can show repetitive firing, as is shown in Figure 4L. The control and KO are similar, so this difference is likely not important in terms of the effects of the KO, but it matters for 1) credibility and 2) to provide proper context for readers. Please correct any instances to “short-term culture” or a similar term, not “acutely-dissociated” (observed at least in Results and Methods).

“Therefore, Fasn deletion in SGC does not affect functional properties of DRG neurons IN THE NAÏVE DISSOCIATED CONDITION.”

Regarding the effect of Fasn-cKO on axonal regeneration and the priming-effect:

- The in vitro work used DRG taken from E13.5 mice, which were maintained in vitro for 7d before additional injury (achieving a relative “age” of E20.5), and fixed 24h later (achieving a relative “age” of E21.5/P0). This time is largely considered still developmental for sensory neurons, in which injury is more an effect on the developmental program (outgrowth and survival) than on the adult injury/stress response, which can include axonal regeneration. The data can still have value, of course, but perhaps should not be labeled as regeneration per se, especially when other aspects of the project are examining true regeneration. Making a terminological equivalence between these models suggests a mechanistic one, which is not a given, and certainly not supported in the narrative. Unless the authors provide some rationale in the text (not just in response to the reviewer) to use the same terms for these different models, the Embryonic injury and growth model should be given another term, perhaps “developmental axonal growth”, “developmental axonal response to injury”, or something similar.

“For adult DRG cultures DRG were dissected from naive mice.”

- Provide the age/sex of the mice.

“Contralateral nerve and DRG served as uninjured controls, when needed.”

- Please indicate if the contralateral side was sham (and if so if it was skin incision with/without muscle dissection) or intact.

The model uses a 3d post-crush time for regeneration assessment. This seems to offer a limited picture. It does not allow for assessing the normal range of “initiation rates” known to exist across the different types of peripheral axon types. It limits this assessment to only the fastest responders, but the claims are made broadly. It is entirely possible that the neurons which are

slower-initiators in the WT have an identical initiation rate in the KO, but this possibility is not addressed.

The images of the nerve in the figure imply a rather significant qualitative difference in the architecture of the distal nerve, with the KO nerve looking much closer to normal (with visible undulations), and even with what appear to be SCG10+ structures far distal to the injury site. This should be addressed better. Providing images of axonal structure (beta-3 tubulin?) would significantly enhance the context for readers.

"...the injury site was defined as the area with maximal SCG10 intensity."

- Please indicate if this was established from a single section, or a consensus across many sections. Also indicate what qualified as "area with maximal SCG10" – a single pixel? 10 contiguous pixels? Etc. This method of defining the injury site may have been convenient, but it is not (at least as presented here) terribly clear or strong. Using markers of denervated Schwann cells or infiltrating cells, even as a proof of concept for the SCG10-intensity method, would have strengthened the assessment significantly. This point of clarity matters greatly as it is the basis for the quantitative measures.

- Certainly there are differences detected, but the 3 day post-injury time point seems sub-optimal for the claims made. It is at the leading edge of what is reported for the latency to the cell-body reaction and start of axonal regeneration, which would fit well if the only goal were to assess the fastest response-initiation. Although that is included, the manuscript also uses these data to support claims regarding the regeneration process in general. Assessments at later time points would have greatly strengthened the claims. The claims throughout the manuscript regarding regeneration should be revised to reflect this point.

"Neurons in injured FasnKO displayed reduced neurite length compared to injured controls, but similar initiation rates (Figure 5D-F)."

- The outcome measure relies on the "longest 10 axons". There is nothing intrinsically flawed in this, but it does introduce a bias toward the most effective and/or fastest-starting axons. This is problematic because these may be a subpopulation, or the effect of Fasn/PPAR may only be on the - It is possible that the "initiation rate" was unaffected by the Fasn-KO as claimed, but there are other interpretations that could exist (false-negative). As described, the "initiation rate" was determined by the post-plating time at which neurites emerged to some qualifying characteristic. This was done without regard to the type of sensory neuron giving rise to those neurites. Unless all SGC express the same level of Fasn, and all DRG neurons express the same level of PPAR, then it is possible that the neurons that are responsible for providing the "equivalent initiation rate" are a subpopulation that, in WT, have some reduced/different PPAR-expression/function and/or are surrounded by SGC with lower Fasn (Figure 3 appears to indicate that there may be some significant heterogeneity in the Fasn expression). In this case, the loss of Fasn (in the KO) would largely be irrelevant as that subpopulation did not have a high degree of SGC-Fasn and neuronal PPAR anyway, and the conclusion could be incorrect. This can be addressed either by providing data regarding the uniformity of the PPAR-expression across neuron types AND the uniformity of Fasn-expression across SGC, OR the assessment of "initiation rate" can be done with sensitivity to neuronal sub-types. Alternatively, the manuscript can alter the claim and provide better presentation of the possible interpretations of the data.

- Using the in vivo outcome to determine "initiation" with the in vivo outcome of "length" as was done leads a particular problem. There is no means for determining if the populations are the same. This again is not necessarily a flaw, except that the narrative regarding both outcomes implies both a uniformity across the entire population of neurons, and implies a union of the population of axons that are longest and first to grow in vitro. These implications are not supported by the experimental design, though they may be true. Therefore, either new data must be provided demonstrating the uniformity and union, or the narrative must be adjusted to clearly provide to the reader an interpretation of the data that includes these possibilities and limits.

The outcome measure of "initiation rate" may have additional issues. It appears to be less about

the latency of when the range of axons start to regenerate and more about the proportion of neurons that display neurite-outgrowth at a given time. It is not clear that "rate" is a suitable term in the in vitro case. Perhaps "proportion" or similar term would be more accurate.

"These results indicate that Fasn in SGC is required for the conditioning effect, and specifically for the elongating phase of axon growth"

- The conclusion of "required" is not supported. Fasn may influence the priming effect, but even this may not be well-supported. Fasn may regulate, to some unknown degree, the priming effect in some population of neurons. However, the limited assessment time does not lend itself to a thorough assessment of priming or regeneration, particularly the elongation phase.

"...SCG10 intensity was quantified at 1, 2, and 3mm from the injury site as previously described"

- A brief description needs to be provided. It is not clear if the quantification was the average intensity of a row/line of single pixels (e.g., the demarcation line), or some area of pixels.

"For both LENGTH AND INTENSITY quantifications, five sections per biological replicate were averaged."

In standard situation, PPAR-g is neuronal and PPAR-a is SGC. However, it appears that this pattern was not ensured in the Fasn-cKO. This weakens the conclusion (assumption?) that the fenofibrate (PPAR-a-agonist) is working only on SGC, especially since the effective concentration of fenofibrate was not determined. This weakness is mitigated by the data that fenofibrate does not improve axon regeneration in neuronal cultures that do not contain SGC. The authors should consider including a statement to clarify the context, or doing some direct assessments.

DISCUSSION

"The role of SGC in nerve regeneration has been largely ignored..."

The term "ignored" is inaccurate. It implies that the role was known but not considered. But the role was not really known until this work – this is even stated in the following sentence. Please reword.

"Although addressing this question will require further investigation, our data indicate that Fasn expression in SGC is required for the proper expression of at least one regeneration associated gene after injury, Atf3"

- The conclusion of "required" is not supported.

"Our study suggests that neuronal "intrinsic" response to injury may not be purely neuronal but rather includes the SGC surrounding the neuronal soma..."

- Unless this statement is referring to something other than the neuronal ATF3 response (which the authors have not specifically identified) as the "intrinsic" response, then the statement is not supported.

METHODS

"...and both injured dorsal root ganglia and sciatic nerve were dissected at the indicated time post-surgery"

- Specify which DRGs were designated as "injured" and collected.

Specify the control procedures for immunohistochemistry and western blot.

"Cultures were then used for electrophysiological recording 24h after plating or fixed and stained with the indicated antibody"

- Electrophysiological recordings starting 24h after plating does not qualify as recording from

"acutely dissociated" DRG neurons, as indicated in the Results.

Figure 4H shows results from qPCR of ATF3, but there are no primer sequences provided for ATF3. Please provide the primer information.

"An Automated analysis for axon tracing and neurons soma count was used for ex-vivo adult DRG culture experiments (Nikon elements)."

- Please provide specifics for the "automated analysis".

"Statistics was performed using GraphPad (Prism8) for either Student's t-test or ANOVA analysis."

- This is not sufficient. Please ensure that this section is complete and that the description of the statistics in Methods and elsewhere are consistent and clear.

- Indicate type of ANOVA

- Perhaps the authors have omitted mention of the post-ANOVA tests in the description in the Methods. Figure 2 legend mentions the Sidak multiple comparison test, suggesting this terse description in the Methods is in error.

- Gene expression mentioned use of FDR, but this is not described.

- Most legends indicate which statistical tests were used, but not all. Figure 4 legend has p-values but no indication of the tests used

Minor

"Structural neuron-glia units, similar to these, do not exist in the central nervous system"

Commas likely not necessary

Throughout, please ensure that the language consistently and properly uses the terms "gene" and "transcript". In many places the term "gene" is used where "transcript" should be used.

"Our single cell RNAseq analysis using freshly dissociated tissue thus UNRAVELS a unique transcriptional profile of SGC in DRG..."

"We found that SGC share about 10% of THOSE gene TRANSCRIPTS with astrocytes, among them Fabp7..."

"We crossed BLBPCre-ER to the Rosa26-fs-TRAP and observed that following 10 days OF tamoxifen treatment..."

OR

"We crossed BLBPCre-ER to the Rosa26-fs-TRAP and observed that following A 10 day tamoxifen treatment..."

"Pathway analysis of other major CELL types in the DRG confirmed fatty acid metabolism as a unique pathway enriched in SGC..."

"...our pathway analysis INDICATED that the cell cycle term is enriched in macrophages, but not SGC..."

Also, this sentence is awkward. The "cell cycle term" is not enriched in macrophages. Please reword.

"...loss of Fasn in SGC does not lead to morphological deficits in the DRG and does not ELICIT a stress response in neurons"

"Fenofibrate is used clinically to treat lipid disorders, but has been unexpectedly been shown in clinical trials..."

- "been" is redundant and grammatically incorrect

"During surgery, 8-12 week old C57Bl/6 mice OF the indicate genotype were anesthetized"

Review provided by Jeffrey C. Petruska, Ph.D.

Reviewer #4:

Remarks to the Author:

Overview of the study:

In their manuscript titled Fatty acid synthesis in satellite glial cell promotes regenerative growth in sensory neurons, Avraham et al. have first used scRNA-seq to study the non-neuronal cell populations of mouse dorsal root ganglia in both naïve and injured conditions, focusing on the satellite glial cells (SGC) that surround all sensory neurons. They show that SCGs can be distinguished from Schwann cells by their gene expression profiles and define Fatty acid binding protein 7 (Fabp7) as a specific marker for SGCs in the peripheral nervous system. The authors further show evidence that crush nerve injury upregulates fatty acid metabolism in injured DRGs. Using a SGC-specific knock-out of the Fatty acid synthase gene (Fasn) together with TEM, injury/apoptosis markers, and electrophysiology, they show that the deleting this gene in SCGs has little impact on the properties of uninjured DRG neurons, but the regeneration abilities of these neurons appear compromised. Avraham and colleagues then show that this deficit in regeneration ability can be rescued in a neuron-independent fashion by feeding in vitro embryonic cells or intact adult animals with a specific PPAR α agonist, Fenofibrate. Based on their results, the authors suggest that fatty acid synthesis and PPAR α -signaling in SGCs are fundamental to axon regeneration in adult peripheral nerves and highlight the functional connectivity of the sensory neuron and its glial coat.

General remarks:

This is an important study for the field of sensory neurobiology, with novel results underscoring the role of satellite glial cells in peripheral nerve physiology. The study is of high quality, well written, and based on a solid set of experiments. Results from the laboratory experiments are clear and give strong support for the claims made by the authors. The scRNA-seq part is straight-forward and the results appear to be sound; however, to be considered for publishing, I invite the authors to address the concerns I have listed below.

Issue 1:

While the scRNA-seq analysis appears appropriate, the authors have not disclosed any information on how this analysis was done, apart from mentioning that a proprietary software package (Partek Flow) was used for the analysis. Therefore, for a proper evaluation of the analysis, the information below needs to be added to the paper in a way that the analysis can be reproduced.

How was the data quality controlled?

What algorithms were used for the analysis?

What were the most important parameters chosen for each critical step of the analysis? What statistical testing was used for differential expression analysis between clusters?

How were the comparisons between datasets done (comparisons to neurons and astrocytes)?

How was the KEGG analysis done?

Issue 2:

The authors should add a small section to the discussion where they compare their results to those in this recent paper dealing with satellite glia after injury.

Glia. 2020 Feb 11. doi: 10.1002/glia.23785.

Changes in the transcriptional fingerprint of satellite glial cells following peripheral nerve injury.

Minor issues:

Please increase font size in Figures 1D and S1B (currently unreadable) and inform in the figure legend (Figure S1B) what the x-axis shows.

Please add to the GEO database a raw expression matrix of the data with cell-cluster and cell-replicate annotation. This will help other researchers to use the data more easily.

Figure 3D says "Scale bar: xx μm "

Detailed responses to reviewers.

Reviewers' comments are in black. Authors' responses are in blue.

Reviewer #1 (Remarks to the Author):

In this manuscript titled “Fatty acid synthesis in satellite glial cell promotes regenerative growth in sensory neurons”, Avraham and colleagues provide strong evidence that satellite glial cells (SGC) play a previously unknown role in sensory neuron regeneration after axonal injury. Using single-cell RNA sequencing of mouse DRG in naive and injured conditions, the authors demonstrate that SGCs are molecularly distinct from other glial cells in the DRG and they can be labeled with a specific marker called Fatty acid-binding protein 7 (Fabp7 or BLBP). They also discover an upregulation of lipid metabolism pathways in the SGCs of injured DRG compared to the naive. The authors then utilize transgenic mice expressing Cre recombinase under the BLBP promoter to manipulate lipid metabolism in SGCs by ablating fatty acid synthase (Fasn). This leads to a novel finding that the deletion of Fasn in SGCs impairs the axon regeneration of DRG neurons post-injury. And this impairment can be rescued by activating the downstream target of Fasn, peroxisome proliferator-activated receptors alpha (PPAR α). The investigators then use both in vitro and in vivo assays to screen synthetic fatty acid compounds and demonstrate that dietary supplement with two of these compounds, clofibrate, and fenofibrate, is enough to rescue proper axon regeneration by DRG sensory neurons. From these data, the authors propose a mechanism for the transcriptional response of SGCs after an injury that is mediated via peroxisome proliferator-activated receptors (PPARs), and support this claim by rescuing the impairments in axon regeneration in their Fasn cKOs via PPAR activation. The work presented by the authors, both in vivo and in vitro, strongly support their overall conclusions and reveal an important contribution of SGC fatty acid synthesis pathway in axon regeneration after peripheral nerve injury. The manuscript is well written and the authors should be commended that they addressed both the limitations of specific techniques but also highlighted the importance of present findings, which are likely to lead to future areas of study. As it stands, the work is largely ready for publication, however, there are a few minor concerns/points listed below, which through addressing would strengthen the work further and also improve the clarity of text and figures.

We thank the reviewer for his/her thoughtful reading of our manuscript and the constructive and detailed suggestions. We have performed the suggested revisions that have increased the quality and depth of the paper.

Figures:

1. Figure 2D is difficult to interpret because there is no clear marker of the injury site for the reader to use as a reference for the proximal and distal site of injury.

We agree and have now provided a scheme to indicate how far from injury site picture is taken, and provided details in figure legend accordingly.

2. The legend for figure 3D does not include the size of the scale bar.

Thanks for catching this, the scale bar size is now indicated in the legend.

3. In figure 4A, the SGCs enveloping the DRG cell body in the Fasn ckO condition appears to be larger compared to the SGCs in the wildtype condition. Is this a common difference between

conditions or just the choice of this image? If there is a size change, quantification of SGC size may be necessary to determine whether or not Fasn deletion alters SGC morphology.

We agree and have performed the suggested quantification to determine whether Fasn deletion alters SGC morphology. Because the area of the SGC largely depends on where the section is taken, we quantified the nucleus area and nuclear circularity. We now report area as well as the ratio between x-y axis to reflect circularity (new Fig. 4b,c). Interestingly, we found that the SGC nuclear area is larger and more circular in the FasnCKO. It has been reported that under certain conditions, such as alterations in lipid composition within a cell, the overall nuclear structure can be modified (Walters et al, JCB,2012). A study in yeast also demonstrated that deletion of certain genes affecting lipid biosynthesis, leads to nuclear expansion (Walters et al, JCB,2012). We also measured the nuclear circularity in neurons and found no differences between control and FasnCKO (now Fig. 4d), suggesting that Fasn deletion in SGC does not affect neuronal morphology.

a. The authors should highlight which part of the top images in figure 4A was enlarged for closer observation in the bottom images.

We agree and this is now indicated in Fig. 4a

b. Is the magnification of both bottom images the same? The bottom right image appears to be of a higher magnification compared to the bottom left image.

We agree, that was corrected

4. Please provide sample sizes for figures 4C and 4D in the figure legends.

Sample sizes for Fig. 4c and 4d (new Figure 4f-i) are now provided in the figure legends. Axon diameter is now presented as a distribution histogram (new Fig. 4f). g-ratio is presented in new Fig. 4g.

5. There was no clear explanation of what exactly g-ratio is measuring in figures 4D and 4E.

An explanation was added to the text in result section. g-ratio demonstrates myelin thickness, measured as the ratio between the inner and outer diameter of the myelin sheath.

6. The representative images used for the FABP7 KO condition in Fig. S2A appears to be of a higher magnification compared to the control.

We thank the reviewer for catching this, the magnification was corrected

7. There appears to be more FABP7 positive fluorescence in the FABP7 KO condition compared to the Control condition in Fig. S2C.

We have quantified the Fabp7 and Tuj1 intensity in DRGs and nerves. These results are presented now in the new Fig. S2c, S2e.

Text:

1. Missing word: The SGC cluster identity ... SGC markers such (as) Kir4.1.

corrected in the text

2. One interpretation of our findings is that in SGC ... promote axon regeneration. (This statement needs to be rewritten for better clarity).

We have rewritten this statement which now reads as:

“Our findings thus suggest that in SGC, Fasn generates ligands for PPAR α following nerve injury and that Fasn dependent PPAR α activation contribute to promote axon regeneration..”
p. 19

3. Missing word: DRG sections from Fabp7KO and control mice were (immunostained) for and Tuji (red).

corrected

4. The figure S3 legend title should be bolded for consistency. corrected

Out of the scope of the current study but should be discussed in the manuscript as possibilities or future directions:

1) In figure 6, the authors make use of an *in vitro* system to screen for synthetic fatty acid compounds that could rescue the regeneration impairment they have found in their FASN conditional knockout animals. However, the authors do not test the endogenous fatty acid, palmitic acid, in its ability to rescue this phenotype. One advantage of this approach the authors mention is the specificity synthetic compounds provide for a particular pathway in SGC versus neurons, however, it would be helpful to have an idea of the dose-response to the endogenous PPAR ligand for the interpretation of certain aspects of the data namely, that low doses of certain compounds seem to be worsening the phenotype (GW7647 100 nM) while at higher doses they appear to be rescuing this phenotype.

We appreciate the reviewer's suggestion to test the ability of palmitic acid to rescue the FasnCKO phenotype, in addition to the synthetic PPAR activators. However, it is important to note that it is not palmitate *per se* that would be required for the rescue, but instead phospholipids. Indeed, the endogenous activators for PPAR α have been shown to be phospholipids, not free fatty acids (see for example Chakravarthy et al, 2009, Lodhi et al 2012). Phospholipid synthesis through the Kennedy pathway preferentially requires endogenous fatty acid synthesis by Fasn. In nearly all tissues in which Fasn was knocked out, the addition of palmitate in cultured cells from those tissues did not rescue the phenotypic effects of the knockout (Razani et al. JBC 2011;286:30949, Wei et al. JBC 2011;286:2933). This is not unexpected because in these models, the tissue specific knockout of Fasn in key organs controlling lipid metabolism such as liver does not cause a deficiency of circulating levels of palmitate (Chakravarthy MV Cell Metabolism 2005;1:309 and others). Indeed, palmitate is provided mostly by the diet. It is thus expected that Fasn deletion in SGC would have little to no effects on circulating fatty acid levels. Despite the availability of palmitate (since it moves freely across membranes), the FasnCKO animals display a phenotype, so adding back palmitate would be unlikely to impact the phenotype. Based on these considerations, we believe that testing if palmitic acid can rescue this phenotype is not highly relevant.

We have however revised the text in the results (p. 14) and discussion (p. 19) section to better reflect the difference between fatty acids and phospholipids in their ability to activate PPAR α .

We agree that a higher dose of GW7647 has a better effect on axon length, but at these doses there might be non-specific. At higher dose, clofibrate led to cell death. We just selected the agonist that was increased axon length at both lower and higher dose without causing cell death.

2) In figure 7, the investigators show that fenofibrate dietary supplement is sufficient to rescue the axon regeneration impairments they find in the FASN conditional knockout animals. While the authors point out that fenofibrate preferentially activates PPAR, which is enriched in SGC, they cannot rule out a potential role for PPAR signaling in other cell types during this process. The authors could perform the necessity experiment by conditionally knocking out PPAR in SGC and test whether this also leads to the same predicted impairment and test whether fenofibrate is still able to rescue this impairment. This would strengthen the case that this response is primarily mediated by SGC as opposed to other cell types.

We agree that in our *in vivo* experiments, we can't exclude the potential role for PPAR α signaling in other cell types. However, our single cell data strongly suggest that PPAR α is

enriched in SGC, whereas PPAR γ is enriched in other cell types, such as macrophages and endothelial cells (Fig. 6a). We agree that conditionally knocking out PPAR α in SGC and testing whether this also leads to the same impairments as knocking out Fasn would strengthen the case that PPAR α plays a role in SGC as opposed to other cell types. Because these experiments would require a substantial amount of time, and as indicated by the reviewer, would be out of the scope of the current study, we have edited our manuscript (on p.19) to indicate these experiments as possibilities or future directions.

Reviewer #2 (Remarks to the Author):

Avraham et al report on the requirement of fatty acid synthase in satellite glial cells to support the sensory axon regeneration. Through the single cell seq approach, they identified several genes enriched in SGC but not in Schwann cells. One of them, BLBP, was used to label SGC and BLBPCre-ER was used to manipulate SGCs. Further analysis showed that lipid metabolism genes were upregulated in SGC by sciatic nerve injury. In BLBPCre-ER;Fasn conditional KO mice, peripheral sensory axon regeneration was mildly suppressed after crush injury, especially for the fastest growing axons and at 2mm distal to the lesion. Consistently, in the isolated DRG culture, axon elongation induced by conditioning lesion was partially suppressed. Then, through a pharmacological approach in vitro and in vivo, they proposed that PPAR α may function downstream in the axon growth effect shown in Fasn KO mice.

It is always challenging to study cell-cell interaction. The hypothesis is of considerable interest, to propose the interaction between SGC and DRG neurons through single cell analysis. However, there are some deficiencies that reduce enthusiasm.

Major points

1. In the single cell profiling, DRG neurons were not adequately represented. Because there are quite a few reports on the single neuron seq by 10X, it will be important to demonstrate that the isolation protocol worked properly. Also, Fabp7 was the only gene that was verified through another approach. How about other genes?

We agree that there have been many reports using single cell approaches (10x, In-drop/drop-seq) to study neurons in the CNS. Most studies use a single nucleus rather than single cell approach. In the PNS, only one study (Sharma et al 2020) describes single cell RNAseq from DRG neurons. This study was focused on developmental time points. Embryonic and early adult tissue is known to be easier to dissociate. Our method achieved recovery of satellite glial cells, but might not be suitable for the analysis of neurons from DRG, given the low number of neurons recovered (1% rather than the 10% expected). We have not optimized our protocol further to obtain more neurons, since our goal was to analyze SGC.

Fabp7 is the novel gene we identified as specifically expressed in SGC. Fabp7 was shown to label glial cells during development (Britsch et al. 2001) and was used in a recent paper to immunostain SGC (Trevisan et al 2020). Our data combining single cell data and KO antibody validation clearly demonstrate that Fabp7 is a specific marker for adult SGC. The goal of our study was then to exploit this novel marker and perform functional studies using a conditional approach to unravel the role of SGC in nerve repair. We have not focused on the validation of other potential markers. However, we have revised the result section to cite other

SGC makers used in the literature and how specific they are based on our scRNAseq. The most widely used markers for SGC so far have been glutamine synthetase (GS), Kir4.1 CDH19 and SK3. Although by IF, GS appear specific (Fig S2b, and see also Jager et al 2020), we find that GS lacks specificity in our single cell data (Fig 1f,S1d). SK3 was also reported as a marker of SGC in rats, but we did not detect SK3 in our single cell data. We have revised our results (p. 6) and discussion (p. 17) section, as also suggested by reviewer #4 to discuss the use of GS to purify SGC in the recent Jager et al 2020 paper.

2. It is interesting to see that axon injury can regulate gene expression in SGC. How about other aspects, such as cell number? Do they proliferate or differentiate after injury? These are fundamental to understand the role of SGC in axon growth in response to injury.

We agree and have revised the results section in Figure 1 to discuss the cell number and cell proliferation. This is important given prior studies referring to increased SGC proliferation after nerve injury (for example Christie et al 2015). Our data was performed 3 days post nerve injury and we found some elevation in the number of SGC and Macrophages. However, the cell cycle markers ki67, and cdk1 were detected mainly in Macrophages and Blood cells/Monocytes but not in SGC 3 days post injury (Figure S1b). Furthermore, downregulated genes were enriched for the cell cycle pathway (now added to Table 3,4). This is in agreement with Jager et al, 2020 who showed cell proliferation in Macrophages but not SGC. Our results thus suggest that at 3 days post injury, there is no active proliferation in SGC and the increase in cell number could be a result of some minor differences in tissue dissection/dissociation between naïve and injured conditions. No differentiation markers such as Sox10, GFAP or MBP were significantly changed in SGC 3 days post injury (Table 3).

3. Although gene expression data were shown, it is still not clear what happened in SGC after injury. Which step of the fatty acid metabolism is regulated by injury, fatty acid synthesis, uptake or turnover? Why Fasn was chosen to study? It appears that Fasn is not upregulated after injury, which makes it unlikely to be the mediator of fatty acid metabolism change, if any. If fatty acid synthesis is the key to mediate the effect, fatty acid level should be directly measured in SGC, neurons or other cell types, and whole DRG in naive and injured conditions. Does Fasn KO change the fatty acid level in SGC? The same argument can be applied on FABP7, which could mediate fatty acid uptake.

We have revised the results section to better explain the genes underlying the enriched GO analysis. We now provide a table related to Fig 3a which indicates the genes that underlie each GO term (Table 4).

Although the expression of Fasn itself was not significantly regulated by injury, we focused on Fasn because both fatty acid synthesis and PPAR α signaling were enriched. Fasn controls the committed step in endogenous fatty acid synthesis, and its product is utilized for phospholipid synthesis through the Kennedy pathway. Phospholipids, but not free fatty acids, represent the endogenous activators for PPAR α (see for example Chakravarthy et al, 2009, Lodhi et al 2012). Fasn has been knocked out in multiple tissues. This intervention does not decrease circulating fatty acids. This is true even when Fasn is inactivated in liver, a major mediator of lipid metabolism (Chakravarthy MV Cell Metabolism 2005;1:309). This is not unexpected because palmitate is provided mostly by the diet. Fasn deletion in SGC would thus have little to no effects on circulating fatty acid levels. Accordingly, fatty acid mass spec in whole DRG may not be helpful, as there may be abundant palmitate derived from the diet in the DRG. It could be helpful in future experiments to perform mass spec analyses on phospholipids (total

PC and total PE). However, the phospholipids species that are ligands for PPAR are likely to be minor components given their role in signaling, so this approach would not necessarily prove that the change in any species is responsible for the effects on PPAR activation in SGC. Furthermore, because lipidomics requires abundant material, these could only be done on whole DRG making conclusions regarding cell specificity challenging. Based on these considerations, we believe that measuring fatty acid or phospholipid levels are beyond the scope of the current study, but they are planned for the future in the context of affinity capture experiments to enhance the likelihood of enriching for relevant lipids.

4. All the in-vivo and in-vitro experiments cannot exclude the role of schwann cell in axon regeneration. Fig2A indicates that FABP7 have no staining in schwann cells in DRG, but in Fig2B some Schwann cells and even some other cell types indeed have high expression of FABP7. It should be noticed that FABP7 is upregulated in schwann cells in nerve after injury. It is not clear how Tamoxifen was given. If tamoxifen diet was still given at the time of injury, it's still possible BLBPCreER may also have leakage expression in schwann cells after injury. Authors should also check FASN staining in the sciatic nerve in FASNcKO mice after injury.

We appreciate the reviewer concern about Fabp7 expression in Schwann cells and other cells. Regarding Fig. 2a and 2b, it is important to note that Fabp7 is a very highly expressed genes, to much higher levels than other genes in SGC. Whereas Fabp7 is highly expressed in all SGC, it is only expressed at a moderate level in a subset of other cell types including macrophages, endothelial cells and mesenchymal (Fig. S1d). We have revised the results section (p. 9) to better reflect the possibility of low Fabp7 expression in other cell types and the possibility that BLBPCreER may also have low expression in a subset of other cells.

We have also edited the text to better explain the timing of tamoxifen treatment: It now state on p.13: "Two weeks after tamoxifen treatment was completed, we performed a sciatic nerve crush injury"

We are aware of the concern related to Fabp7 expression in Schwann cells (Miller et al 2003). We have performed extensive experiments to show that the BLBPCre-ER mouse line can be used to label and manipulate SGC, with minimal impacts on Schwann cells at the 3 days post injury. Fabp7/BLBP is expressed 2 to 3 weeks after nerve injury, when Schwann cell process formation is exuberant (Miller et al 2003). We thus examined injured nerve, and found that 3 days post injury, only 4% and 7% of the nuclei expressed the GFP reporter proximal and distal to the injury site, respectively (Fig 2C,D). We also analyzed single cell data set from injured nerve (9 days post injury, from Carr et al 2019 and found that less than 5% of cells in the Schwann cell cluster express Fabp7 (Fig. S2f). Typically, 50% of nuclei express c-jun, a marker for Schwann cells response to injury, two days after nerve injury (Bloom et al 2014). The BLBP-creER may drive expression in a very small subset of Schwann after nerve injury, which are unlikely to represent a majority of repair Schwann cells.

We show in Figure 3D-F that Fasn expression is not altered in the nerve of FasnCKO mice. We followed the suggestion and have examined Fasn expression in control and FasnCKO, in naive and 3 days post injury, by western blot. The results are presented in new Figure 3e-f. In naive conditions, Fasn expression was reduced in the DRG, but not the nerve of FasnCKO. Nerve injury increased the protein level of Fasn in the DRG of FasnCKO but not control mice. Nerve injury also increased Fasn levels in the nerve of both control and FasnCKO. This could be due to an increase in Fasn levels in Schwann cells that has been reported in a recent RNAseq

analysis of the injured nerve (Shin et al 2018). These results indicate that Fasn can be efficiently deleted from SGC in the DRG, with minimal impact in Schwann cells.

5. It is very difficult to interpret the regeneration data shown in Fig 7B-D. What is the difference between the length of 10 longest axons (median at 3000um) vs. regeneration index at 3000um? The index difference was only seen at 2000um.

The crush site was determined according to highest SCG10 intensity along the nerve. We used two measurements to quantify axon regeneration. First, we measured the length of the 10 longest axons, which reflect the extent of axon elongation, regardless of the number of axon that regenerate. Second, we measured a regeneration index by normalizing the average SCG10 intensity at distances away from the crush site to the SCG10 intensity at the crush site.

The length of the 10 longest axons reflect only the extent of axon elongation, regardless of the number of axon that regenerate. We have now better explained this in the results section on p.13.

6. The interpretation of the Fenofibrate diet in vivo experiment could be many ways and it is premature to conclude that it is through SGC rather other other possibilities. A more specific regulation on SGC is required to draw a conclusion.

We appreciate the reviewer's concern. However, fenofibrate is a very specific agonist of PPAR α and does not affect PPAR γ or PPAR δ . Our single cell data show that PPAR α and PPAR α target genes are enriched in SGC, with minimal expression in other cells. However, we agree that the fenofibrate could rescue the FasnKO defects in axon generation by impacting other cell types. As suggested by reviewer #1, testing if conditional deletion of PPAR α in SGC mimics Fasn deletion and testing if the defects can or not be rescued by fenofibrate would strengthen the case that fenofibrate acts primarily on SGC. However, and as suggested by reviewer #1, these experiments are beyond the scope of the current paper. We have revised the results (p.16) and discussion (p.19) section to better reflect that fenofibrate might act on other cell types beyond SGC.

Minor points

Authors used Tuj1 staining in all the DRG sections. To validate the expression pattern of FABP7 or FASN in different scenarios, it would be better to use specific SGC and schwann cell markers for co-staining.

We appreciate the reviewer suggestion, but our data suggest that many Schwann cell markers are also expressed in SGC (Figure 1g and S1e). We have attempted staining of DRG sections with PRX and NCMAP, which appear to be more Schwann cells specific, but these antibodies have not been working in our hand. There has been so far, to our knowledge, no specific SGC marker validated in a KO mouse as we did here for Fabp7. Most markers are validated by the localization of the signal surrounding Tuj1 positive neurons. GS has been the most widely used, but our single cell data indicate that at the gene expression level it is not very specific. We believe that co-staining with a neuronal marker to reflect localization around neurons, coupled with validation in a KO is the best approach to validate Fabp7 and Fasn expression in SGC.

Before analyzing transcription change in SGC, it's important to quantify the number of SGC or other cell types in DRG before and after injury.

We agree and as stated in our response to major point #2 we have revised the Figure 1 and results section to present an analysis of cell numbers and cell proliferation markers in naive and injured conditions.

In Fig7E, to draw the conclusion “fenofibrate rescues the impaired axon regeneration in FasncKO mice”, a complete group of WT naïve, WT injured, WT naive+fenofibrate, WT injured+fenofibrate should be included into the experiment.

We agree and now present this complete experiment in Figure 7f-h.

Reviewer #3 (Remarks to the Author):

This manuscript from Cavalli and colleagues reports on work examining the biology of satellite glial cells of the dorsal root ganglion. The work is highly novel, reporting a number of new characteristics of the SGCs, including a transcriptional profile providing a set of SGC-specific genes, their intimate relationship with sensory neurons (possibly constituting a functional unit), responsiveness to nerve injury (which is a stimulus that is remote to them), and the potential of a regulatory role in the sensory neuron response to the injury, including at least some aspects of axonal regeneration. It is largely a very well-controlled and designed study with strong quality control on reagents and on genetic manipulations. The work provides specific mechanisms where only general suggestions were available previously, and promises to be welcomed by the field as a cornerstone for future reference. There are a few points that require attention to ensure the reality meets the potential. It is likely that no new data is required (though the authors may wish to add), but simply revision in the text to enhance clarity and credibility, particularly regarding axonal regeneration. There is sufficient over-interpretation of some data to warrant revision. The most significant single point appears to be the treatment of the data related to regeneration, and the framing of the manuscript. Reading the Title, Summary, Abstract, and Introduction would lead one to believe that the manuscript is largely concerning the regulation of axonal regeneration by SGC-derived fatty acids. But this is not the actual major content of the manuscript, and the data supporting that message are not the strongest in the manuscript. The manuscript appears to base many the strongest claims on the effects on expression of ATF3, but the assessment of ATF3 expression is not sufficient to support those claims. It bases other strong claims on assessments of an unspecified very small group of axons/neurons and uses those observations to extend the claims to the entire population. The claims may yet be true, but they are not sufficiently-supported. The authors need to revise some of the claims and should consider re-framing the manuscript to center on the strongest and greatest-volume of the data – SGC-neuron structural relationship and SGC biology – OR they can enhance the data related to regeneration to bring those data to the point where they can adequately support the focus of the message on axonal regeneration.

We thank the reviewer for his careful and thoughtful reading of our manuscript and his constructive and detailed suggestions. We have performed the suggested revisions in the text to enhance clarity and credibility, avoid over interpretation. We believe these revisions have increased the quality and depth of the paper.

Introduction

“In adult animals, multiple SGC form an envelope that completely enwraps each sensory neuron 5-7. The number of SGC surrounding sensory neurons increases with increasing soma size in mammals 8, 9. The neuron and its surrounding SGC thus form a distinct morphological and functional unit”

- The statement “The neuron and its surrounding SGC thus form a distinct morphological and functional unit” seems a bit premature in this place in the manuscript. The statements preceding this in the Introduction only suggest the possibility of a functional unit (correlative).

Although the data generated and analyzed in this report somewhat support the claim of a functional unit, those data have not been presented yet in the manuscript. The claim at the end of the Introduction (“These results also highlight that the neuron and its surrounding glial coat form a functional unit that orchestrates nerve repair”) is much more appropriate as it follows from a summary of the data to be presented. (However, there remains a concern that those data may not support that conclusion, as outlined below)

We agree that in this part of the introduction, we can simply refer to SGC and neuron forming morphological units.

“Structural neuron-glia units, similar to these, do not exist in the central nervous system”

This could be debated. The authors may wish to provide some more description to clarify their point and ensure credibility.

We have provided references to key papers studying the morphology of SGC neuron to support this claim.

“These studies show that communication between neuron and SGC is critical for neuronal excitability and nociception...”

It is not clear that those cited studies indicated a relationship that is “critical”. The author’s own language elsewhere implies a less-direct relationship (“modulate” as opposed to “regulate” or “is required for”).

We agree and have edited this sentence and revised the entire paragraph.

With the exception of a description of SGCs, the overall design of the Introduction narrative lacks a relationship to the major experimental manipulations and conclusions. The major GOF/LOF manipulations focus on SGC Fasn and PPAR-alpha and their role in modulating regenerative axon growth, but there is nothing in the introduction that speaks to anything that is known about these systems in axonal regeneration, even though there are relevant data.

Describing those data will provide better context and does not impact the novelty or impact of the current work. These changes are only suggested.

We agree and have added a paragraph in the introduction that focuses on Fasn and PPAR α and their role in modulating axon growth.

RESULTS and METHODS

“We anticipated obtaining a larger representation of neurons in our dataset, but their large cell diameter (up to 50 micron for proprioceptors), is likely limiting them from passing through the microfluidic pipes in the Chromium 10X device.”

- It is not clear if the speculation is that all neurons are impeded through the microfluidics, or only the large diameter. There should be data available from the company or core facility regarding the tolerance of the microfluidics. The vast majority of neurons are small and medium, so the implication that the small proportion of neurons that are 40-50um might account for the unexpectedly low recovery of neurons is not entirely logical.

- It is also possible that the large cells died, as they are more susceptible to mechanical dissociation.

- Examination of genes specific for, or enriched in, different populations of sensory neuron types, could be useful to determine if the recovery/loss was selective. Ultimately, however, it is not clear why this speculation is included in the Results. It may be more suitable for the Discussion.

We have revised the result section according to reviewer #2 to include cell numbers . We also have include in Figure S1c an analysis of the sensory neuron subtypes recovered. The majority of neurons (58%) were small diameter (nociceptors, TrkA/CGRP). We have revised the results section accordingly indicating the distribution of neuronal populations in our scRNAseq (p. 6).

The manuscript must enhance its description of the bioinformatic and gene-expression analyses. The existing description is terse and not sufficient for reviewers to determine suitability or to provide utility for readers. For example, the description: "...using the Chromium Single Cell Gene Expression Solution (10x Genomics) (Figure 1A). Graph-based clustering, using Partek flow analysis package..." is not at all sufficient for assessment or reproducibility. For further example: "We thus examined the transcriptional similarity between astrocytes and SGC by comparing the top expressed genes in each cell type, using our scRNAseq data for SGC and a previously published transcriptional analysis of astrocytes (500 genes). We found that SGC share about 10% of genes with astrocytes..."

- This description (not supplemented by anything useful in the Methods) does not describe how "top expressed genes" were determined, or what cut-offs were used to determine expression levels for comparing between cell types.

We agree and have revised both results (p. 5) and method section (p. 32) to provide a better description of single cell analysis, gene-expression analyses and filtering criteria of cells.

"We next compared the top expressed genes in SGC (605 genes) and Schwann cells (572 genes), which revealed that SGC share only 2% of genes with Schwann cells in the DRG"

- Correct to "...share only 2% of THOSE gene TRANSCRIPTS with..."

Corrected

- Again, not clear what criteria establishes the "top expressed" genes.

The section of top expressed genes is now explained in detail on p.7

The manuscript includes excellent data demonstrating that the Fabp7 antibody does not bind elsewhere – examined in KO mouse tissue – but it is not clear that there are data showing it does bind to Fabp7 (although it does bind to the expected cells). Clarification of this point would strengthen the manuscript.

Validation of antibodies in KO cell lines or KO animals is the gold standard and this is the procedure recommend by NIH guidelines on for authentication of reagents. We believe that the location of the Fabp7 staining, observed by us as well as others (Trevisan et al 2020) combined with the validation in a genetic null mouse and our single cell data is sufficient to demonstrate that Fabp7 binds the target protein in SGC. We believe this point was clearly addressed in results section, p.8.

“We observed no abnormalities in the morphology of the SGC surrounding the neuron somas in the FasnKO animals compared to controls...”

- Unless there was a quantitative assessment of some set of criteria, this should be corrected to read “We observed no QUALITATIVE abnormalities...”. If a quantitative analysis was performed it should be described.

We agree and have performed a quantification of the SGC nucleus area and circularity, which is now presented in Fig. 4b,c. As described above for reviewer #1, the area of the SGC largely depends on where the section is taken with respect of the 3D arrangement of the SGC coat around the neuron. We thus quantified the nucleus area and now report area as well as the ratio between x-y axis to reflect circularity.

[The examination of ATF3 was a strength of the project – it could have revealed stress on neurons or other cells (SGC) related to the KO. Unfortunately, the strength was not fully-realized because only the mRNA from homogenized DRG was examined, so cell-type specificity is absent.]

We agree that examination of ATF3 as a stress marker is important. We considered dissociating neurons in control vs WT DRG to assess ATF3 levels. However, the dissociation and culture process are in itself a stress and would not allow us to make strong conclusion (see Wangzhou et al 2020, BioRxiv). However, in section of whole DRG, most of the ATF3 staining is in neurons (now presented in new Figure S6A) and also well document by prior studies that use ATF3 expression as neuronal injury reporter (Holland et al 2019). We therefore think that qPCR for ATF3 from whole DRG may represent relatively well ATF3 expression in neurons.

We now also provide additional data in Fig. 7b that shows that mRNA levels of ATF3 as well as Gap43 are downregulated in FasnKO, and rescued by fenofibrate treatment (Fig. 6b and S6b).

Using cleaved-caspase as an indicator of cell death assumes only a single mechanism of death (and stereological counts were not performed, which could have revealed a difference in overall cell number, which could have indicated cell death via some other mechanism(s)). This is largely acceptable because it represents the mechanism known to exist in WT animals, but it should be acknowledged in the Results or Discussion that other forms of cell death exist and were not assessed.

We appreciate the suggestion and have edited the results section to reflect that other types of cells death can't be excluded on p.11.

The reduced ATF3 expression in the injury condition was assessed by qPCR and the interpretation of the data appear to assume that the DRG neurons are the only source of post-injury ATF3. Regardless, the difference could be due to decreased expression per cell or fewer cells expressing at the same level/cell as control. Also, homogenized DRG includes SGC, which might (do?) express ATF3 after nerve injury. The reduced ATF3 mRNA could be related in some way to the loss of Fasn in SGC, as implied in the text. However, the text currently implies the reduced ATF3 is only in neurons and so the effect of Fasn-KO in SGC is somehow trans-cellular, but this is not necessarily true. This possibility can be dealt with in two ways – either the interpretation can be expanded to allow for these possibilities, or new data can be collected to assess ATF3 expression in terms of number and type of cell, as well as magnitude per cell (if necessary).

We appreciate the suggestion and have revised the text to better account for the different

possibilities on p. 16 . As stated above, we also added new data to support the fact that the ATF3 mRNA increase after nerve injury is largely neuronal. We also provide a new data set showing that fenofibrate treatment rescues the decreased ATF3 expression after injury in FasnCKO.

Average axon diameter (Figure 4C) may not be the best measure, because there is such a wide range of sizes with very different numbers giving rise to the different ends of the spectrum. A cumulative sum histogram (or similar) would be far more representative of reality.

We agree and have modified the graph to a frequency histogram, now present in Fig. 4f.

“The robust bidirectional communication between neurons and SGC is critical for neuronal excitability and nociception and the excitability of sensory neurons is controlled in part by the surrounding SGC”

- As with the comment regarding a similar claim in the Introduction, please ensure that the terms “critical” and “controlled” are accurate representations of what the cited articles show.

We agree and have revised the text accordingly

“Whole-cell recordings were performed in acutely dissociated cocultures of DRG neurons and glia”

- Electrophysiological recordings starting 24h after plating does not qualify as recording from “acutely dissociated” DRG neurons. Acutely-dissociated neurons from naïve animals rarely fire multiple APs to the depolarizing stimulus applied, but neurons from cultured DRG neurons or from injured DRG neurons (acute-dissociated or cultured) can show repetitive firing, as is shown in Figure 4L. The control and KO are similar, so this difference is likely not important in terms of the effects of the KO, but it matters for 1) credibility and 2) to provide proper context for readers. Please correct any instances to “short-term culture” or a similar term, not “acutely-dissociated” (observed at least in Results and Methods).

We agree and have revised the text accordingly, referring to this culture as “short-term culture”

“Therefore, Fasn deletion in SGC does not affect functional properties of DRG neurons IN THE NAÏVE DISSOCIATED CONDITION.”

We agree and have revised the text accordingly

Regarding the effect of Fasn-cKO on axonal regeneration and the priming-effect:

- The in vitro work used DRG taken from E13.5 mice, which were maintained in vitro for 7d before additional injury (achieving a relative “age” of E20.5), and fixed 24h later (achieving a relative “age” of E21.5/P0). This time is largely considered still developmental for sensory neurons, in which injury is more an effect on the developmental program (outgrowth and survival) than on the adult injury/stress response, which can include axonal regeneration. The data can still have value, of course, but perhaps should not be labeled as regeneration per se, especially when other aspects of the project are examining true regeneration. Making a terminological equivalence between these models suggests a mechanistic one, which is not a given, and certainly not supported in the narrative. Unless the authors provide some rationale in the text (not just in response to the reviewer) to use the same terms for these different models, the Embryonic injury and growth model should be given

another term, perhaps “developmental axonal growth”, “developmental axonal response to injury”, or something similar.

The eDRG culture model has been used extensively by us and others to study axon regeneration. While we agree that these are embryonic neurons with high growth capacity, our findings in eDRG have been recapitulated in large part in adult *in vivo* system (Cho et al, 2012, 2014, 2015). Our unpublished RNAseq analyses of such eDRG culture also suggest that *in vitro* axotomy recapitulates expression of the expected regeneration associated genes and downregulation of genes related to ion channels observed *in vivo* (Lisi et al 2017).

“For adult DRG cultures DRG were dissected from naive mice.”

- Provide the age/sex of the mice.

Age and sex is now provided in method section

“Contralateral nerve and DRG served as uninjured controls, when needed.”

- Please indicate if the contralateral side was sham (and if so if it was skin incision with/without muscle dissection) or intact.

The contralateral is the uninjured side, not a sham surgery. Details is now provided in method section

The model uses a 3d post-crush time for regeneration assessment. This seems to offer a limited picture. It does not allow for assessing the normal range of “initiation rates” known to exist across the different types of peripheral axon types. It limits this assessment to only the fastest responders, but the claims are made broadly. It is entirely possible that the neurons which are slower-initiators in the WT have an identical initiation rate in the KO, but this possibility is not addressed.

3d post crush allows us to relate our finding to most of ours and others’ previous findings and is a time at which defects can be observed. Shorter time, such as 1d post crush, does not allow to see differences in KO models such as DLKKO , HIF1a cKO that affect activation of a pro-regenerative program (see Cho et al 2015 and Shin et al 2012). 3 days allows the regeneration program to be turned on and if impaired, defects can be observed. We also previously showed that nociceptor growth is similar to other neuron types growth (Carlin et al 2019, Abe et al 2010) Examination of other time points and neuronal subtypes is beyond the scope of the current study. The possibility that different neuronal subtypes are differentially affected by PPAR α signaling SGC remains is now been mentioned in the discussion, p. 18.

The images of the nerve in the figure imply a rather significant qualitative difference in the architecture of the distal nerve, with the KO nerve looking much closer to normal (with visible undulations), and even with what appear to be SCG10+ structures far distal to the injury site. This should be addressed better. Providing images of axonal structure (beta-3 tubulin?) would significantly enhance the context for readers.

We have added images of nerve sections stained for Tuj1 from control and FasnckKO in figure S4a, showing that the overall architecture of the nerve is similar.

Arrow heads have been added to Fig. 5a and 7c to emphasis the longest regenerating axons.

“...the injury site was defined as the area with maximal SCG10 intensity.”

- Please indicate if this was established from a single section, or a consensus across many

sections. Also indicate what qualified as “area with maximal SCG10” – a single pixel? 10 contiguous pixels? Etc. This method of defining the injury site may have been convenient, but it is not (at least as presented here) terribly clear or strong. Using markers of denervated Schwann cells or infiltrating cells, even as a proof of concept for the SCG10-intensity method, would have strengthened the assessment significantly. This point of clarity matters greatly as it is the basis for the quantitative measures.

This is an assay that we and others have used extensively. The crush site is defined as the site along the nerve length where SCG10 intensity is maximal when measured in a vertical line was across the nerve. The quantification assay is explained in details in methods, p. 34. SCG10 was chosen, because it selectively labels regenerating axons and displays higher specificity than GAP43 or YFP in the early stage of axon regeneration after nerve crush (Shin et al 2013).

- Certainly there are differences detected, but the 3 day post-injury time point seems sub-optimal for the claims made. It is at the leading edge of what is reported for the latency to the cell-body reaction and start of axonal regeneration, which would fit well if the only goal were to assess the fastest response-initiation. Although that is included, the manuscript also uses these data to support claims regarding the regeneration process in general. Assessments at later time points would have greatly strengthened the claims. The claims throughout the manuscript regarding regeneration should be revised to reflect this point.

We have now included in the discussion that whether SGC contribute to sustain regenerative growth until target reinnervation remains to be determined (p. 18).

“Neurons in injured FasnKO displayed reduced neurite length compared to injured controls, but similar initiation rates (Figure 5D-F).”

- The outcome measure relies on the “longest 10 axons”. There is nothing intrinsically flawed in this, but it does introduce a bias toward the most effective and/or fastest-starting axons. This is problematic because these may be a subpopulation, or the effect of Fasn/PPAR may only be on the

Figure 5D-F is in vitro growth, and the measure is axon length per neuron, not longest 10 axons

- It is possible that the “initiation rate” was unaffected by the Fasn-KO as claimed, but there are other interpretations that could exist (false-negative). As described, the “initiation rate” was determined by the post-plating time at which neurites emerged to some qualifying characteristic. This was done without regard to the type of sensory neuron giving rise to those neurites. Unless all SGC express the same level of Fasn, and all DRG neurons express the same level of PPAR, then it is possible that the neurons that are responsible for providing the “equivalent initiation rate” are a subpopulation that, in WT, have some reduced/different PPAR-expression/function and/or are surrounded by SGC with lower Fasn (Figure 3 appears to indicate that there may be some significant heterogeneity in the Fasn expression). In this case, the loss of Fasn (in the KO) would largely be irrelevant as that subpopulation did not have a high degree of SGC-Fasn and neuronal PPAR anyway,

and the conclusion could be incorrect. This can be addressed either by providing data regarding the uniformity of the PPAR-expression across neuron types AND the uniformity of Fasn-expression across SGC, OR the assessment of “initiation rate” can be done with sensitivity to neuronal sub-types. Alternatively, the manuscript can alter the claim and provide better presentation of the possible interpretations of the data.

We have revised the text and rather than initiation rate, refer to percent of neuron initiating axon growth in Fig. 5f and 7h. Whether SGC differently affect neuronal subtypes is now discussed (p.18). The uniformity of Fasn-expression across SGC might be interesting. However, without a morphological correlation of SGC surrounding a given neuron, this information might not explain the effect of Fasn/PPAR expression in different SGC on different subtypes. Whether different SGC subtypes surround different neurons subtypes is a major ongoing study in the lab, but is beyond the scope of the current manuscript.

- Using the in vivo outcome to determine “initiation” with the in vivo outcome of “length” as was done leads a particular problem. There is no means for determining if the populations are the same. This again is not necessarily a flaw, except that the narrative regarding both outcomes implies both a uniformity across the entire population of neurons, and implies a union of the population of axons that are longest and first to grow in vitro. These implications are not supported by the experimental design, though they may be true. Therefore, either new data must be provided demonstrating the uniformity and union, or the narrative must be adjusted to clearly provide to the reader an interpretation of the data that includes these possibilities and limits.

We agree that which population of neurons is examined in the in vitro and in vivo assays is not determined in our current experiments. We have added a comment to this effect in the discussion on p.18. As stated above, whether SGC affect neurons subtypes differently is an ongoing study in the lab, but is beyond the scope of the current manuscript.

The outcome measure of “initiation rate” may have additional issues. It appears to be less about the latency of when the range of axons start to regenerate and more about the proportion of neurons that display neurite-outgrowth at a given time. It is not clear that “rate” is a suitable term in the in vitro case. Perhaps “proportion” or similar term would be more accurate.

We agree and have revised the text and rather than initiation rate, refer to percent of neuron initiating axon growth in Figure 5f and 7h.

“These results indicate that Fasn in SGC is required for the conditioning effect, and specifically for the elongating phase of axon growth”

- The conclusion of “required” is not supported. Fasn may influence the priming effect, but even this may not be well-supported. Fasn may regulate, to some unknown degree, the priming effect in some population of neurons. However, the limited assessment time does not lend itself to a thorough assessment of priming or regeneration, particularly the elongation phase.

We agree and have use “contribute”

“...SCG10 intensity was quantified at 1, 2, and 3mm from the injury site as previously described”

- A brief description needs to be provided. It is not clear if the quantification was the average intensity of a row/line of single pixels (e.g., the demarcation line), or some area of pixels.

Details on quantification are now provide in the method section

“For both LENGTH AND INTENSITY quantifications, five sections per biological replicate were averaged.”

We made these changes

In standard situation, PPAR-g is neuronal and PPAR-a is SGC. However, it appears that this pattern was not ensured in the Fasn-cKO. This weakens the conclusion (assumption?) that the fenofibrate (PPAR-a-agonist) is working only on SGC, especially since the effective concentration of fenofibrate was not determined. This weakness is mitigated by the data that fenofibrate does not improve axon regeneration in neuronal cultures that do not contain SGC. The authors should consider including a statement to clarify the context, or doing some direct assessments.

We have included statement in the results and discussion acknowledging the fact that fenofibrate might act on other cells beyond SGC (p.16 and p.19)

DISCUSSION

“The role of SGC in nerve regeneration has been largely ignored...”

The term “ignored” is inaccurate. It implies that the role was known but not considered. But the role was not really known until this work – this is even stated in the following sentence. Please reword.

We agree and have revised the text accordingly

“Although addressing this question will require further investigation, our data indicate that Fasn expression in SGC is required for the proper expression of at least one regeneration associated gene after injury, Atf3”

- The conclusion of “required” is not supported.

We agree and have revised the text accordingly

“Our study suggests that neuronal “intrinsic” response to injury may not be purely neuronal but rather includes the SGC surrounding the neuronal soma...”

- Unless this statement is referring to something other than the neuronal ATF3 response (which the authors have not specifically identified) as the “intrinsic” response, then the statement is not supported.

We agree and have revised the text accordingly

METHODS

“...and both injured dorsal root ganglia and sciatic nerve were dissected at the indicated time post-surgery”

- Specify which DRGs were designated as “injured” and collected. The designated DRG’s were described in figure legends and main text of the original manuscript. This information is now also added to the methods section.

Specify the control procedures for immunohistochemistry and western blot.

The details for control procedures were added

“Cultures were then used for electrophysiological recording 24h after plating or fixed and stained with the indicated antibody”

- Electrophysiological recordings starting 24h after plating does not qualify as recording from “acutely dissociated” DRG neurons, as indicated in the Results.

Agreed and changed to “short-term cultures”

Figure 4H shows results from qPCR of ATF3, but there are no primer sequences provided for ATF3. Please provide the primer information.

ATF3 primers sequence were added

“An Automated analysis for axon tracing and neurons soma count was used for ex-vivo adult DRG culture experiments (Nikon elements).”

- Please provide specifics for the “automated analysis”.

Nikon elements is a commercial software package for image analysis using an algorithm for automatic digital reconstruction of axons. The specific analysis code is available upon request (requires NIS-Elements and General Analysis).

“Statistics was performed using GraphPad (Prism8) for either Student’s t-test or ANOVA analysis.”

- This is not sufficient. Please ensure that this section is complete and that the description of the statistics in Methods and elsewhere are consistent and clear.

- Indicate type of ANOVA

- Perhaps the authors have omitted mention of the post-ANOVA tests in the description in the Methods. Figure 2 legend mentions the Sidak multiple comparison test, suggesting this terse description in the Methods is in error. Sidak or Dunnett tests were used as part of one- and two-way ANOVA for multiple comparison tests.

A description of the statistical tests was added

- Gene expression mentioned use of FDR, but this is not described. An FDR value is a p-value adjusted for multiple tests (by the Benjamini-Hochberg procedure). It stands for the “false discovery rate” it corrects for multiple testing by giving the proportion of tests above threshold alpha that will be false positives (i.e., detected when the null hypothesis is true). A description was added.

- Most legends indicate which statistical tests were used, but not all. Figure 4 legend has p-values but no indication of the tests used

A description of the statistical tests was added to Fig. 4 legend

Minor

“Structural neuron-glia units, similar to these, do not exist in the central nervous system”

Commas likely not necessary

Commas were removed

Throughout, please ensure that the language consistently and properly uses the terms “gene” and “transcript”. In many places the term “gene” is used where “transcript” should be used.

The Chromium Single-Cell 3’ Solution we used for the scRNASeq can recognize a general gene but not a unique transcript or splicing variants, therefore we chose to use the term ‘gene expression’ and not ‘transcript expression’ to avoid misleading the reader.

“Our single cell RNAseq analysis using freshly dissociated tissue thus UNRAVELS a unique transcriptional profile of SGC in DRG...”

corrected

“We found that SGC share about 10% of THOSE gene TRANSCRIPTS with astrocytes, among them Fabp7...”

corrected

“We crossed BLBPCre-ER to the Rosa26-fs-TRAP and observed that following 10 days OF tamoxifen treatment...”

OR

“We crossed BLBPCre-ER to the Rosa26-fs-TRAP and observed that following A 10 day tamoxifen treatment...”

corrected

“Pathway analysis of other major CELL types in the DRG confirmed fatty acid metabolism as a unique pathway enriched in SGC...”

corrected

“...our pathway analysis INDICATED that the cell cycle term is enriched in macrophages, but not SGC...”

Also, this sentence is awkward. The “cell cycle term” is not enriched in macrophages. Please reword.

corrected

“...loss of Fasn in SGC does not lead to morphological deficits in the DRG and does not ELICIT a stress response in neurons”

corrected

“Fenofibrate is used clinically to treat lipid disorders, but has been unexpectedly been shown in clinical trials...”

- “been” is redundant and grammatically incorrect

corrected

“During surgery, 8-12 week old C57Bl/6 mice OF the indicate genotype were anesthetized”

Corrected

Review provided by Jeffrey C. Petruska, Ph.D.

Reviewer #4 (Remarks to the Author):

Overview of the study:

In their manuscript titled Fatty acid synthesis in satellite glial cell promotes regenerative growth in sensory neurons, Avraham et al. have first used scRNA-seq to study the non-neuronal cell populations of mouse dorsal root ganglia in both naïve and injured conditions, focusing on the satellite glial cells (SGC) that surround all sensory neurons. They show that SCGs can be distinguished from Schwann cells by their gene expression profiles and define Fatty acid binding protein 7 (Fabp7) as a specific marker for SGCs in the peripheral nervous system. The authors further show evidence that crush nerve injury upregulates fatty acid metabolism in injured DRGs. Using a SGC-specific knock-out of the Fatty acid synthase gene (Fasn) together with TEM, injury/apoptosis markers, and electrophysiology, they show that the deleting this gene in SCGs has little impact on the properties of uninjured DRG neurons, but the regeneration abilities of these neurons appear compromised. Avraham and colleagues then show that this deficit in regeneration ability can be rescued in a neuron-independent fashion by feeding in vitro embryonic cells or intact adult animals with a specific

PPAR α agonist, Fenofibrate. Based on their results, the authors suggest that fatty acid synthesis and PPAR α -signaling in SGCs are fundamental to axon regeneration in adult peripheral nerves and highlight the functional connectivity of the sensory neuron and its glial coat.

General remarks:

This is an important study for the field of sensory neurobiology, with novel results underscoring the role of satellite glial cells in peripheral nerve physiology. The study is of high quality, well written, and based on a solid set of experiments. Results from the laboratory experiments are clear and give strong support for the claims made by the authors. The scRNA-seq part is straight-forward and the results appear to be sound; however, to be considered for publishing, I invite the authors to address the concerns I have listed below.

We thank the reviewer for the thoughtful and constructive suggestions. We have performed the suggested revisions that have increased the quality and depth of the paper.

Issue 1:

While the scRNA-seq analysis appears appropriate, the authors have not disclosed any information on how this analysis was done, apart from mentioning that a proprietary software package (Partek Flow) was used for the analysis. Therefore, for a proper evaluation of the analysis, the information below needs to be added to the paper in a way that the analysis can be reproduced.

How was the data quality controlled? What algorithms were used for the analysis?
What were the most important parameters chosen for each critical step of the analysis? What statistical testing was used for differential expression analysis between clusters?
How were the comparisons between datasets done (comparisons to neurons and astrocytes)?
How was the KEGG analysis done?

We agree and have heavily revised the method section and added information to better disclosed how analysis was done. QC, filtering criteria and statistical testing information have been added to the methods section.

Issue 2:

The authors should add a small section to the discussion where they compare their results to those in this recent paper dealing with satellite glia after injury.

Glia. 2020 Feb 11. doi: 10.1002/glia.23785.

Changes in the transcriptional fingerprint of satellite glial cells following peripheral nerve injury. We totally agree, and have added two paragraphs in the discussion section to discuss our results in the context of this recent paper.

Minor issues:

Please increase font size in Figures 1D and S1B (currently unreadable) and inform in the figure legend (Figure S1B) what the x-axis shows.

We have revised Fig.1 1 and S1 and ensured that fonts are readable and axis are labeled

Please add to the GEO database a raw expression matrix of the data with cell-cluster and cell-replicate annotation. This will help other researchers to use the data more easily.

The raw matrix was added to the GEO data base. To help other researchers use the data, we are planning to deposit our data in an open-source project dedicated to processing and visualization of single-cell RNA-seq data. This tool is currently being develop by a colleague at Washington University and will be available soon.

Figure 3D says "Scale bar: xx μm "
corrected

Reviewers' Comments:

Reviewer #1:

Remarks to the Author:

The authors were responsive to reviewers' comments and my concerns are fully resolved after revisions.

Reviewer #2:

Remarks to the Author:

More direct evidence is needed to support the statement that "These results identify fatty acid synthesis in SGC as a fundamental novel mechanism mediating axon regeneration in adult peripheral nerves".

1. Lipidomics or other metabolic measure is actually crucial to support the claim on fatty acid synthesis or lipid metabolism. If the data cannot be provided due to technical difficulty, the conclusion should be tuned down.
2. Fig 7a did not sufficiently support the claim "Activation of PPAR α in SGC" because the whole DRGs were used to do RT-PCR. PPAR α immunostaining in DRG w and w/o injury, and w and w/o fenofibrate diet should help to clarify and differentiate the PPAR α expression in SGC and neurons. PPAR α is within the scope of the manuscript because it is the claimed mechanism to support the conclusion.
3. Direct manipulation of PPAR α in SGC or DRG neurons is still important to address the concern. It is actually not a very difficult experiment. One possibility is to knockdown PPAR α in DRG neurons using virus and add fenofibrate.
4. Fig 5A, 7C are not representative to the quantification of 5C, 7E. At 1000um, axons from KO are much fewer than control.

Reviewer #3:

Remarks to the Author:

** Reviewer comments on revision in RED and with **

This manuscript from Cavalli and colleagues reports on work examining the biology of satellite glial cells of the dorsal root ganglion. The work is highly novel, reporting a number of new characteristics of the SGCs, including a transcriptional profile providing a set of SGC-specific genes, their intimate relationship with sensory neurons (possibly constituting a functional unit), responsiveness to nerve injury (which is a stimulus that is remote to them), and the potential of a regulatory role in the sensory neuron response to the injury, including at least some aspects of axonal regeneration. It is largely a very well-controlled and designed study with strong quality control on reagents and on genetic manipulations. The work provides specific mechanisms where only general suggestions were available previously, and promises to be welcomed by the field as a cornerstone for future reference. There are a few points that require attention to ensure the reality meets the potential. It is likely that no new data is required (though the authors may wish to add), but simply revision in the text to enhance clarity and credibility, particularly regarding axonal regeneration. There is sufficient over-interpretation of some data to warrant revision. The most significant single point appears to be the treatment of the data related to regeneration, and the framing of the manuscript. Reading the Title, Summary, Abstract, and Introduction would lead one to believe that the manuscript is largely concerning the regulation of axonal regeneration by SGC-derived fatty acids. But this is not the actual major content of the manuscript, and the data

supporting that message are not the strongest in the manuscript. The manuscript appears to base many the strongest claims on the effects on expression of ATF3, but the assessment of ATF3 expression is not sufficient to support those claims. It bases other strong claims on assessments of an unspecified very small group of axons/neurons and uses those observations to extend the claims to the entire population. The claims may yet be true, but they are not sufficiently-supported. The authors need to revise some of the claims and should consider re-framing the manuscript to center on the strongest and greatest-volume of the data – SGC-neuron structural relationship and SGC biology – OR they can enhance the data related to regeneration to bring those data to the point where they can adequately support the focus of the message on axonal regeneration.

We thank the reviewer for his careful and thoughtful reading of our manuscript and his constructive and detailed suggestions. We have performed the suggested revisions in the text to enhance clarity and credibility, avoid over interpretation. We believe these revisions have increased the quality and depth of the paper.

**The revisions have been very thorough, and only a few items remain to be addressed. I look forward to seeing this work published.

Introduction

“In adult animals, multiple SGC form an envelope that completely enwraps each sensory neuron 5-7. The number of SGC surrounding sensory neurons increases with increasing soma size in mammals 8, 9. The neuron and its surrounding SGC thus form a distinct morphological and functional unit”

- The statement “The neuron and its surrounding SGC thus form a distinct morphological and functional unit” seems a bit premature in this place in the manuscript. The statements preceding this in the Introduction only suggest the possibility of a functional unit (correlative).

Although the data generated and analyzed in this report somewhat support the claim of a functional unit, those data have not been presented yet in the manuscript. The claim at the end of the Introduction (“These results also highlight that the neuron and its surrounding glial coat form a functional unit that orchestrates nerve repair”) is much more appropriate as it follows from a summary of the data to be presented. (However, there remains a concern that those data may not support that conclusion, as outlined below)

We agree that in this part of the introduction, we can simply refer to SGC and neuron forming morphological units.

**OK

“Structural neuron-glia units, similar to these, do not exist in the central nervous system”

This could be debated. The authors may wish to provide some more description to clarify their point and ensure credibility.

We have provided references to key papers studying the morphology of SGC neuron to support this claim.

**The additional references appear to relate only to the existence of a neuron-glia relationship in a single peripheral ganglion, not to the lack of their existence in the CNS. The original concern I expressed relates to the original claim that similar relationships do NOT EXIST in the CNS. If the cited papers contain information regarding the lack of neuron-glia units in the CNS, please clarify. If those papers cite other papers demonstrating that these relationships DO NOT exist in the CNS, please cite those papers. In a quick check of the 2005 paper (including a text search of terms) I could not find anything that supported this claim. Since the claim regards a negative, which is difficult to prove, I suggest instead that the authors consider the value of this statement for their message and perhaps edit it to a positive statement that can be supported.

“These studies show that communication between neuron and SGC is critical for neuronal excitability and nociception...”

It is not clear that those cited studies indicated a relationship that is “critical”. The author’s own

language elsewhere implies a less-direct relationship (“modulate” as opposed to “regulate” or “is required for”).

We agree and have edited this sentence and revised the entire paragraph.

**OK

With the exception of a description of SGCs, the overall design of the Introduction narrative lacks a relationship to the major experimental manipulations and conclusions. The major GOF/LOF manipulations focus on SGC Fasn and PPAR-alpha and their role in modulating regenerative axon growth, but there is nothing in the introduction that speaks to anything that is known about these systems in axonal regeneration, even though there are relevant data. Describing those data will provide better context and does not impact the novelty or impact of the current work. These changes are only suggested.

We agree and have added a paragraph in the introduction that focuses on Fasn and PPAR α and their role in modulating axon growth.

**OK.

RESULTS and METHODS

“We anticipated obtaining a larger representation of neurons in our dataset, but their large cell diameter (up to 50 micron for proprioceptors), is likely limiting them from passing through the microfluidic pipes in the Chromium 10X device.”

- It is not clear if the speculation is that all neurons are impeded through the microfluidics, or only the large diameter. There should be data available from the company or core facility regarding the tolerance of the microfluidics. The vast majority of neurons are small and medium, so the implication that the small proportion of neurons that are 40-50um might account for the unexpectedly low recovery of neurons is not entirely logical.

- It is also possible that the large cells died, as they are more susceptible to mechanical dissociation.

- Examination of genes specific for, or enriched in, different populations of sensory neuron types, could be useful to determine if the recovery/loss was selective. Ultimately, however, it is not clear why this speculation is included in the Results. It may be more suitable for the Discussion.

We have revised the result section according to reviewer #2 to include cell numbers. We also have include in Figure S1c an analysis of the sensory neuron subtypes recovered. The majority of neurons (58%) were small diameter (nociceptors, TrkA/CGRP). We have revised the results section accordingly indicating the distribution of neuronal populations in our scRNAseq (p. 6).

**OK.

The manuscript must enhance its description of the bioinformatic and gene-expression analyses. The existing description is terse and not sufficient for reviewers to determine suitability or to provide utility for readers. For example, the description: “...using the Chromium Single Cell Gene Expression Solution (10x Genomics) (Figure 1A). Graph-based clustering, using Partek flow analysis package...” is not at all sufficient for assessment or reproducibility.

For further example: “We thus examined the transcriptional similarity between astrocytes and SGC by comparing the top expressed genes in each cell type, using our scRNAseq data for SGC and a previously published transcriptional analysis of astrocytes (500 genes). We found that SGC share about 10% of genes with astrocytes...”

- This description (not supplemented by anything useful in the Methods) does not describe how “top expressed genes” were determined, or what cut-offs were used to determine expression levels for comparing between cell types.

We agree and have revised both results (p. 5) and method section (p. 32) to provide a better description of single cell analysis, gene-expression analyses and filtering criteria of cells.

**This is improved. However, there is no section in the methods called “filtering criteria”, as implied in the Results (“see filtering criteria in the methods”).

“We next compared the top expressed genes in SGC (605 genes) and Schwann cells (572 genes),

which revealed that SGC share only 2% of genes with Schwann cells in the DRG”

- Correct to “...share only 2% of THOSE gene TRANSCRIPTS with...”

Corrected

- Again, not clear what criteria establishes the “top expressed” genes.

The section of top expressed genes is now explained in detail on p.7

**I assume this is the explanation “ANOVA threshold >4 fold change p-value<0.05 compared to all other populations in the DRG” as this seems to make sense. However, this needs a bit of editing to make it grammatically correct and clear. Perhaps something like “fold-change >4, significant differences across groups by ANOVA, and p<0.05...[for some other comparison that is not clear]” Is the comparison to other populations an average of those groups as implied on p.5, or is it a pairwise comparison which also has pairwise t-tests performed?

The manuscript includes excellent data demonstrating that the Fabp7 antibody does not bind elsewhere – examined in KO mouse tissue – but it is not clear that there are data showing it does bind to Fabp7 (although it does bind to the expected cells). Clarification of this point would strengthen the manuscript.

Validation of antibodies in KO cell lines or KO animals is the gold standard and this is the procedure recommend by NIH guidelines on for authentication of reagents. We believe that the location of the Fabp7 staining, observed by us as well as others (Trevisan et al 2020) combined with the validation in a genetic null mouse and our single cell data is sufficient to demonstrate that Fabp7 binds the target protein in SGC. We believe this point was clearly addressed in results section, p.8.

**The point was that overexpression or heterologous expression would have added further support, but the data provided is certainly sufficient.

“We observed no abnormalities in the morphology of the SGC surrounding the neuron somas in the FasnckO animals compared to controls...”

- Unless there was a quantitative assessment of some set of criteria, this should be corrected to read “We observed no QUALITATIVE abnormalities...”. If a quantitative analysis was performed it should be described.

We agree and have performed a quantification of the SGC nucleus area and circularity, which is now presented in Fig. 4b,c. As described above for reviewer #1, the area of the SGC largely depends on where the section is taken with respect of the 3D arrangement of the SGC coat around the neuron. We thus quantified the nucleus area and now report area as well as the ratio between x-y axis to reflect circularity.

**This is an impressive response!

[The examination of ATF3 was a strength of the project – it could have revealed stress on neurons or other cells (SGC) related to the KO. Unfortunately, the strength was not fully-realized because only the mRNA from homogenized DRG was examined, so cell-type specificity is absent.]

We agree that examination of ATF3 as a stress marker is important. We considered dissociating neurons in control vs WT DRG to assess ATF3 levels. However, the dissociation and culture process are in itself a stress and would not allow us to make strong conclusion (see Wangzhou et al 2020, BioRxiv). However, in section of whole DRG, most of the AFT3 staining is in neurons (now presented in new Figure S6A) and also well document by prior studies that use ATF3 expression as neuronal injury reporter (Holland et al 2019). We therefore think that qPCR for ATF3 from whole DRG may represent relatively well ATF3 expression in neurons.

We now also provide additional data in Fig. 7b that shows that mRNA levels of ATF3 as well as Gap43 are downregulated in FasnckO, and rescued by fenofibrate treatment (Fig. 6b and S6b).

**OK. With dissociation and plating, ATF3 protein is not expressed for about 8 hours if you had wanted to go that way, but the better way is tissue section so it seems that Figure 6S really serves the purpose.

Using cleaved-caspase as an indicator of cell death assumes only a single mechanism of death (and stereological counts were not performed, which could have revealed a difference in overall

cell number, which could have indicated cell death via some other mechanism(s)). This is largely acceptable because it represents the mechanism known to exist in WT animals, but it should be acknowledged in the Results or Discussion that other forms of cell death exist and were not assessed.

We appreciate the suggestion and have edited the results section to reflect that other types of cells death can't be excluded on p.11.

**OK.

The reduced ATF3 expression in the injury condition was assessed by qPCR and the interpretation of the data appear to assume that the DRG neurons are the only source of post-injury ATF3. Regardless, the difference could be due to decreased expression per cell or fewer cells expressing at the same level/cell as control. Also, homogenized DRG includes SGC, which might (do?) express ATF3 after nerve injury. The reduced ATF3 mRNA could be related in some way to the loss of Fasn in SGC, as implied in the text. However, the text currently implies the reduced ATF3 is only in neurons and so the effect of Fasn-KO in SGC is somehow trans-cellular, but this is not necessarily true. This possibility can be dealt with in two ways – either the interpretation can be expanded to allow for these possibilities, or new data can be collected to assess ATF3 expression in terms of number and type of cell, as well as magnitude per cell (if necessary).

We appreciate the suggestion and have revised the text to better account for the different possibilities on p. 16 . As stated above, we also added new data to support the fact that the ATF3 mRNA increase after nerve injury is largely neuronal. We also provide a new data set showing that fenofibrate treatment rescues the decreased ATF3 expression after injury in FasnCKO.

**OK!

Average axon diameter (Figure 4C) may not be the best measure, because there is such a wide range of sizes with very different numbers giving rise to the different ends of the spectrum. A cumulative sum histogram (or similar) would be far more representative of reality.

We agree and have modified the graph to a frequency histogram, now present in Fig. 4f.

**OK.

"The robust bidirectional communication between neurons and SGC is critical for neuronal excitability and nociception and the excitability of sensory neurons is controlled in part by the surrounding SGC"

- As with the comment regarding a similar claim in the Introduction, please ensure that the terms "critical" and "controlled" are accurate representations of what the cited articles show.

We agree and have revised the text accordingly

**OK.

"Whole-cell recordings were performed in acutely dissociated cocultures of DRG neurons and glia"

- Electrophysiological recordings starting 24h after plating does not qualify as recording from "acutely dissociated" DRG neurons. Acutely-dissociated neurons from naïve animals rarely fire multiple APs to the depolarizing stimulus applied, but neurons from cultured DRG neurons or from injured DRG neurons (acute-dissociated or cultured) can show repetitive firing, as is shown in Figure 4L. The control and KO are similar, so this difference is likely not important in terms of the effects of the KO, but it matters for 1) credibility and 2) to provide proper context for readers. Please correct any instances to "short-term culture" or a similar term, not "acutely-dissociated" (observed at least in Results and Methods).

We agree and have revised the text accordingly, referring to this culture as "short-term culture"

**OK.

"Therefore, Fasn deletion in SGC does not affect functional properties of DRG neurons IN THE NAÏVE DISSOCIATED CONDITION."

We agree and have revised the text accordingly

**OK.

Regarding the effect of Fasn-cKO on axonal regeneration and the priming-effect:

- The in vitro work used DRG taken from E13.5 mice, which were maintained in vitro for 7d before additional injury (achieving a relative "age" of E20.5), and fixed 24h later (achieving a relative "age" of E21.5/P0). This time is largely considered still developmental for sensory neurons, in which injury is more an effect on the developmental program (outgrowth and survival) than on the adult injury/stress response, which can include axonal regeneration. The data can still have value, of course, but perhaps should not be labeled as regeneration per se, especially when other aspects of the project are examining true regeneration. Making a terminological equivalence between these models suggests a mechanistic one, which is not a given, and certainly not supported in the narrative. Unless the authors provide some rationale in the text (not just in response to the reviewer) to use the same terms for these different models, the Embryonic injury and growth model should be given

another term, perhaps "developmental axonal growth", "developmental axonal response to injury", or something similar.

The eDRG culture model has been used extensively by us and others to study axon regeneration. While we agree that these are embryonic neurons with high growth capacity, our findings in eDRG have been recapitulated in large part in adult in vivo system (Cho et al, 2012, 2014, 2015). Our unpublished RNAseq analyses of such eDRG culture also suggest that in vitro axotomy recapitulates expression of the expected regeneration associated genes and downregulation of genes related to ion channels observed in vivo (Lisi et al 2017).

**OK, and thank you. This was informative. I'll look forward to seeing that RNAseq data published.

"For adult DRG cultures DRG were dissected from naive mice."

- Provide the age/sex of the mice.

Age and sex is now provided in method section

**OK.

"Contralateral nerve and DRG served as uninjured controls, when needed."

- Please indicate if the contralateral side was sham (and if so if it was skin incision with/without muscle dissection) or intact.

The contralateral is the uninjured side, not a sham surgery. Details is now provided in method section

**OK.

The model uses a 3d post-crush time for regeneration assessment. This seems to offer a limited picture. It does not allow for assessing the normal range of "initiation rates" known to exist across the different types of peripheral axon types. It limits this assessment to only the fastest responders, but the claims are made broadly. It is entirely possible that the neurons which are slower-initiators in the WT have an identical initiation rate in the KO, but this possibility is not addressed.

3d post crush allows us to relate our finding to most of ours and others' previous findings and is a time at which defects can be observed. Shorter time, such as 1d post crush, does not allow to see differences in KO models such as DLKKO, HIF1a cKO that affect activation of a pro-regenerative program (see Cho et al 2015 and Shin et al 2012). 3 days allows the regeneration program to be turned on and if impaired, defects can be observed. We also previously showed that nociceptor growth is similar to other neuron types growth (Carlin et al 2019, Abe et al 2010)

Examination of other time points and neuronal subtypes is beyond the scope of the current study. The possibility that different neuronal subtypes are differentially affected by PPAR α signaling SGC remains is now been mentioned in the discussion, p. 18.

**OK. The point on p. 18 addresses this concern satisfactorily, and I entirely agree that the point does not warrant any additional data collection.

The images of the nerve in the figure imply a rather significant qualitative difference in the

architecture of the distal nerve, with the KO nerve looking much closer to normal (with visible undulations), and even with what appear to be SCG10+ structures far distal to the injury site. This should be addressed better. Providing images of axonal structure (beta-3 tubulin?) would significantly enhance the context for readers.

We have added images of nerve sections stained for Tuj1 from control and FasncKO in figure S4a, showing that the overall architecture of the nerve is similar.

Arrow heads have been added to Fig. 5a and 7c to emphasize the longest regenerating axons.

"...the injury site was defined as the area with maximal SCG10 intensity."

- Please indicate if this was established from a single section, or a consensus across many sections. Also indicate what qualified as "area with maximal SCG10" – a single pixel? 10 contiguous pixels? Etc. This method of defining the injury site may have been convenient, but it is not (at least as presented here) terribly clear or strong. Using markers of denervated Schwann cells or infiltrating cells, even as a proof of concept for the SCG10-intensity method, would have strengthened the assessment significantly. This point of clarity matters greatly as it is the basis for the quantitative measures.

This is an assay that we and others have used extensively. The crush site is defined as the site along the nerve length where SCG10 intensity is maximal when measured in a vertical line was across the nerve. The quantification assay is explained in details in methods, p. 34. SCG10 was chosen, because it selectively labels regenerating axons and displays higher specificity than GAP43 or YFP in the early stage of axon regeneration after nerve crush (Shin et al 2013).

**I understand that this method has been used previously. That does not address the main concern, though, and, as stated previously, clarity in this methodological point matters greatly. Regarding simply referencing prior publications: the Shin 2012 article appears to have even less detail with regard to this method than does this manuscript. The Cho 2015 article states "To assess the regenerative capacity of injured axons, the ratio of SCG10 fluorescence intensity proximal and distal to the axotomy line was measured...", but it does not actually state how the "axotomy line" was determined to any greater degree than this manuscript. Figure 4 legend states "Dotted lines indicate the crush site, identified as the maximal SCG10 intensity", which simply raises the same questions. The Methods section states "SCG10 fluorescence intensity was measured along the length of the nerve using a line scan macro in ImageJ", which would be an excellent addition to this manuscript because it suggests that the "highest intensity" was determined objectively (which was not clear before), but still leaves some questions. How many line-scans were used? How does the procedure deal with a situation in which 2 or more pixels across the width of the nerve section all have the same or very similar "highest-intensity" yet are spaced far-apart along the length of the nerve section?

** Therefore, please state clearly in this manuscript:

1) if the "area with maximal SCG10 intensity" was determined for each section. If not done on a per-section basis, then state how the site was determined for the other sections in which the "maximal SCG10 intensity" was not determined. If it was done objectively as implied by the reference to the Cho paper, state that here as well, and add the clarification about how many lines were used and any procedures to deal with conflicts.

2) what qualified as "area with maximal SCG10" – a single pixel? 10 contiguous pixels? Etc. Perhaps it was a single pixel and the "injury-line" was then drawn perpendicular to the long-axis of the nerve section through that point?

3) if there was any procedure to ensure that the injury site determined within any single section corresponded to those determined in others – i.e., they were not 2mm apart. This addresses the possibility that the "injury site" determined in each section may define an injury that does not reflect physical reality. This could be as simple as something like ensuring that all SCG10-defined injury sites were within the same distance (which should correspond very closely to the width of the forceps used to crush the nerve) from the cut-end of the nerve sample, or something similar.

- Certainly there are differences detected, but the 3 day post-injury time point seems sub-optimal for the claims made. It is at the leading edge of what is reported for the latency to the cell-body reaction and start of axonal regeneration, which would fit well if the only goal were to assess the

fastest response-initiation. Although that is included, the manuscript also uses these data to support claims regarding the regeneration process in general. Assessments at later time points would have greatly strengthened the claims. The claims throughout the manuscript regarding regeneration should be revised to reflect this point.

We have now included in the discussion that whether SGC contribute to sustain regenerative growth until target reinnervation remains to be determined (p. 18).

**OK.

"Neurons in injured FasncKO displayed reduced neurite length compared to injured controls, but similar initiation rates (Figure 5D-F)."

- The outcome measure relies on the "longest 10 axons". There is nothing intrinsically flawed in this, but it does introduce a bias toward the most effective and/or fastest-starting axons. This is problematic because these may be a subpopulation, or the effect of Fasn/PPAR may only be on the Figure 5D-F is in vitro growth, and the measure is axon length per neuron, not longest 10 axons

**YES! Apologies – I incorrectly merged these 2 things.

- It is possible that the "initiation rate" was unaffected by the Fasn-KO as claimed, but there are other interpretations that could exist (false-negative). As described, the "initiation rate" was determined by the post-plating time at which neurites emerged to some qualifying characteristic. This was done without regard to the type of sensory neuron giving rise to those neurites. Unless all SGC express the same level of Fasn, and all DRG neurons express the same level of PPAR, then it is possible that the neurons that are responsible for providing the "equivalent initiation rate" are a subpopulation that, in WT, have some reduced/different PPAR-expression/function and/or are surrounded by SGC with lower Fasn (Figure 3 appears to indicate that there may be some significant heterogeneity in the Fasn expression). In this case, the loss of Fasn (in the KO) would largely be irrelevant as that subpopulation did not have a high degree of SGC-Fasn and neuronal PPAR anyway, and the conclusion could be incorrect. This can be addressed either by providing data regarding the uniformity of the PPAR-expression across neuron types AND the uniformity of Fasn-expression across SGC, OR the assessment of "initiation rate" can be done with sensitivity to neuronal sub-types. Alternatively, the manuscript can alter the claim and provide better presentation of the possible interpretations of the data.

We have revised the text and rather than initiation rate, refer to percent of neuron initiating axon growth in Fig. 5f and 7h. Whether SGC differently affect neuronal subtypes is now discussed (p.18). The uniformity of Fasn-expression across SGC might be interesting. However, without a morphological correlation of SGC surrounding a given neuron, this information might not explain the effect of Fasn/PPAR expression in different SGC on different subtypes. Whether different SGC subtypes surround different neurons subtypes is a major ongoing study in the lab, but is beyond the scope of the current manuscript.

**OK.

- Using the in vivo outcome to determine "initiation" with the in vivo outcome of "length" as was done leads a particular problem. There is no means for determining if the populations are the same. This again is not necessarily a flaw, except that the narrative regarding both outcomes implies both a uniformity across the entire population of neurons, and implies a union of the population of axons that are longest and first to grow in vitro. These implications are not supported by the experimental design, though they may be true. Therefore, either new data must be provided demonstrating the uniformity and union, or the narrative must be adjusted to clearly provide to the reader an interpretation of the data that includes these possibilities and limits.

We agree that which population of neurons is examined in the in vitro and in vivo assays is not determined in our current experiments. We have added a comment to this effect in the discussion on p.18. As stated above, whether SGC affect neurons subtypes differently is an ongoing study in the lab, but is beyond the scope of the current manuscript.

**I made some errors in stating my original concern which may have made this difficult to understand. Apologies if this was the case. It would be ideal if the manuscript acknowledged the fact that the in vivo and in vitro assays may not be dealing with the same population while the

conclusions assume they are the same, but this is not vital. The rebuttal is acceptable nonetheless.

The outcome measure of "initiation rate" may have additional issues. It appears to be less about the latency of when the range of axons start to regenerate and more about the proportion of neurons that display neurite-outgrowth at a given time. It is not clear that "rate" is a suitable term in the in vitro case. Perhaps "proportion" or similar term would be more accurate.

We agree and have revised the text and rather than initiation rate, refer to percent of neuron initiating axon growth in Figure 5f and 7h.

**OK.

"These results indicate that Fasn in SGC is required for the conditioning effect, and specifically for the elongating phase of axon growth"

- The conclusion of "required" is not supported. Fasn may influence the priming effect, but even this may not be well-supported. Fasn may regulate, to some unknown degree, the priming effect in some population of neurons. However, the limited assessment time does not lend itself to a thorough assessment of priming or regeneration, particularly the elongation phase.

We agree and have used "contribute"

**OK.

"...SCG10 intensity was quantified at 1, 2, and 3mm from the injury site as previously described"

- A brief description needs to be provided. It is not clear if the quantification was the average intensity of a row/line of single pixels (e.g., the demarcation line), or some area of pixels. Details on quantification are now provided in the method section

**OK.

"For both LENGTH AND INTENSITY quantifications, five sections per biological replicate were averaged."

We made these changes

**OK.

In standard situation, PPAR-g is neuronal and PPAR-a is SGC. However, it appears that this pattern was not ensured in the Fasn-CKO. This weakens the conclusion (assumption?) that the fenofibrate (PPAR-a-agonist) is working only on SGC, especially since the effective concentration of fenofibrate was not determined. This weakness is mitigated by the data that fenofibrate does not improve axon regeneration in neuronal cultures that do not contain SGC. The authors should consider including a statement to clarify the context, or doing some direct assessments.

We have included statement in the results and discussion acknowledging the fact that fenofibrate might act on other cells beyond SGC (p.16 and p.19)

**OK.

DISCUSSION

"The role of SGC in nerve regeneration has been largely ignored..."

The term "ignored" is inaccurate. It implies that the role was known but not considered. But the role was not really known until this work – this is even stated in the following sentence. Please reword.

We agree and have revised the text accordingly

**OK.

"Although addressing this question will require further investigation, our data indicate that Fasn expression in SGC is required for the proper expression of at least one regeneration associated gene after injury, Atf3"

- The conclusion of "required" is not supported.

We agree and have revised the text accordingly

**OK, but please correct "participate" to "participates" and "gene" to "genes"

"Our study suggests that neuronal "intrinsic" response to injury may not be purely neuronal but rather includes the SGC surrounding the neuronal soma..."

- Unless this statement is referring to something other than the neuronal ATF3 response (which the authors have not specifically identified) as the "intrinsic" response, then the statement is not supported.

We agree and have revised the text accordingly

**OK.

METHODS

"...and both injured dorsal root ganglia and sciatic nerve were dissected at the indicated time post-surgery"

- Specify which DRGs were designated as "injured" and collected. The designated DRG's were described in figure legends and main text of the original manuscript. This information is now also added to the methods section.

**OK.

Specify the control procedures for immunohistochemistry and western blot.

The details for control procedures were added

**OK for immunohistochemistry. Not added for western blot.

"Cultures were then used for electrophysiological recording 24h after plating or fixed and stained with the indicated antibody"

- Electrophysiological recordings starting 24h after plating does not qualify as recording from "acutely dissociated" DRG neurons, as indicated in the Results.

Agreed and changed to "short-term cultures"

**OK.

Figure 4H shows results from qPCR of ATF3, but there are no primer sequences provided for ATF3. Please provide the primer information.

ATF3 primers sequence were added

**OK.

"An Automated analysis for axon tracing and neurons soma count was used for ex-vivo adult DRG culture experiments (Nikon elements)."

- Please provide specifics for the "automated analysis".

Nikon elements is a commercial software package for image analysis using an algorithm for automatic digital reconstruction of axons. The specific analysis code is available upon request (requires NIS-Elements and General Analysis).

** Yes. We use Elements as well, as do many others. What is needed here is not the algorithm or code, but the user interaction parameters and procedures. Even with an "automated procedure", the user does have to teach Elements what features of the image are of interest (i.e., to include in measurements). In many cases of analyzing images from in vitro systems, a thresholding step is used. Utilizing fluorescent signal to create a binary layer, through thresholding, allows a user to teach Elements biological features such as an axon and a soma. In this case the user must define for Elements what the threshold is, and the user must somehow define the thresholding settings. In many cases, there is a background correction involved. Also, there can be manual or automatic selection of "regions of interest" or "objects", and often a removal of "non-relevant signal" such as axon debris. On the acquisition side, it is not clear that there was a common set of image capture protocols used across conditions.

Perhaps the lab uses a module called "General Analysis" that can make automation a bit easier.

Even if so, users can share these recipes with other users that would allow them to replicate the analyses (assuming the acquisition parameters are also replicated).

The manuscript must include a description of these and other relevant procedures.

"Statistics was performed using GraphPad (Prism8) for either Student's t-test or ANOVA analysis."

- This is not sufficient. Please ensure that this section is complete and that the description of the statistics in Methods and elsewhere are consistent and clear.

- Indicate type of ANOVA

- Perhaps the authors have omitted mention of the post-ANOVA tests in the description in the Methods. Figure 2 legend mentions the Sidak multiple comparison test, suggesting this terse description in the Methods is in error. Sidak or Dunnett tests were used as part of one- and two-way ANOVA for multiple comparison tests.

A description of the statistical tests was added

**OK.

- Gene expression mentioned use of FDR, but this is not described. An FDR value is a p-value adjusted for multiple tests (by the Benjamini-Hochberg procedure). It stands for the "false discovery rate" it corrects for multiple testing by giving the proportion of tests above threshold alpha that will be false positives (i.e., detected when the null hypothesis is true). A description was added.

**Yes, I'm aware of what FDR is, but thank you for the explanation in case I was not. The comment was because the abbreviation FDR was not defined, and it was not mentioned as part of the statistical procedures. Looks OK now.

- Most legends indicate which statistical tests were used, but not all. Figure 4 legend has p-values but no indication of the tests used

A description of the statistical tests was added to Fig. 4 legend

**OK.

Minor

"Structural neuron-glia units, similar to these, do not exist in the central nervous system"

Commas likely not necessary

Commas were removed

**OK.

Throughout, please ensure that the language consistently and properly uses the terms "gene" and "transcript". In many places the term "gene" is used where "transcript" should be used. The Chromium Single-Cell 3' Solution we used for the scRNASeq can recognize a general gene but not a unique transcript or splicing variants, therefore we chose to use the term 'gene expression' and not 'transcript expression' to avoid misleading the reader.

**OK. I see the point. I think the greatest clarity would come from stating the above in the manuscript.

"Our single cell RNAseq analysis using freshly dissociated tissue thus UNRAVELS a unique transcriptional profile of SGC in DRG..."

corrected

"We found that SGC share about 10% of THOSE gene TRANSCRIPTS with astrocytes, among them Fabp7..."

corrected

"We crossed BLBPCre-ER to the Rosa26-fs-TRAP and observed that following 10 days OF tamoxifen treatment..."

OR

"We crossed BLBPCre-ER to the Rosa26-fs-TRAP and observed that following A 10 day tamoxifen treatment..."

corrected

"Pathway analysis of other major CELL types in the DRG confirmed fatty acid metabolism as a

unique pathway enriched in SGC..."

corrected

"...our pathway analysis INDICATED that the cell cycle term is enriched in macrophages, but not SGC..."

Also, this sentence is awkward. The "cell cycle term" is not enriched in macrophages. Please reword.

corrected

"...loss of Fasn in SGC does not lead to morphological deficits in the DRG and does not ELICIT a stress response in neurons"

corrected

"Fenofibrate is used clinically to treat lipid disorders, but has been unexpectedly been shown in clinical trials..."

- "been" is redundant and grammatically incorrect

corrected

"During surgery, 8-12 week old C57Bl/6 mice OF the indicate genotype were anesthetized"

Corrected

**OK to all.

Reviewer #4:

Remarks to the Author:

The authors have addressed my concerns appropriately. I am happy to endorse this manuscript for publication in Nature Communications.

Detailed responses to reviewers.

Reviewers' comments are in black. Authors' responses are in blue.

Reviewer #1 (Remarks to the Author):

The authors were responsive to reviewers' comments and my concerns are fully resolved after revisions.

We are pleased that this revised version has fully satisfied this reviewer.

Reviewer #2 (Remarks to the Author):

More direct evidence is needed to support the statement that “These results identify fatty acid synthesis in SGC as a fundamental novel mechanism mediating axon regeneration in adult peripheral nerves”.

We appreciate the reviewer's suggestion to perform more direct experiments in support of this statement. We addressed two of the reviewer's points with the new experimental evidence and additional clarifications, as described in detail below. We also believe that one of the remaining two points is caused by a misunderstanding, which we clarify below. Finally, we believe that the remaining suggested experiments are not informative and will not allow us to provide any additional direct evidence to support that statement. Moreover, we are not aware of any other approaches that could provide more direct evidence than the set of experiments we already presented. We detail below for each point why the suggested experiments will not provide any additional direct support to our conclusion. Nevertheless, following the reviewer's suggestion, we revised our statement.

1. Lipidomics or other metabolic measure is actually crucial to support the claim on fatty acid synthesis or lipid metabolism. If the data cannot be provided due to technical difficulty, the conclusion should be tuned down.

We appreciate the reviewer's suggestion. However, as already mentioned in the first round of revision, lipidomics requires abundant material. Therefore, these experiments can only be done on whole DRG, making any results regarding cell-specific changes (and thus changes in SGC) inconclusive. Furthermore, phospholipid species that are ligands for PPAR are likely to be a minor fraction of total lipid given their role in signaling. Therefore, an approach using mass spec analyses of phospholipids of entire DRG is unlikely to prove that the change in any species is responsible for the effects on PPAR activation in SGC. Based on these considerations, we believe that measuring fatty acid or phospholipid levels in whole DRG will not allow us to provide any further direct evidence to support a role for fatty acid synthesis in SGC.

We believe that the genetic experiment we presented, in which we deleted fatty acid synthase (the enzyme that controls the critical step in endogenous fatty acid synthesis) specifically in satellite glia, provides the most direct evidence for a role for fatty acid synthesis in SGC. Nevertheless, following the reviewer's suggestion, we revised and tuned down our statement to be more specific, which now reads as:

“These results indicate that PPAR α activity downstream of Fasn in SGC represents a novel mechanism mediating axon regeneration in adult peripheral nerves.”

2. Fig 7a did not sufficiently support the claim “Activation of PPAR α in SGC” because the whole DRGs were used to do RT-PCR. PPAR α immunostaining in DRG w and w/o injury, and w and

w/o fenofibrate diet should help to clarify and differentiate the PPAR α expression in SGC and neurons. PPAR α is within the scope of the manuscript because it is the claimed mechanism to support the conclusion.

We thank the reviewer for this point and agree that determining PPAR α expression in SGC and neurons is central to the manuscript. Indeed, in the first submission of the manuscript we provided data to show that PPAR α and PPAR α target genes are upregulated in SGC but not in neurons after injury (**Fig 6b**). We also showed in an *in vitro* assay that fenofibrate does not promote growth in pure neuronal cultures, further supporting that PPAR α is not expressed in neurons (**Fig. 6g and S5**). Following the reviewer's current suggestion, we performed immunostaining for PPAR α in naïve DRG, in DRG from mice that had a prior nerve injury and in DRG from mice that received a fenofibrate diet for 2 weeks. We present these results for naïve mice in **new Fig 6c** and injured and fenofibrate in **new Fig. S6a**. These results clearly demonstrate that PPAR α is expressed in SGC but not neurons, and that neither injury nor fenofibrate treatment lead to PPAR α expression in neurons.

Furthermore, a recent transcriptional profiling study of sensory neurons at single cell resolution provided an online resource at www.painseq.com (Renthal W. et. al., BioRxiv, 2019). This independent online resource confirms that PPAR α is not expressed in neurons, neither in naïve conditions nor following sciatic nerve crush injury.

We believe that our analysis together with the added immunostaining for PPAR α , and combined with the single cell data available from others (Renthal et al 2019), strongly support the notion that PPAR α and PPAR α target genes are expressed in SGC but not neurons. Moreover, neither injury nor fenofibrate lead to PPAR α expression in neurons. Therefore, we are now confident that the analysis of PPAR α and PPAR α target gene expression in whole DRG as presented in Fig. 7a supports the notion that fenofibrate leads to elevated PPAR activity in SGC, but not neurons. We cannot exclude that fenofibrate act also on other cells beyond SGC, and this was acknowledged in the first revised version.

3. Direct manipulation of PPAR α in SGC or DRG neurons is still important to address the concern. It is actually not a very difficult experiment. One possibility is to knockdown PPAR α in DRG neurons using virus and add fenofibrate.

We appreciate the reviewer's point but we believe this concern is caused by a misunderstanding.

First, it is not clear if the suggested direct manipulation of PPAR α is meant to be achieved in vivo or in vitro.

Second, it is unclear why we would need to delete PPAR α in neurons in which it is not expressed. Indeed, as discussed above in point #2, two independent data set (Carlin et al 2019, Renthal et al 2019) that examined transcriptional profile of neurons in naïve and injured conditions, revealed that PPAR α and PPAR α target genes are not expressed in neurons. The immunostaining that we now provided further support that PPAR α is not expressed in neurons.

Third, in vivo manipulation of SGC would require crossing PPAR α floxed mice to BLBPcre-ER. As stated by reviewer #1 in the first round of review, genetic manipulation of PPAR α expression is important but beyond the scope of the current manuscript. Furthermore, knockdown of PPAR α can cause compensatory effects by other PPAR, like PPAR γ (Patsouris D. et. al.

Endocrinology, 2006). Therefore, knocking down PPAR α could lead to results that would be difficult to interpret and would not provide the direct evidence that the reviewer requests.

The alternative suggested by the reviewer is to perform knock down (KD) using a viral approach. KD efficiency is not as robust as genetic deletion, and the number of cells targeted is variable, rendering interpretation of the outcome measures difficult. Furthermore, there is no established approach to target specifically and efficiently SGC with a viral approach. This is why we characterized a novel cre driver line to manipulate SGC genetically in vivo in this study. While we could attempt to use the Fabp7 promoter for viral targeting of SGC and test several AAV serotypes, this is rather a difficult and lengthy set of experiments that will require extensive validation, which is beyond the scope of the current manuscript.

If the reviewer meant to perform these knock down experiments in neurons and SGC in vitro, the same concerns regarding efficiency of knock down and infection exist, as well as no yet proven viral vector to specifically target SGC in vitro. Furthermore, we show in Fig5g and S5c that fenofibrate has no effect on axon regeneration in neuronal only cultures. This further supports the notion that PPAR α is not expressed in neurons.

We believe that our analyses provide strong support to the notion that PPAR signaling in SGC is important for axon regeneration in adult peripheral nerves. The experiments suggested by the reviewer will not allow us to provide any further direct evidence beyond the analyses presented in this revised manuscript.

4. Fig 5A, 7C are not representative to the quantification of 5C, 7E. At 1000um, axons from KO are much fewer than control.

We thank the reviewer for this point and would like to clarify that the regeneration index in Fig.5c and 7e represent the SCG10 intensity at different distances from the injury site relative to injury site in each nerve section. A direct comparison of intensity between different nerves is not relevant. We clarified this further with a more detailed explanation in the Method section for image analysis on p.35.

Reviewer #3 (Remarks to the Author):

** Reviewer comments on revision in RED and with **

This manuscript from Cavalli and colleagues reports on work examining the biology of satellite glial cells of the dorsal root ganglion. The work is highly novel, reporting a number of new characteristics of the SGCs, including a transcriptional profile providing a set of SGC-specific genes, their intimate relationship with sensory neurons (possibly constituting a functional unit), responsiveness to nerve injury (which is a stimulus that is remote to them), and the potential of a regulatory role in the sensory neuron response to the injury, including at least some aspects of axonal regeneration. It is largely a very well-controlled and designed study with strong quality control on reagents and on genetic manipulations. The work provides specific mechanisms where only general suggestions were available previously, and promises to be welcomed by the field as a cornerstone for future reference. There are a few points that require attention to ensure the reality meets the potential. It is likely that no new data is required (though the authors may wish to add), but simply revision in the text to enhance clarity and credibility, particularly regarding axonal regeneration. There is sufficient over-interpretation of some data to warrant revision. The most significant single point

appears to be the treatment of the data related to regeneration, and the framing of the manuscript. Reading the Title, Summary, Abstract, and Introduction would lead one to believe that the manuscript is largely concerning the regulation of axonal regeneration by SGC-derived fatty acids. But this is not the actual major content of the manuscript, and the data supporting that message are not the strongest in the manuscript. The manuscript appears to base many the strongest claims on the effects on expression of ATF3, but the assessment of ATF3 expression is not sufficient to support those claims. It bases other strong claims on assessments of an unspecified very small group of axons/neurons and uses those observations to extend the claims to the entire population. The claims may yet be true, but they are not sufficiently-supported. The authors need to revise some of the claims and should consider re-framing the manuscript to center on the strongest and greatest-volume of the data – SGC-neuron structural relationship and SGC biology – OR they can enhance the data related to regeneration to bring those data to the point where they can adequately support the focus of the message on axonal regeneration.

We thank the reviewer for his careful and thoughtful reading of our manuscript and his constructive and detailed suggestions. We have performed the suggested revisions in the text to enhance clarity and credibility, avoid over interpretation. We believe these revisions have increased the quality and depth of the paper.

**The revisions have been very thorough, and only a few items remain to be addressed. I look forward to seeing this work published.

We are pleased that this reviewer is satisfied with the revision. We addressed all remaining minor suggestions in the revised text, as follows:

Introduction

“In adult animals, multiple SGC form an envelope that completely enwraps each sensory neuron 5-7. The number of SGC surrounding sensory neurons increases with increasing soma size in mammals 8, 9. The neuron and its surrounding SGC thus form a distinct morphological and functional unit”

- The statement “The neuron and its surrounding SGC thus form a distinct morphological and functional unit” seems a bit premature in this place in the manuscript. The statements preceding this in the Introduction only suggest the possibility of a functional unit (correlative).

Although the data generated and analyzed in this report somewhat support the claim of a functional unit, those data have not been presented yet in the manuscript. The claim at the end of the Introduction (“These results also highlight that the neuron and its surrounding glial coat form a functional unit that orchestrates nerve repair”) is much more appropriate as it follows from a summary of the data to be presented. (However, there remains a concern that those data may not support that conclusion, as outlined below)

We agree that in this part of the introduction, we can simply refer to SGC and neuron forming morphological units.

**OK

“Structural neuron-glia units, similar to these, do not exist in the central nervous system”

This could be debated. The authors may wish to provide some more description to clarify their point and ensure credibility.

We have provided references to key papers studying the morphology of SGC neuron to support this claim.

**The additional references appear to relate only to the existence of a neuron-glia relationship in a single peripheral ganglion, not to the lack of their existence in the CNS. The original concern I

expressed relates to the original claim that similar relationships do NOT EXIST in the CNS. If the cited papers contain information regarding the lack of neuron-glia units in the CNS, please clarify. If those papers cite other papers demonstrating that these relationships DO NOT exist in the CNS, please cite those papers. In a quick check of the 2005 paper (including a text search of terms) I could not find anything that supported this claim. Since the claim regards a negative, which is difficult to prove, I suggest instead that the authors consider the value of this statement for their message and perhaps edit it to a positive statement that can be supported.

We totally agree and have removed this statement from the introduction section.

“These studies show that communication between neuron and SGC is critical for neuronal excitability and nociception...”

It is not clear that those cited studies indicated a relationship that is “critical”. The author’s own language elsewhere implies a less-direct relationship (“modulate” as opposed to “regulate” or “is required for”).

We agree and have edited this sentence and revised the entire paragraph.

**OK

With the exception of a description of SGCs, the overall design of the Introduction narrative lacks a relationship to the major experimental manipulations and conclusions. The major GOF/LOF manipulations focus on SGC Fasn and PPAR-alpha and their role in modulating regenerative axon growth, but there is nothing in the introduction that speaks to anything that is known about these systems in axonal regeneration, even though there are relevant data.

Describing those data will provide better context and does not impact the novelty or impact of the current work. These changes are only suggested.

We agree and have added a paragraph in the introduction that focuses on Fasn and PPAR α and their role in modulating axon growth.

**OK.

RESULTS and METHODS

“We anticipated obtaining a larger representation of neurons in our dataset, but their large cell diameter (up to 50 micron for proprioceptors), is likely limiting them from passing through the microfluidic pipes in the Chromium 10X device.”

- It is not clear if the speculation is that all neurons are impeded through the microfluidics, or only the large diameter. There should be data available from the company or core facility regarding the tolerance of the microfluidics. The vast majority of neurons are small and medium, so the implication that the small proportion of neurons that are 40-50um might account for the unexpectedly low recovery of neurons is not entirely logical.

- It is also possible that the large cells died, as they are more susceptible to mechanical dissociation.

- Examination of genes specific for, or enriched in, different populations of sensory neuron types, could be useful to determine if the recovery/loss was selective. Ultimately, however, it is not clear why this speculation is included in the Results. It may be more suitable for the Discussion.

We have revised the result section according to reviewer #2 to include cell numbers . We also have include in Figure S1c an analysis of the sensory neuron subtupesrecovered. The majority of neurons (58%) were small diameter (nociceptors, TrkA/CGRP). We have revised the results section accordingly indicating the distribution of neuronal populations in our scRNAseq (p. 6).

**OK.

The manuscript must enhance its description of the bioinformatic and gene-expression analyses. The existing description is terse and not sufficient for reviewers to determine suitability or to provide utility for readers. For example, the description: "...using the Chromium Single Cell Gene Expression Solution (10x Genomics) (Figure 1A). Graph-based clustering, using Partek flow analysis package..." is not at all sufficient for assessment or reproducibility. For further example: "We thus examined the transcriptional similarity between astrocytes and SGC by comparing the top expressed genes in each cell type, using our scRNAseq data for SGC and a previously published transcriptional analysis of astrocytes (500 genes). We found that SGC share about 10% of genes with astrocytes..."

- This description (not supplemented by anything useful in the Methods) does not describe how "top expressed genes" were determined, or what cut-offs were used to determine expression levels for comparing between cell types.

We agree and have revised both results (p. 5) and method section (p. 32) to provide a better description of single cell analysis, gene-expression analyses and filtering criteria of cells.

**This is improved. However, there is no section in the methods called "filtering criteria", as implied in the Results ("see filtering criteria in the methods").

We agree and have edited the method section to describe filtering criteria

"Low quality cells and potential doublets were filtered out from analysis using the following parameters: total reads per cell: 600-15000, expressed genes per cell: 500-4000, mitochondrial reads <10%. A noise reduction was applied to remove low expressing genes <=1 count." This statement appears in the Methods, now presented as a separate section for better clarity.

"We next compared the top expressed genes in SGC (605 genes) and Schwann cells (572 genes), which revealed that SGC share only 2% of genes with Schwann cells in the DRG"

- Correct to "...share only 2% of THOSE gene TRANSCRIPTS with..."

Corrected

- Again, not clear what criteria establishes the "top expressed" genes.

The section of top expressed genes is now explained in detail on p.7

**I assume this is the explanation "ANOVA threshold >4 fold change p-value<0.05 compared to all other populations in the DRG" as this seems to make sense. However, this needs a bit of editing to make it grammatically correct and clear. Perhaps something like "fold-change >4, significant differences across groups by ANOVA, and p<0.05...[for some other comparison that is not clear]" Is the comparison to other populations an average of those groups as implied on p.5, or is it a pairwise comparison which also has pairwise t-tests performed?

We thank the reviewer and have revised this sentence as suggested and clarify the comparison analysis on p.7.

The manuscript includes excellent data demonstrating that the Fabp7 antibody does not bind elsewhere – examined in KO mouse tissue – but it is not clear that there are data showing it does bind to Fabp7 (although it does bind to the expected cells). Clarification of this point would strengthen the manuscript.

Validation of antibodies in KO cell lines or KO animals is the gold standard and this is the procedure recommend by NIH guidelines on for authentication of reagents. We believe that the location of the Fabp7 staining, observed by us as well as others (Trevisan et al 2020) combined with the validation in a genetic null mouse and our single cell data is sufficient to demonstrate that Fabp7 binds the target protein in SGC. We believe this point was clearly addressed in results section, p.8.

**The point was that overexpression or heterologous expression would have added further support, but the data provided is certainly sufficient.

Thank you.

“We observed no abnormalities in the morphology of the SGC surrounding the neuron somas in the FasnckO animals compared to controls...”

- Unless there was a quantitative assessment of some set of criteria, this should be corrected to read “We observed no QUALITATIVE abnormalities...”. If a quantitative analysis was performed it should be described.

We agree and have performed a quantification of the SGC nucleus area and circularity, which is now presented in Fig. 4b,c. As described above for reviewer #1, the area of the SGC largely depends on where the section is taken with respect of the 3D arrangement of the SGC coat around the neuron. We thus quantified the nucleus area and now report area as well as the ratio between x-y axis to reflect circularity.

**This is an impressive response!

Thank you

[The examination of ATF3 was a strength of the project – it could have revealed stress on neurons or other cells (SGC) related to the KO. Unfortunately, the strength was not fully-realized because only the mRNA from homogenized DRG was examined, so cell-type specificity is absent.]

We agree that examination of ATF3 as a stress marker is important. We considered dissociating neurons in control vs WT DRG to assess ATF3 levels. However, the dissociation and culture process are in itself a stress and would not allow us to make strong conclusion (see Wangzhou et al 2020, BioRxiv). However, in section of whole DRG, most of the AFT3 staining is in neurons (now presented in new Figure S6A) and also well document by prior studies that use ATF3 expression as neuronal injury reporter (Holland et al 2019). We therefore think that qPCR for ATF3 from whole DRG may represent relatively well ATF3 expression in neurons.

We now also provide additional data in Fig. 7b that shows that mRNA levels of ATF3 as well as Gap43 are downregulated in FasnckO, and rescued by fenofibrate treatment (Fig. 6b and S6b).

**OK. With dissociation and plating, ATF3 protein is not expressed for about 8 hours if you had wanted to go that way, but the better way is tissue section so it seems that Figure 6S really serves the purpose.

Thank you

Using cleaved-caspase as an indicator of cell death assumes only a single mechanism of death (and stereological counts were not performed, which could have revealed a difference in overall cell number, which could have indicated cell death via some other mechanism(s)). This is largely acceptable because it represents the mechanism known to exist in WT animals, but it should be acknowledged in the Results or Discussion that other forms of cell death exist and were not assessed.

We appreciate the suggestion and have edited the results section to reflect that other types of cells death can't be excluded on p.11.

**OK.

The reduced ATF3 expression in the injury condition was assessed by qPCR and the interpretation of the data appear to assume that the DRG neurons are the only source of post-injury ATF3. Regardless, the difference could be due to decreased expression per cell or fewer cells expressing at the same level/cell as control. Also, homogenized DRG includes SGC, which might (do?) express ATF3 after nerve injury. The reduced ATF3 mRNA could be related in

some way to the loss of Fasn in SGC, as implied in the text. However, the text currently implies the reduced ATF3 is only in neurons and so the effect of Fasn-KO in SGC is somehow trans-cellular, but this is not necessarily true. This possibility can be dealt with in two ways – either the interpretation can be expanded to allow for these possibilities, or new data can be collected to assess ATF3 expression in terms of number and type of cell, as well as magnitude per cell (if necessary).

We appreciate the suggestion and have revised the text to better account for the different possibilities on p. 16. As stated above, we also added new data to support the fact that the ATF3 mRNA increase after nerve injury is largely neuronal. We also provide a new data set showing that fenofibrate treatment rescues the decreased ATF3 expression after injury in FasnCKO.

**OK!

Average axon diameter (Figure 4C) may not be the best measure, because there is such a wide range of sizes with very different numbers giving rise to the different ends of the spectrum. A cumulative sum histogram (or similar) would be far more representative of reality.

We agree and have modified the graph to a frequency histogram, now present in Fig. 4f.

**OK.

“The robust bidirectional communication between neurons and SGC is critical for neuronal excitability and nociception and the excitability of sensory neurons is controlled in part by the surrounding SGC”

- As with the comment regarding a similar claim in the Introduction, please ensure that the terms “critical” and “controlled” are accurate representations of what the cited articles show.

We agree and have revised the text accordingly

**OK.

“Whole-cell recordings were performed in acutely dissociated cocultures of DRG neurons and glia”

- Electrophysiological recordings starting 24h after plating does not qualify as recording from “acutely dissociated” DRG neurons. Acutely-dissociated neurons from naïve animals rarely fire multiple APs to the depolarizing stimulus applied, but neurons from cultured DRG neurons or from injured DRG neurons (acute-dissociated or cultured) can show repetitive firing, as is shown in Figure 4L. The control and KO are similar, so this difference is likely not important in terms of the effects of the KO, but it matters for 1) credibility and 2) to provide proper context for readers. Please correct any instances to “short-term culture” or a similar term, not “acutely-dissociated” (observed at least in Results and Methods).

We agree and have revised the text accordingly, referring to this culture as “short-term culture”

**OK.

“Therefore, Fasn deletion in SGC does not affect functional properties of DRG neurons IN THE NAÏVE DISSOCIATED CONDITION.”

We agree and have revised the text accordingly

**OK.

Regarding the effect of Fasn-cKO on axonal regeneration and the priming-effect:

- The in vitro work used DRG taken from E13.5 mice, which were maintained in vitro for 7d before additional injury (achieving a relative “age” of E20.5), and fixed 24h later (achieving a relative “age” of E21.5/P0). This time is largely considered still developmental for sensory neurons, in which injury is more an effect on the developmental program (outgrowth and survival) than on the adult injury/stress response, which can include axonal regeneration. The

data can still have value, of course, but perhaps should not be labeled as regeneration per se, especially when other aspects of the project are examining true regeneration. Making a terminological equivalence between these models suggests a mechanistic one, which is not a given, and certainly not supported in the narrative. Unless the authors provide some rationale in the text (not just in response to the reviewer) to use the same terms for these different models, the Embryonic injury and growth model should be given another term, perhaps “developmental axonal growth”, “developmental axonal response to injury”, or something similar.

The eDRG culture model has been used extensively by us and others to study axon regeneration. While we agree that these are embryonic neurons with high growth capacity, our findings in eDRG have been recapitulated in large part in adult in vivo system (Cho et al, 2012, 2014, 2015). Our unpublished RNAseq analyses of such eDRG culture also suggest that in vitro axotomy recapitulates expression of the expected regeneration associated genes and downregulation of genes related to ion channels observed in vivo (Lisi et al 2017).

**OK, and thank you. This was informative. I'll look forward to seeing that RNAseq data published.

“For adult DRG cultures DRG were dissected from naive mice.”

- Provide the age/sex of the mice.

Age and sex is now provided in method section

**OK.

“Contralateral nerve and DRG served as uninjured controls, when needed.”

- Please indicate if the contralateral side was sham (and if so if it was skin incision with/without muscle dissection) or intact.

The contralateral is the uninjured side, not a sham surgery. Details is now provided in method section

**OK.

The model uses a 3d post-crush time for regeneration assessment. This seems to offer a limited picture. It does not allow for assessing the normal range of “initiation rates” known to exist across the different types of peripheral axon types. It limits this assessment to only the fastest responders, but the claims are made broadly. It is entirely possible that the neurons which are slower-initiators in the WT have an identical initiation rate in the KO, but this possibility is not addressed.

3d post crush allows us to relate our finding to most of ours and others' previous findings and is a time at which defects can be observed. Shorter time, such as 1d post crush, does not allow to see differences in KO models such as DLKKO, HIF1a cKO that affect activation of a pro-regenerative program (see Cho et al 2015 and Shin et al 2012). 3 days allows the regeneration program to be turned on and if impaired, defects can be observed. We also previously showed that nociceptor growth is similar to other neuron types growth (Carlin et al 2019, Abe et al 2010) Examination of other time points and neuronal subtypes is beyond the scope of the current study. The possibility that different neuronal subtypes are differentially affected by PPAR α signaling SGC remains is now been mentioned in the discussion, p. 18.

**OK. The point on p. 18 addresses this concern satisfactorily, and I entirely agree that the point does not warrant any additional data collection.

The images of the nerve in the figure imply a rather significant qualitative difference in the architecture of the distal nerve, with the KO nerve looking much closer to normal (with visible

undulations), and even with what appear to be SCG10+ structures far distal to the injury site. This should be addressed better. Providing images of axonal structure (beta-3 tubulin?) would significantly enhance the context for readers.

We have added images of nerve sections stained for Tuj1 from control and FasncKO in figure S4a, showing that the overall architecture of the nerve is similar.

Arrow heads have been added to Fig. 5a and 7c to emphasize the longest regenerating axons.

“...the injury site was defined as the area with maximal SCG10 intensity.”

- Please indicate if this was established from a single section, or a consensus across many sections. Also indicate what qualified as “area with maximal SCG10” – a single pixel? 10 contiguous pixels? Etc. This method of defining the injury site may have been convenient, but it is not (at least as presented here) terribly clear or strong. Using markers of denervated Schwann cells or infiltrating cells, even as a proof of concept for the SCG10-intensity method, would have strengthened the assessment significantly. This point of clarity matters greatly as it is the basis for the quantitative measures.

This is an assay that we and others have used extensively. The crush site is defined as the site along the nerve length where SCG10 intensity is maximal when measured in a vertical line across the nerve. The quantification assay is explained in details in methods, p. 34. SCG10 was chosen, because it selectively labels regenerating axons and displays higher specificity than GAP43 or YFP in the early stage of axon regeneration after nerve crush (Shin et al 2013).

**I understand that this method has been used previously. That does not address the main concern, though, and, as stated previously, clarity in this methodological point matters greatly. Regarding simply referencing prior publications: the Shin 2012 article appears to have even less detail with regard to this method than does this manuscript. The Cho 2015 article states “To assess the regenerative capacity of injured axons, the ratio of SCG10 fluorescence intensity proximal and distal to the axotomy line was measured...”, but it does not actually state how the “axotomy line” was determined to any greater degree than this manuscript. Figure 4 legend states “Dotted lines indicate the crush site, identified as the maximal SCG10 intensity”, which simply raises the same questions. The Methods section states “SCG10 fluorescence intensity was measured along the length of the nerve using a line scan macro in ImageJ”, which would be an excellent addition to this manuscript because it suggests that the “highest intensity” was determined objectively (which was not clear before), but still leaves some questions. How many line-scans were used? How does the procedure deal with a situation in which 2 or more pixels across the width of the nerve section all have the same or very similar “highest-intensity” yet are spaced far-apart along the length of the nerve section?

** Therefore, please state clearly in this manuscript:

1) if the “area with maximal SCG10 intensity” was determined for each section. If not done on a per-section basis, then state how the site was determined for the other sections in which the “maximal SCG10 intensity” was not determined. If it was done objectively as implied by the reference to the Cho paper, state that here as well, and add the clarification about how many lines were used and any procedures to deal with conflicts.

We now clarify in the revision (p.36) that the area with maximal SCG10 intensity was determined for each nerve section.

2) what qualified as “area with maximal SCG10” – a single pixel? 10 contiguous pixels? Etc. Perhaps it was a single pixel and the “injury-line” was then drawn perpendicular to the long-axis of the nerve section through that point?

We now clarify on p.36 that lines were drawn perpendicular to the long axis for every nerve section. About 5 lines (16X32 pixels per line) were drawn and the line with higher intensity was determined as the injury site.

3) if there was any procedure to ensure that the injury site determined within any single section corresponded to those determined in others – i.e., they were not 2mm apart. This addresses the possibility that the “injury site” determined in each section may define an injury that does not reflect physical reality. This could be as simple as something like ensuring that all SCG10-defined injury sites were within the same distance (which should correspond very closely to the width of the forceps used to crush the nerve) from the cut-end of the nerve sample, or something similar.

Charcoal was used while performing the crush injury to label the injury area after tissue fixation and sectioning. Image analysis procedure are now clarified in the method section p.36 as requested.

- Certainly there are differences detected, but the 3 day post-injury time point seems sub-optimal for the claims made. It is at the leading edge of what is reported for the latency to the cell-body reaction and start of axonal regeneration, which would fit well if the only goal were to assess the fastest response-initiation. Although that is included, the manuscript also uses these data to support claims regarding the regeneration process in general. Assessments at later time points would have greatly strengthened the claims. The claims throughout the manuscript regarding regeneration should be revised to reflect this point.

We have now included in the discussion that whether SGC contribute to sustain regenerative growth until target reinnervation remains to be determined (p. 18).

**OK.

“Neurons in injured FasncKO displayed reduced neurite length compared to injured controls, but similar initiation rates (Figure 5D-F).”

- The outcome measure relies on the “longest 10 axons”. There is nothing intrinsically flawed in this, but it does introduce a bias toward the most effective and/or fastest-starting axons. This is problematic because these may be a subpopulation, or the effect of Fasn/PPAR may only be on the

Figure 5D-F is in vitro growth, and the measure is axon length per neuron, not longest 10 axons

**YES! Apologies – I incorrectly merged these 2 things.

- It is possible that the “initiation rate” was unaffected by the Fasn-KO as claimed, but there are other interpretations that could exist (false-negative). As described, the “initiation rate” was determined by the post-plating time at which neurites emerged to some qualifying characteristic. This was done without regard to the type of sensory neuron giving rise to those neurites. Unless all SGC express the same level of Fasn, and all DRG neurons express the same level of PPAR, then it is possible that the neurons that are responsible for providing the “equivalent initiation rate” are a subpopulation that, in WT, have some reduced/different PPAR-expression/function and/or are surrounded by SGC with lower Fasn (Figure 3 appears to indicate that there may be some significant heterogeneity in the Fasn expression). In this case, the loss of Fasn (in the KO) would largely be irrelevant as that subpopulation did not have a high degree of SGC-Fasn and neuronal PPAR anyway, and the conclusion could be incorrect. This can be addressed either by providing data regarding the uniformity of the PPAR-expression across neuron types AND the uniformity of Fasn-expression across SGC, OR the assessment of “initiation rate” can be done with sensitivity to neuronal sub-types. Alternatively, the manuscript can alter the claim and provide better

presentation of the possible interpretations of the data.

We have revised the text and rather than initiation rate, refer to percent of neuron initiating axon growth in Fig. 5f and 7h. Whether SGC differently affect neuronal subtypes is now discussed (p.18). The uniformity of Fasn-expression across SGC might be interesting. However, without a morphological correlation of SGC surrounding a given neuron, this information might not explain the effect of Fasn/PPAR expression in different SGC on different subtypes. Whether different SGC subtypes surround different neurons subtypes is a major ongoing study in the lab, but is beyond the scope of the current manuscript.

**OK.

- Using the in vivo outcome to determine “initiation” with the in vivo outcome of “length” as was done leads a particular problem. There is no means for determining if the populations are the same. This again is not necessarily a flaw, except that the narrative regarding both outcomes implies both a uniformity across the entire population of neurons, and implies a union of the population of axons that are longest and first to grow in vitro. These implications are not supported by the experimental design, though they may be true. Therefore, either new data must be provided demonstrating the uniformity and union, or the narrative must be adjusted to clearly provide to the reader an interpretation of the data that includes these possibilities and limits.

We agree that which population of neurons is examined in the in vitro and in vivo assays is not determined in our current experiments. We have added a comment to this effect in the discussion on p.18. As stated above, whether SGC affect neurons subtypes differently is an ongoing study in the lab, but is beyond the scope of the current manuscript.

**I made some errors in stating my original concern which may have made this difficult to understand. Apologies if this was the case. It would be ideal if the manuscript acknowledged the fact that the in vivo and in vitro assays may not be dealing with the same population while the conclusions assume they are the same, but this is not vital. The rebuttal is acceptable nonetheless.

Thank you

The outcome measure of “initiation rate” may have additional issues. It appears to be less about the latency of when the range of axons start to regenerate and more about the proportion of neurons that display neurite-outgrowth at a given time. It is not clear that “rate” is a suitable term in the in vitro case. Perhaps “proportion” or similar term would be more accurate.

We agree and have revised the text and rather than initiation rate, refer to percent of neuron initiating axon growth in Figure 5f and 7h.

**OK.

“These results indicate that Fasn in SGC is required for the conditioning effect, and specifically for the elongating phase of axon growth”

- The conclusion of “required” is not supported. Fasn may influence the priming effect, but even this may not be well-supported. Fasn may regulate, to some unknown degree, the priming effect in some population of neurons. However, the limited assessment time does not lend itself to a thorough assessment of priming or regeneration, particularly the elongation phase.

We agree and have use “contribute”

**OK.

“...SCG10 intensity was quantified at 1, 2, and 3mm from the injury site as previously described”

- A brief description needs to be provided. It is not clear if the quantification was the average intensity of a row/line of single pixels (e.g., the demarcation line), or some area of pixels.

Details on quantification are now provide in the method section

**OK.

“For both LENGTH AND INTENSITY quantifications, five sections per biological replicate were averaged.”

We made these changes

**OK.

In standard situation, PPAR-g is neuronal and PPAR-a is SGC. However, it appears that this pattern was not ensured in the Fasn-cKO. This weakens the conclusion (assumption?) that the fenofibrate (PPAR-a-agonist) is working only on SGC, especially since the effective concentration of fenofibrate was not determined. This weakness is mitigated by the data that fenofibrate does not improve axon regeneration in neuronal cultures that do not contain SGC. The authors should consider including a statement to clarify the context, or doing some direct assessments.

We have included statement in the results and discussion acknowledging the fact that fenofibrate might act on other cells beyond SGC (p.16 and p.19)

**OK.

DISCUSSION

“The role of SGC in nerve regeneration has been largely ignored...”

The term “ignored” is inaccurate. It implies that the role was known but not considered. But the role was not really known until this work – this is even stated in the following sentence. Please reword.

We agree and have revised the text accordingly

**OK.

“Although addressing this question will require further investigation, our data indicate that Fasn expression in SGC is required for the proper expression of at least one regeneration associated gene after injury, Atf3”

- The conclusion of “required” is not supported.

We agree and have revised the text accordingly

**OK, but please correct “participate” to “participates” and “gene” to “genes”

Corrected as suggested, Thank you

“Our study suggests that neuronal “intrinsic” response to injury may not be purely neuronal but rather includes the SGC surrounding the neuronal soma...”

- Unless this statement is referring to something other than the neuronal ATF3 response (which the authors have not specifically identified) as the “intrinsic” response, then the statement is not supported.

We agree and have revised the text accordingly

**OK.

METHODS

“...and both injured dorsal root ganglia and sciatic nerve were dissected at the indicated time post-surgery”

- Specify which DRGs were designated as “injured” and collected. The designated DRG’s were described in figure legends and main text of the original manuscript. This information is now

also added to the methods section.

**OK.

Specify the control procedures for immunohistochemistry and western blot.

The details for control procedures were added

**OK for immunohistochemistry. Not added for western blot.

Gapdh (Santa Cruz, catalog# sc25778) was used as loading control for quantification of the western blots, Fasn intensity was normalized to the intensity of Gapdh. Control procedures for western blot is included in methods (p.38) figure legend (Fig.3f).

“Cultures were then used for electrophysiological recording 24h after plating or fixed and stained with the indicated antibody”

- Electrophysiological recordings starting 24h after plating does not qualify as recording from “acutely dissociated” DRG neurons, as indicated in the Results.

Agreed and changed to “short-term cultures”

**OK.

Figure 4H shows results from qPCR of ATF3, but there are no primer sequences provided for ATF3. Please provide the primer information.

ATF3 primers sequence were added

**OK.

“An Automated analysis for axon tracing and neurons soma count was used for ex-vivo adult DRG culture experiments (Nikon elements).”

- Please provide specifics for the “automated analysis”.

Nikon elements is a commercial software package for image analysis using an algorithm for automatic digital reconstruction of axons. The specific analysis code is available upon request (requires NIS-Elements and General Analysis).

** Yes. We use Elements as well, as do many others. What is needed here is not the algorithm or code, but the user interaction parameters and procedures. Even with an “automated procedure”, the user does have to teach Elements what features of the image are of interest (i.e., to include in measurements). In many cases of analyzing images from in vitro systems, a thresholding step is used. Utilizing fluorescent signal to create a binary layer, through thresholding, allows a user to teach Elements biological features such as an axon and a soma. In this case the user must define for Elements what the threshold is, and the user must somehow define the thresholding settings. In many cases, there is a background correction involved. Also, there can be manual or automatic selection of “regions of interest” or “objects”, and often a removal of “non-relevant signal” such as axon debris. On the acquisition side, it is not clear that there was a common set of image capture protocols used across conditions.

Perhaps the lab uses a module called “General Analysis” that can make automation a bit easier. Even if so, users can share these recipes with other users that would allow them to replicate the analyses (assuming the acquisition parameters are also replicated).

The manuscript must include a description of these and other relevant procedures.

We absolutely agree; description of user interaction parameters and procedures have been added in the Method section (p. 39)

“Statistics was performed using GraphPad (Prism8) for either Student’s t-test or ANOVA analysis.”

- This is not sufficient. Please ensure that this section is complete and that the description of the statistics in Methods and elsewhere are consistent and clear.

- Indicate type of ANOVA

- Perhaps the authors have omitted mention of the post-ANOVA tests in the description in the Methods. Figure 2 legend mentions the Sidak multiple comparison test, suggesting this terse description in the Methods is in error. Sidak or Dunnett tests were used as part of one- and two-way ANOVA for multiple comparison tests.

A description of the statistical tests was added

**OK.

- Gene expression mentioned use of FDR, but this is not described. An FDR value is a p-value adjusted for multiple tests (by the Benjamini-Hochberg procedure). It stands for the “false discovery rate” it corrects for multiple testing by giving the proportion of tests above threshold alpha that will be false positives (i.e., detected when the null hypothesis is true). A description was added.

**Yes, I’m aware of what FDR is, but thank you for the explanation in case I was not. The comment was because the abbreviation FDR was not defined, and it was not mentioned as part of the statistical procedures. Looks OK now.

Thank you

- Most legends indicate which statistical tests were used, but not all. Figure 4 legend has p-values but no indication of the tests used

A description of the statistical tests was added to Fig. 4 legend

**OK.

Minor

“Structural neuron-glia units, similar to these, do not exist in the central nervous system”

Commas likely not necessary

Commas were removed

**OK.

Throughout, please ensure that the language consistently and properly uses the terms “gene” and “transcript”. In many places the term “gene” is used where “transcript” should be used. The Chromium Single-Cell 3’ Solution we used for the scRNASeq can recognize a general gene but not a unique transcript or splicing variants, therefore we chose to use the term ‘gene expression’ and not ‘transcript expression’ to avoid misleading the reader.

**OK. I see the point. I think the greatest clarity would come from stating the above in the manuscript.

We agree, and this is now stated in manuscript in the Method section (p.33)

“Our single cell RNAseq analysis using freshly dissociated tissue thus UNRAVELS a unique transcriptional profile of SGC in DRG...”

corrected

“We found that SGC share about 10% of THOSE gene TRANSCRIPTS with astrocytes, among them Fabp7...”

corrected

“We crossed BLBPCre-ER to the Rosa26-fs-TRAP and observed that following 10 days OF tamoxifen treatment...”

OR

“We crossed BLBPCre-ER to the Rosa26-fs-TRAP and observed that following A 10 day tamoxifen treatment...”

corrected

“Pathway analysis of other major CELL types in the DRG confirmed fatty acid metabolism as a unique pathway enriched in SGC...”

corrected

“...our pathway analysis INDICATED that the cell cycle term is enriched in macrophages, but not SGC...”

Also, this sentence is awkward. The “cell cycle term” is not enriched in macrophages. Please reword.

corrected

“...loss of Fasn in SGC does not lead to morphological deficits in the DRG and does not ELICIT a stress response in neurons”

corrected

“Fenofibrate is used clinically to treat lipid disorders, but has been unexpectedly been shown in clinical trials...”

- “been” is redundant and grammatically incorrect

corrected

“During surgery, 8-12 week old C57Bl/6 mice OF the indicate genotype were anesthetized”

Corrected

**OK to all.

Reviewer #4 (Remarks to the Author):

The authors have addressed my concerns appropriately. I am happy to endorse this manuscript for publication in Nature Communications.

We are pleased that the revised version has fully satisfied this reviewer and that he endorses our manuscript for publication in Nature Communications.

Reviewers' Comments:

Reviewer #2:

Remarks to the Author:

The specific expression of PPAR α in SGCs made the argument. Good to go.

Detailed responses to reviewers.

Reviewers' comments are in black. Authors' responses are in blue.

Reviewer #2 (Remarks to the Author):

The specific expression of PPAR α in SGCs made the argument. Good to go.

We are pleased that this revised version has fully satisfied this reviewer.

In relation to the specific expression of PPAR α in SGCs, we cited in our response a recent study in BiorXiv. This manuscript has since then been published in Neuron (Renthal et al, 2020). We included this reference in our manuscript on p. 14 to further support the specific expression of PPAR α in SGC.